# The Two-Hump Problem: Bridging the Difficulty Gap in Mathematical Reinforcement Learning

**Lucas Fagan** [* 1]  **Michele Tarquini** [* 1]  **Ali Shehper** [* 1]  **Maksymilian Manko** [2]  **Angus Gruen** [3]  **Coco Huang** [4]
**Giorgi Butbaia** [1]  **Davide Passaro** [1]  **Sergei Gukov** [1]

## Abstract

Mathematical search problems present a unique challenge for Reinforcement Learning (RL) due to vast search spaces and sparse rewards. In previous works, the Andrews-Curtis (AC) conjecture was established as an illustrative example of such problems. In this work, we identify a critical structural barrier in the AC landscape: a "Two-Hump" distribution, where problem instances are either trivially solvable or effectively impossible, with a scarcity of intermediate "hard-but-solvable" instances required for effective learning. We tackle this challenge through two primary avenues: novel data generation techniques to populate the difficulty gap, and significant algorithmic enhancements including the introduction of supermoves and Transformer-based architectures. We demonstrate substantial performance improvements over previous baselines, and release new comprehensive benchmark datasets including **AC-19** (125,192 AC-trivial presentations of varying difficulty with length at most 19) and **AC-1M** (1,136,154 hard AC-trivial presentations of length at most 30), the first large-scale, publicly available datasets of this kind.[1]

## 1. Introduction

The application of reinforcement learning to mathematical reasoning presents challenges fundamentally different from classic domains like Go or StarCraft. In mathematical search problems, there is often no "early failure" signal; an agent can wander indefinitely without knowing if it is making progress.

The Andrews–Curtis (AC) conjecture (Andrews & Curtis, 1965) offers a distilled environment for studying these challenges. It posits that any balanced presentation of the trivial group can be transformed into the trivial presentation via a specific set of moves. Previous work (Shehper et al., 2025) framed this as a Reinforcement Learning (RL) environment, highlighting the difficulty of finding rare trivialization paths.

In this work, we present a deeper analysis of *why* the AC problem is so difficult for RL agents and propose a comprehensive suite of improvements to address these challenges. We combine three key insights: (1) a characterization of the fundamental structure of the problem's difficulty landscape, (2) algorithmic innovations that exploit the mathematical symmetries of group presentations, and (3) targeted data generation techniques to populate the sparse regions of the problem space. Together, these contributions enable us to trivialize over 100 previously unsolved presentations from a standard benchmark and make concrete progress toward resolving this longstanding mathematical conjecture.

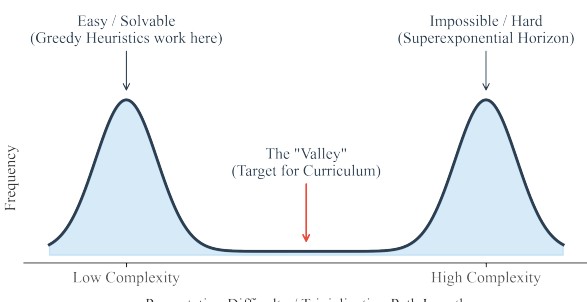

*Figure 1.* The "Two-Hump" Distribution is a general phenomenon: most presentations are either trivial or practically impossible for current agents. The scarcity of data in the middle region hinders learning.

### 1.1. Contributions

Our specific contributions are:

---

[*]Equal contribution  [1]Department of Mathematics, California Institute of Technology, Pasadena, CA, USA [2]Institute of Mathematics, University of Zurich, Zurich, Switzerland [3]Zero Latency Labs [4]Department of Mathematics, Temple University, Philadelphia, PA, USA. Correspondence to: Lucas Fagan <lfagan@caltech.edu>.

*Proceedings of the $43^{rd}$ International Conference on Machine Learning*, Seoul, South Korea. PMLR 306, 2026. Copyright 2026 by the author(s).

[1]Code and datasets: https://github.com/Math-AI-Caltech/ACSolverX

**Identification of the "Two-Hump" difficulty distribution.** We characterize a fundamental challenge in the AC problem: a bimodal difficulty distribution with a large mass of "easy" presentations solvable by greedy length reduction, a large mass of "effectively unsolvable" presentations where no current algorithm makes meaningful progress, and a sparsely populated "valley" containing presentations that require sophisticated planning but remain within algorithmic reach (Figure 1). This lack of intermediate stepping stones makes standard curriculum learning ineffective and motivates our approach combining algorithmic improvements with targeted data generation.

**Substitution supermoves as the right action space.** We identify that *substitutions*—a way of combining many primitive AC-moves into a single move—form the right action space for the AC problem. Using substitutions allows for much larger steps and a significantly reduced state space at the cost of an expanded action space. This insight is critical to all of our subsequent improvements.

**Scaling RL to large action spaces with domain-specific architecture.** Using substitutions presents an opportunity as it creates a larger but also more highly structured action space. To take advantage of this, we developed the *Dual-Ring Transformer*, a specialized architecture that processes the two relators as cyclic sequences with explicit cross-attention mechanisms to identify optimal substitution moves. Despite the significantly larger action space, our PPO agent achieves substantial improvements over the prior approach (Shehper et al., 2025), and consistently finds shorter solution paths than classical searches, with the improvement especially pronounced on harder presentations. This demonstrates that reinforcement learning can discover non-obvious mathematical structure even in challenging search spaces.

**Targeted data generation and novel benchmark datasets.** We obtain high-quality training data in the difficulty "valley." In particular, we use exhaustive enumeration to produce **AC-19**, the largest known dataset of short AC-trivial presentations. It consists of 125,192 presentations of varying difficulty and length $\leq 19$, which were verified through our improved substitution-based classical search methods. We also show how automorphisms can be used to generate hard-but-solvable presentations by classical methods and by an ML-based generator-solver game, and combine these methods to produce **AC-1M**, a dataset of 1,136,154 hard AC-trivial presentations of length at most 30.

**Concrete mathematical progress.** This work makes concrete progress on the Andrews–Curtis conjecture itself through organizing and reducing the hard hump. Our technical improvements allowed us to successfully trivialize over 100 new presentations in a benchmark dataset from the Miller–Schupp family—a famous two-parameter family of potential counterexamples—whose status was previously unknown. Among the 550 remaining unsolved examples, we discover AC-equivalences that reduce these to just 261 equivalence classes and determine minimal representatives for each class, providing the community with a dataset against which to measure meaningful progress.

## 1.2. Related Work

**Computational Approaches to Andrews–Curtis.** The AC conjecture has attracted algorithmic attention since its formulation (Andrews & Curtis, 1965). Bridson (Bridson, 2015) and Lishak (Lishak, 2017) established that certain AC-trivial presentations require superexponentially long trivialization paths, revealing the fundamental computational difficulty. Havas and Ramsay (Havas & Ramsay, 2003) used exhaustive enumeration to classify all presentations up to length 13. Miasnikov (Miasnikov, 1999) and Bowman and McCaul (Bowman & McCaul, 2006) developed heuristic search methods. Most directly related, Shehper et al. (Shehper et al., 2025) framed AC as an RL environment and established PPO and greedy search baselines, on which our work builds. Concurrent work (Zhang et al., 2025) combines Lean-based formal verification and LLM-extracted lemmas with RL for AC trivialization.

**Sparse Rewards and Hard Exploration.** RL in sparse-reward environments remains challenging. Approaches include intrinsic motivation (Pathak et al., 2017; Badia et al., 2020), systematic exploration via archiving (Ecoffet et al., 2021), and hierarchical methods (Sutton et al., 1999; Vezhnevets et al., 2017). The Rubik's cube was solved via deep RL and search (Agostinelli et al., 2019), but this approach relies on the finite state space and the ability to generate training data via scrambling from the solved state. In contrast, the AC graph is infinite, and scrambling produces either trivially easy presentations or presentations that reduce back to known hard instances, providing no useful training signal.

**Related Mathematical Domains.** The unknotting problem shares structural similarities with AC (Gukov et al., 2021; Burton et al., 2024): both involve simplifying combinatorial objects through local moves with potentially deceptive length-based heuristics. Techniques may transfer between domains. In formal theorem proving (Yang & Deng, 2019; Polu & Sutskever, 2020; Yang et al., 2023; DeepMind, 2024; Trinh et al., 2024), curriculum learning has proven essential (Polu et al., 2023). However, these settings provide intermediate verification signals, whereas AC provides reward only upon complete trivialization, making curriculum generation more critical.

## Conflict of Interest Disclosure

The authors declare no financial conflicts of interest.

## 2. The Andrews–Curtis Conjecture and Substitutions

The Andrews–Curtis conjecture (Andrews & Curtis, 1965) is among the most prominent open problems in combinatorial group theory, and connects directly to low-dimensional topology: every potential AC counterexample yields a potential counterexample to the Smooth Poincaré Conjecture in dimension 4 (Freedman et al., 2010), and validating one would disprove the Generalized Property R conjecture (Gompf et al., 2010). Unlike many open problems at its level, AC admits an immediate, discrete, efficiently-checkable formulation as a pure search problem—making it an unusually strong testbed for AI-driven mathematical search.

The Andrews–Curtis conjecture concerns balanced presentations of the trivial group $\langle x, y \mid r_1, r_2 \rangle$, where $x$ and $y$ are called *generators* and $r_1$ and $r_2$ are called *relators* (words in the generators that the presentation declares to equal the identity). The conjecture states that any such presentation can be transformed into the trivial presentation $\langle x, y \mid x, y \rangle$ using a finite sequence of the following moves (called AC-moves):

- **(AC1)** Replace a relator $r_i$ with its inverse $r_i^{-1}$.

- **(AC2)** Replace a relator $r_i$ with $r_i r_j$ for $i \neq j$.

- **(AC3)** Replace a relator $r_i$ with $w r_i w^{-1}$ for any word $w$ in the generators.

Any presentation that can be connected to the trivial presentation using these moves is called *AC-trivial*. We call $|r_1| + |r_2|$ the *length* of the presentation.

This gives rise to a graph structure where vertices are balanced presentations of the trivial group and edges connect presentations related by a single AC-move. The problem of trivializing a presentation becomes a graph search problem: finding a path from the given presentation to the trivial presentation. The Andrews–Curtis conjecture asserts that this graph has only one connected component—that is, every balanced presentation of the trivial group can reach the trivial presentation through some sequence of AC-moves.

### 2.1. Potential Counterexamples and Benchmark Data

Mathematicians have proposed several families of balanced presentations that are suspected to be counterexamples to the Andrews–Curtis conjecture. The most famous such family is the Akbulut–Kirby family (Akbulut & Kirby, 1985):

$$\mathrm{AK}(n) = \langle x, y \mid xyx = yxy, x^n = y^{n+1} \rangle.$$

While AK(1) and AK(2) are known to be trivializable, the status of AK(n) for $n \geq 3$ remains open. The presentation

AK(3) is the shortest potential two-generator counterexample: in (Havas & Ramsay, 2003), it was shown that all presentations of length up to 13 are trivializable or AC-equivalent to AK(3).

A second family containing many potential counterexamples is the Miller–Schupp series (Miller & Schupp, 1999), defined for integers $n$ and words $w$ in $x$ and $y$ with 0 exponent sum on $x$:

$$\mathrm{MS}(n, w) = \langle x, y \mid x^{-1} y^n x = y^{n+1}, x = w \rangle.$$

Note that the MS series contains the AK series: AK(n) is AC-equivalent to $\mathrm{MS}(n, y^{-1} x^{-1} y x y)$ (Myasnikov et al., 2002).

**Validation Benchmark.** Prior work (Shehper et al., 2025) established a dataset of 1190 MS presentations as a benchmark for evaluating AC trivialization methods. We adopt this dataset as our primary *validation set* throughout this work, enabling direct comparison with previous results. However, we find that most presentations in this dataset are either easy or too hard, as shown in Figure 3. This is discussed further in Section 3, where we also describe novel data generation methods designed to populate the difficulty gap and facilitate robust training.

### 2.2. Baseline Algorithms

Prior work (Shehper et al., 2025) established two baselines for AC trivialization: greedy search and Proximal Policy Optimization (PPO). We briefly describe these methods here, as they form the foundation for our improvements.

**Greedy Search.** A natural baseline for exploring the AC graph is **greedy search**: a best-first search that prioritizes moves reducing the presentation length. The intuition is that shorter presentations are "closer" to the trivial presentation. In practice, greedy search expands nodes in order of length, exploring shorter presentations before longer ones, up to a fixed budget of visited nodes. Despite its simplicity, greedy search outperforms other classical approaches including breadth-first search and genetic algorithms (Miasnikov, 1999; Bowman & McCaul, 2006) on the AC problem. However, greedy search fundamentally struggles with instances requiring *length-increasing* intermediate steps, when presentations along the path must temporarily grow longer before eventually reaching the trivial presentation.

**Proximal Policy Optimization (PPO).** PPO (Schulman et al., 2017) is a policy gradient method for reinforcement learning that has demonstrated strong empirical performance across diverse domains. In (Shehper et al., 2025), PPO was established as the best-performing reinforcement learning algorithm for the AC problem, outperforming other RL methods including DQN and AlphaZero. The prior work used a ResNet architecture to represent the policy and value

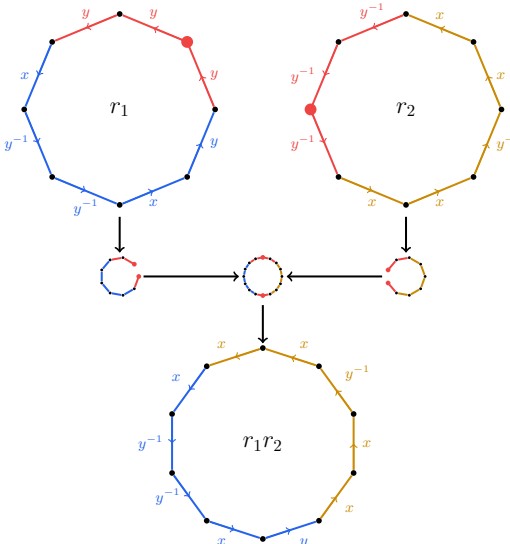

*Figure 2.* Substitution move visualization. Both relators are viewed as cyclic sequences (rings). A substitution selects insertion points in each relator where one can be inserted into the other with at least one element canceling, combining many AC1, AC2, and AC3 moves into a single action.

functions, operating on elementary AC-moves.

### 2.3. Substitutions

While the conjecture is defined in terms of elementary AC-moves, more efficient graph traversal can be achieved through *supermoves*—composite sequences of AC-moves applied as a single action. We identify *substitution moves* as particularly effective: a substitution, as shown in Figure 2, views both relators as cyclically symmetric strings and selects a point in each relator to insert one into the other such that at least one element cancels. Formally:

**Definition 2.1** (Substitution Move). A **substitution move** is parametrized by a tuple $(i, j, k_1, k_2) \in \{1, 2\} \times \{-1, 1\} \times \mathbb{Z} \times \mathbb{Z}$ and replaces the relator $r_i$ by:

$$\mathrm{rot}^{k_1}(r_i)\,\mathrm{rot}^{k_2}(r_{3-i}^j),$$

where $\mathrm{rot}^k(r)$ means rotating $r$ cyclically $k$ steps. We consider valid substitutions to be only those where the last element of $\mathrm{rot}^{k_1}(r_i)$ is the inverse of the first element of $\mathrm{rot}^{k_2}(r_{3-i}^j)$ (leading to a cancellation).

Because substitutions are indifferent to cyclic permutation, we can choose a canonical representative (classical algorithms) or use an architecture that is equivariant to cyclic permutations (RL). This dramatically reduces the effective state space while allowing algorithms to take larger, more meaningful steps through the AC graph. The rigorous definition of canonical form and its properties are provided in Appendix E.

Importantly, Ivanov (Ivanov, 2018) proved that if the AC conjecture is true, substitutions suffice to realize all AC-transformations.

**Implications for algorithms.** Substitution moves enable improvements to both classical search and reinforcement learning approaches. They allow for bigger steps in the graph and are indifferent to cyclic permutations of the relators, allowing us to significantly reduce the effective state space.

- **Improved Greedy Search:** We substantially improved greedy search by using substitution moves and reducing the state space by implementing a canonical form of the presentation, explained in detail in Appendix E. This new greedy search yields an approximately $1600\times$ exploration efficiency improvement over the baseline from (Shehper et al., 2025). Precisely, our greedy search with substitutions can solve all presentations that required 1M nodes in the prior work using only 614 nodes. Throughout the remainder of this paper, "greedy search" or "GS" refers to greedy search with substitution moves unless explicitly stated otherwise.

- **PPO with Large Action Spaces:** Extending PPO to use substitution moves requires a specialized architecture designed to respect the cyclic invariance of the relators while handling the significantly increased size of the action space. We develop the Dual-Ring Transformer, introduced in Section 4, to address this challenge and produce a more effective agent compared to the PPO baseline with basic AC-moves (Shehper et al., 2025).

## 3. The Landscape and Data Generation

With the improved tools discussed in Section 2.3, we can now systematically analyze the difficulty landscape of the Andrews–Curtis graph.

### 3.1. The Two-Hump Phenomenon on the Miller–Schupp Benchmark

Our experiments on the 1190-presentation Miller–Schupp benchmark dataset from (Shehper et al., 2025) reveal a stark bimodal distribution we call the "Two-Hump" phenomenon. As shown in Figure 3, presentations exhibit a polarized outcome: they are usually either easily trivializable or effectively impossible to solve with current methods. Using our improved greedy search with substitutions (see Table 1 for performance details), we find that 604 presentations are solved within 10,000 nodes. However, increasing the node budget $100\times$ to 1 million nodes solves only 36 additional presentations and extending the budget to 10M gives no additional improvement, leaving 550 still unsolved. In other

words, there are very few hard-but-solvable presentations in the intermediate "valley" between the two humps.

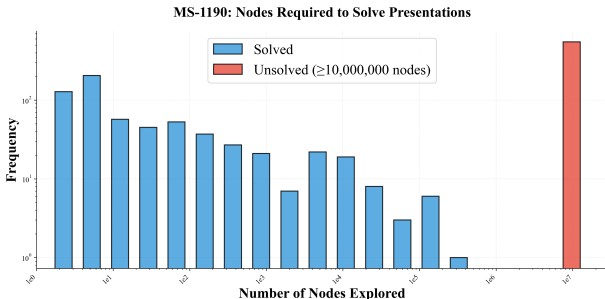

*Figure 3.* Empirical validation of the Two-Hump phenomenon on the 1190-presentation Miller–Schupp benchmark. The histogram shows solve difficulty (measured by greedy search node count) using **logarithmic scales on both axes**, suggesting a super-exponential difficulty distribution with a potentially unsolvable end. Difficulty exhibits a stark bimodal distribution: most presentations are either trivially easy (solvable in $< 1000$ nodes) or effectively unsolvable (no method solves them within 10M nodes), with very few presentations in the intermediate "valley" range. The improved greedy search with substitutions achieves a $1600\times$ efficiency improvement over baseline methods (see Table 1), yet the two-hump structure persists.

This bimodal distribution explains the failure of naive data generation strategies. For example, generating data by "scrambling" the trivial presentation (similar to shuffling a Rubik's cube from the completed state) produces presentations which are easily trivializable through greedy length reduction. Similarly, scrambling known hard presentations like AK(3) produces presentations trapped in the same local minimum: our algorithms simply reduce them back to AK(3), providing no useful signal for learning.

### 3.2. AC-19: Exhaustive Enumeration Reveals Persistent Two-Humps

Given the sparsity of intermediate-difficulty presentations in the Miller–Schupp benchmark, we needed a larger and more comprehensive dataset. We enumerated 213,946 balanced presentations of the trivial group of length at most 19 using techniques first introduced in (Havas & Ramsay, 2003). We then ran greedy search with substitutions (10M node budget) on the entire enumeration and retained the 125,192 presentations that were successfully solved.

This **AC-19** dataset represents the largest known collection of short AC-trivial presentations, providing significantly more presentations overall and many more hard presentations than the Miller–Schupp benchmark. Yet, as shown in Figure 4, the two-hump phenomenon persists: while the dataset contains presentations spanning a wide range of difficulties, the bimodal structure remains evident. The distribution shows strong concentration in the easy regime with a second hump of unsolved presentations that would require

at least 10M nodes to potentially solve.

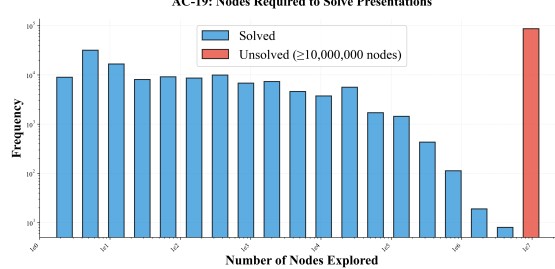

*Figure 4.* **AC-19 Dataset Difficulty Distribution.** Node counts required to solve the 125,192 verified trivial group presentations using greedy search with substitutions, displayed with **logarithmic scales on both axes**. Although AC-19 is a much larger dataset with many more hard presentations than the Miller–Schupp benchmark, the two-hump phenomenon persists: strong concentration in the easy regime (median: 51 nodes) with a substantial tail extending to 10M nodes, and a second hump of unsolved presentations requiring at least 10M nodes.

### 3.3. Populating the Valley: Automorphisms and the Generator-Solver Game

The persistence of the two-hump phenomenon across both the Miller–Schupp benchmark and the much larger **AC-19** dataset suggests this feature is inherent to the Andrews–Curtis problem landscape itself, rather than an artifact of insufficient data. The scarcity of hard-but-solvable presentations in the intermediate valley represents one of the biggest barriers to an agent that is able to learn general behavior to trivialize presentations in the second hump.

To address this fundamental challenge and continue building on **AC-19**, we develop a technique that leverages the mathematical structure of the problem to systematically generate presentations that populate the sparse middle difficulty region. These datasets improve PPO agents and establish a foundation for scaling to more capable learners. As algorithms improve, the availability of diverse, well-calibrated training data will become increasingly critical.

**Automorphism Perturbation.** An automorphism $\phi$ is a mapping from a presentation of the trivial group $P = \langle x, y \mid r_1, r_2 \rangle$ to another $\phi(P)$. In particular, if $P$ is trivializable via a path $\gamma$, then $\phi(P)$ is trivializable via $\phi(\gamma)$.[2] However, the characteristics of such paths may differ drastically. In particular, it is often the case that short paths from automorphic images require steps of large length increase.

Figure 5 demonstrates this phenomenon empirically: starting from the easily-solvable AK(2) presentation, we apply different automorphisms $\phi_i$ to generate AC-equivalent pre-

---

[2]With a few extra steps added at the end to go from $\langle x, y \mid \phi(x), \phi(y) \rangle$ to $\langle x, y \mid x, y \rangle$. The mathematical background regarding automorphisms is explained in Appendix D.

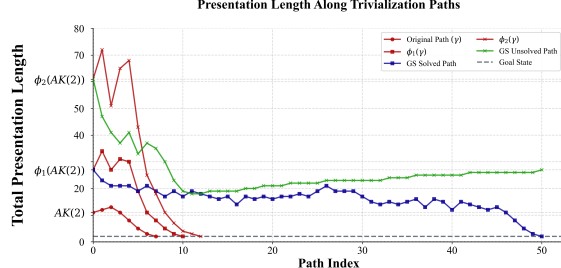

Figure 5. Automorphisms transform easy presentations into hard ones. While a greedy heuristic can easily find a trivialization path for AK(2), it struggles on automorphisms of AK(2) whose short trivialization paths (shown in red) involve length-increasing steps.

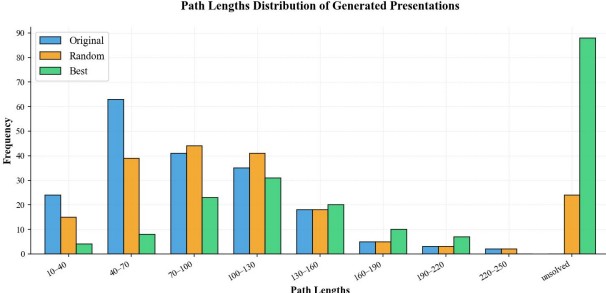

Figure 6. Comparison of difficulty distributions for 200 AC-trivial presentations from AC-19 under different transformations (measured by greedy search path length, 10K-node budget). **Original**: Baseline presentations. **Random**: Applying up to 10 random elementary automorphisms (capped at length 30). **Best**: Applying automorphisms selected by the trained PPO generator network. The trained generator successfully shifts the distribution rightward (harder presentations) and makes many presentations unsolvable within budget, demonstrating it learns to produce systematically harder instances than random automorphisms; this pattern persists at scale.

sentations with dramatically different path characteristics. While short paths exist for all three presentations, if we do not use our knowledge of the automorphism, the short paths for $\phi_1(\text{AK}(2))$ and $\phi_2(\text{AK}(2))$ are very difficult to find. Thus, by applying automorphisms to simple presentations, we can generate a range of more difficult presentations which we know are AC-trivial.

**The Generator-Solver Game.** Given the success in using automorphisms to make easy presentations more difficult, we formulated a two-player asymmetric game where a Generator applies automorphisms to increase the difficulty of presentations while a Solver attempts to trivialize them. This setup relies on the fact that all automorphisms can be generated by a sequence of elementary automorphisms, as discussed in Appendix D.

The *Generator* is a PPO agent with a dual-head feedforward architecture. One head outputs the number of automorphisms to apply, while the other samples from generating automorphisms: inversion ($x_i \mapsto x_i^{-1}$), or concatenation ($x_i \mapsto x_i x_j$ for $i \neq j$). To prevent length explosion from repeated concatenation moves, we impose hard constraints on presentation length. The Generator's reward combines three complexity measures from the Solver's feedback: whether the presentation was solved (incurring a penalty), path length if solved, and maximum length increase along the solution path.

We use greedy search as the *Solver*. In Figure 6, we show a small example on 200 presentations with a 10,000 node budget, where the generator is able to generate significantly harder presentations, including when compared to random automorphisms. By construction, every Generator output is an automorphic image of a known-trivial presentation and therefore carries a known solution path.

In order to populate the "valley," we train a generator using this method on all presentations from **AC-19** solvable within 5–10k nodes, and then use it to generate 10 automorphic images of each presentation by applying sequences of automorphisms, retaining only those of length at most 30. This

gives a new dataset of hard presentations, **AC-1M**, all of which are AC-trivial by construction. See Appendix C for full architectural details and training procedures.

## 4. The Dual-Ring Transformer

To operate effectively in the large substitution action space while respecting the mathematical structure of the Andrews–Curtis problem, we develop the **Dual-Ring Transformer**. The key insight is that relators are cyclic sequences, not linear ones, as illustrated in Figure 2. A network that respects this cyclic symmetry can dramatically reduce the effective state space and generalize better across presentations.

### 4.1. Architecture Overview

The Dual-Ring Transformer processes a presentation $\langle x, y \mid r_1, r_2 \rangle$ through three stages:

**1. Embedding.** Each relator is represented as an array of length 24 where $x, y, x^{-1}, y^{-1}$ become $1, 2, -1, -2$, respectively, and padded with 0s if necessary. Each token is then embedded into a learned vector of dimension $D$. We do not consider presentations with a relator of length greater than 24 (less than 1% of AC-1M); note this is distinct from *presentation length* $|r_1| + |r_2|$, capped at 30 in AC-1M.

**2. Dual-Ring Transformer Blocks.** A stack of transformer blocks processes both relators in parallel. Each block consists of:

- **Self-attention with relative positional encoding:** Each relator attends to itself using cyclic relative position embeddings. For tokens at positions $i$ and $j$ in a

relator of unpadded length $L$, the cyclic distance along the relator is $d(i,j) = (i-j) \bmod L$. The attention score combines content and relative position:

$$A_{ij}^h = (Q_i^h \cdot K_j^h) + (Q_i^h \cdot E_{d(i,j)}^h)$$

where $E^h \in \mathbb{R}^{L \times D_h}$ is a learnable embedding table per head. This gives the self-attention layer a cyclic-rotation-equivariant inductive bias.

- **Cross-attention:** The two relators attend to each other via cross-attention. This allows the network to identify matching substrings between $r_1$ and $r_2$, which is critical for determining valid substitution moves.

- **Feed-forward layers:** Standard MLP blocks with residual connections and layer normalization.

**3. Policy and Value Heads.** After $k$ transformer blocks, we obtain contextual embeddings for each position in both relators.

**Value head:** For each relator, we apply masked mean pooling over the sequence dimension (averaging only over non-padding positions). The resulting two pooled vectors are concatenated and passed through a 2-layer MLP to output a single scalar value estimate.

**Policy head:** For each pair of positions $(k_1, k_2)$ in $r_1$ and $r_2$, we concatenate the corresponding embeddings and pass through an MLP to produce logits for 4 possible actions:

- Which relator to replace: $i \in \{1, 2\}$

- Whether to invert $r_2$: $j \in \{-1, 1\}$ (corresponding to using $r_2^j$)

This gives a distribution over $|r_1| \times |r_2| \times 4$ possible substitution moves. Crucially, we apply a **semantic mask** that sets invalid actions to probability zero: only positions $(i, j)$ where $r_1[i] = \pm r_2[j]$ (enabling cancellation) receive nonzero probability. This enforces the mathematical constraints of valid substitution moves. Full architectural details are provided in Appendix A.

## 5. Experiments

We evaluate our methods on the 1190-presentation Miller–Schupp benchmark dataset from (Shehper et al., 2025). Our experiments are designed to isolate the contributions of our key innovations: substitution supermoves with the Dual-Ring Transformer architecture, targeted training data from **AC-19** and **AC-1M**, and enforcing cyclic symmetry through the Dual-Ring Transformer as opposed to absolute positional encoding and canonical form.

### 5.1. Experimental Setup

**PPO Training.** We train PPO agents using the Dual-Ring Transformer architecture (Section 4, full details in Appendix A) on data from the benchmark dataset and from Section 3. The key training details are:

- **Reward function:** We use a length-based reward: $r(s, a, s') = 1000$ for reaching the trivial presentation, and $\max(-(|r_1'| + |r_2'|), -10)$ at each step otherwise, where $|r_i'|$ denotes relator length in state $s'$. This encourages length reduction close to the trivial presentation and incentivizes finding shorter paths. It is similar to the most effective reward function in (Shehper et al., 2025).

- **Setup:** We train for 250M environment interactions across 2,380 parallel environments. Episodes terminate after 96 steps without trivialization, after which the environment resets. The first 640 parallel environments are reserved for the 640 presentations from the benchmark known to be AC-trivial (determined via greedy search). The remaining environments sample from **AC-19** or **AC-1M** using an adaptive sampling strategy (Appendix B). We found that this allocation outperformed using all 1190 benchmark presentations: we never solved any presentation outside the initial 640 AC-trivial set. More experiment details can be found in Appendix B.

- **Evaluation:** We report the min, max, and mean number of solved presentations from the benchmark dataset over 5 random seeds in Table 1.

**Implementation:** All algorithms are implemented in JAX (Bradbury et al., 2018) and our PPO is based on the Pure-JaxRL implementation (Lu et al., 2022). Code is available at `https://github.com/Math-AI-Caltech/ACSolverX`.

### 5.2. Main Results: Substitutions, Architecture, and Data

Our experiments demonstrate five main results:

**1. Substitutions + Dual-Ring Transformer yield a large performance increase.** The combination of substitution supermoves with the Dual-Ring Transformer yields dramatic improvements: PPO solves 591 presentations (max over 5 seeds) compared to 457 for the baseline, an increase of 134 presentations. This validates that the architecture can effectively handle the large $O(|r_1| \cdot |r_2|)$ action space.

**2. Targeted data provides a meaningful boost.** Adding **AC-19** and **AC-1M** data improves PPO performance from 591 to 610 presentations (max over 5 seeds), an additional 19 solves. Given that we are operating in the hard regime

*Table 1.* Performance on the 1190-presentation Miller–Schupp benchmark. **GS-AC** *(previous SOTA, greedy)*: Greedy search with AC-moves from (Shehper et al., 2025). **GS-Sub**: Our greedy search with substitution supermoves at varying node budgets (parentheses indicate search budget). **PPO-AC-ResNet** *(previous SOTA, RL)*: PPO with AC-moves and ResNet architecture from (Shehper et al., 2025). **PPO-Sub-DRT**: Our PPO with substitutions and Dual-Ring Transformer. **+ AC-19/AC-1M**: Trained with additional data from **AC-19** or **AC-1M** datasets (Section 3). **PPO-Sub-Canon**: Variant using canonical form preprocessing instead of cyclic relative positional encodings (Appendix E). For PPO methods, we report the mean over 5 seeds with min–max range in parentheses. GS is deterministic.

| METHOD | PRESENTATIONS SOLVED |
|---|---|
| *PPO Agents (mean over 5 seeds)* | |
| PPO-AC-RESNET | 457 |
| PPO-SUB-CANON | 562.6 (557–567) |
| PPO-SUB-CANON + **AC-19** | 575.5 (572–579) |
| PPO-SUB-DRT | 588.2 (585–591) |
| PPO-SUB-DRT + **AC-1M** | 605.4 (600–610) |
| PPO-SUB-DRT + **AC-19** | 607.2 (605–610) |
| *Greedy Search (deterministic)* | |
| GS-AC (1M) | 533 |
| GS-SUB (614) | 533 |
| GS-SUB (10K) NODES | 604 |
| GS-SUB (100K) NODES | 634 |
| GS-SUB (1M) NODES | 640 |
| GS-SUB (10M) NODES | 640 |

where each additional solve is valuable, this 19-presentation improvement demonstrates our data generation techniques successfully help the agent learn general strategies for solving hard presentations.

**3. PPO finds substantially better paths than GS, especially on hard presentations.** While greedy search solves more presentations (640 vs 610), PPO discovers significantly shorter solution paths, as shown in Figures 7 and 8 (all path lengths can be found in Appendix G). This advantage is especially pronounced on harder presentations (Figure 8), where PPO's learned value function identifies length-increasing detours that greedy search would not consider. The two approaches are complementary: GS's parallel exploration maximizes raw coverage, while PPO's learned value function finds the structurally efficient paths.

**4. Cyclic relative positional encodings outperform canonical form preprocessing.** Comparing PPO-Sub-DRT (588 presentations) with PPO-Sub-Canon (563 presentations) isolates the architectural choice for handling cyclic symmetry. The Dual-Ring Transformer's learned cyclic relative positional encodings provide a 25-presentation improvement over preprocessing inputs into canonical form with standard absolute positional encodings.

**Further Experiments** As a preliminary test of the comple-

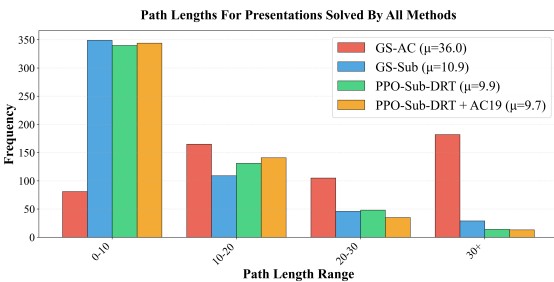

*Figure 7.* Distribution of solution path lengths for presentations solved by each method. Substitution-based approaches find dramatically shorter paths, with PPO leveraging the learned value function to identify even more efficient solutions. Path lengths are reported in substitution steps (supermoves); each step is a composition of AC1, AC2, and AC3 primitive moves.

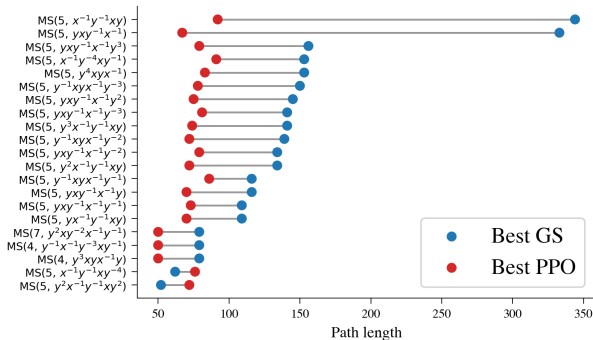

*Figure 8.* Comparison of solution path lengths for hard presentations solved by both greedy search and PPO (restricted to presentations where both methods required $\geq 50$ steps). PPO consistently discovers substantially shorter solution paths than greedy search, with GS finding a shorter path on only 2 instances. This demonstrates that PPO's learned value function successfully identifies length-increasing detours that lead to more efficient overall solutions—moves that greedy search's myopic length-reduction heuristic would reject.

mentarity above, we ran a beam search using PPO's policy as the heuristic. The hybrid solves the same 610 presentations as PPO alone, with the mean substitution path length reduced from 13.77 to 12.39 (a 10% reduction). On the 610 instances solved by both methods, the hybrid finds a strictly shorter path on 41% and matches PPO's length on 56%, suggesting that learned heuristics can improve solution quality on top of classical search. Full details of the beam search procedure are provided in Appendix I.

We also explored using automorphic numbers as an alternative reward signal for greedy search (see Appendix D for details). While this approach can recognize a broader set of terminal states and sometimes finds shorter paths, the computational overhead ($\sim 100\times$ per step) makes it currently infeasible for large-scale RL training and did not yield additional solves on the benchmark.

## 6. Conclusion

We have shown that progress on the Andrews-Curtis conjecture requires understanding the problem's inherent difficulty structure. The "Two-Hump" phenomenon—where presentations are either trivially easy or effectively impossible—explains why previous approaches struggled and motivates our three-pronged solution: substitution supermoves to take larger steps, the Dual-Ring Transformer to handle cyclic symmetry, and targeted data generation to populate the sparse difficulty valley. Our PPO agent solves 153 more presentations than the prior baseline (610 vs 457) and finds shorter paths than classical search algorithms. This progress shows the benefit of using mathematical structure to guide architectural design.

Critically, our progress is valuable regardless of whether the AC conjecture turns out to be true: if true, every new trivialization contributes toward a proof; if false, trivializing apparent counterexamples narrows the search for genuine ones.

The Two-Hump structure is not unique to AC: bimodal difficulty distributions and the need for progress-violating moves recur across mathematical search. Our techniques transfer naturally to these settings.

Looking forward, the generator-solver game and exhaustive enumeration establish systematic methods for data generation in the AC problem. These datasets provide immediate gains, and they become increasingly critical as model capability improves. Our code and datasets are available at https://github.com/Math-AI-Caltech/ACSolverX. We also provide a classification of minimal representatives for the Miller–Schupp benchmark in Appendix F.

Significant challenges remain: many benchmark presentations are unsolved, and famous potential counterexamples like AK(3) lie beyond current reach. However, this work makes concrete progress on a problem that has resisted human proof for over 60 years and provides tools and datasets for making future progress. It also shows that further progress will not come from marginal improvements in classical search, and instead will require learning-based methods that can exploit structure beyond what fixed heuristics can capture.

## Acknowledgments

The project was sponsored by the Defense Advanced Research Projects Agency under cooperative agreement HR0011262E017, by the NSF AIMing grant 2522494, by the DRW Foundation, and by a philanthropic gift from Les Kohn. Additionally, this work was supported with Cloud TPUs from Google's TPU Research Cloud (TRC), with GPUs from the NVIDIA Academic Grant Program, by cloud computing resources provided by Nebius through the Research Program of Nebius Academy, and by Advanced Micro Devices, Inc. under the AMD University Program's AI & HPC Cluster. M.T. is supported by the U.S. Department of Energy (Grant No. DE-SC0011632) and by the Walter Burke Institute for Theoretical Physics. The content of the information does not necessarily reflect the position or the policy of the Government, and no official endorsement should be inferred. Approved for public release; distribution is unlimited.

## Impact Statement

This work contributes to the application of machine learning to open problems in pure mathematics, specifically the Andrews-Curtis conjecture. We view mathematics as a valuable and challenging testbed for advancing machine learning methods, offering several unique advantages over traditional benchmarks. First, mathematical problems provide objectively verifiable success criteria. This eliminates concerns about evaluation metrics, human preference alignment, or subjective quality assessments that complicate other domains. Furthermore, problems like the Andrews-Curtis conjecture have resisted human mathematical effort for over 60 years despite their simple statement. The combinatorial search spaces are vast, and the solution landscape exhibits superexponential difficulty drop-offs that challenge standard RL techniques. Progress on the Andrews-Curtis conjecture demonstrates that RL can succeed in extremely sparse reward environments where solutions may require hundreds of precise sequential decisions, providing algorithmic insights transferable to robotics, theorem proving, program synthesis, and other domains where rewards are rare and delayed.

We also note that progress on pure mathematical conjectures poses essentially no risks of harmful applications, such as generating misinformation, enabling surveillance, or automating problematic decisions. The primary impact is advancing human mathematical knowledge and ML capabilities. We believe mathematics represents an ideal playground for developing more capable AI systems.

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

# A. The Dual-Ring Transformer: Architectural Details

This appendix provides complete architectural specifications for the Dual-Ring Transformer (PPO-Sub-DRT) introduced in Section 4. Note that we also evaluate a baseline variant (PPO-Sub-Canon) that uses canonical form preprocessing instead of cyclic relative positional encodings—see Appendix E for details on that alternative approach.

### A.1. Input Representation

The network input consists of two integer arrays representing the relators $r_1$ and $r_2$. We map generators to integers: $x \mapsto 1$, $x^{-1} \mapsto -1$, $y \mapsto 2$, $y^{-1} \mapsto -2$. The arrays are padded with zeros to a maximum length of $L = 24$.

### A.2. Embedding

Each integer token (shifted by +2 to handle the range $\{-2, -1, 0, 1, 2\}$) is passed through a learned embedding layer with vocabulary size 5 and embedding dimension $D = 32$. The embedding dimension equals num_heads $\times$ head_dim $= 4 \times 8 = 32$.

### A.3. Dual-Ring Transformer Blocks

The core of the architecture is a stack of 2 transformer blocks. Each block processes both relators in parallel with self-attention, cross-attention, and feed-forward layers, all using pre-normalization (LayerNorm before each sub-layer):

1. **Self-Attention with Cyclic Relative Positional Encoding:** Each relator attends to itself using 4-head multi-head attention with head dimension 8. For tokens at positions $i$ and $j$ in a relator of unpadded length $L$, the cyclic distance is $d(i, j) = (i - j) \bmod L$. The attention score combines content and relative position:

$$A_{ij}^h = (Q_i^h \cdot K_j^h) + (Q_i^h \cdot E_{d(i,j)}^h) \qquad (1)$$

where $E^h \in \mathbb{R}^{L \times D_h}$ is a learnable position embedding table per head, initialized with normal distribution (stddev=0.02). This gives the self-attention layer a cyclic-rotation-equivariant inductive bias.

2. **Cross-Attention:** The two relators attend to each other via symmetric cross-attention (4 heads, head dimension 8). This uses standard dot-product attention without relative position bias. This allows the network to identify matching substrings between $r_1$ and $r_2$, critical for determining valid substitution moves.

3. **Feed-Forward Networks:** Standard FFN with GELU activation and hidden dimension 32.

All sub-layers use pre-normalization (LayerNorm applied before the sub-layer) and residual connections.

## A.4. Policy Head

The policy head outputs a distribution over valid substitution moves. For each pair of positions $(k_1, k_2)$ in relators $r_1$ and $r_2$:

1. Concatenate the embeddings at positions $k_1$ and $k_2$: $[h_1[k_1]; h_2[k_2]]$ (dimension $2D = 64$)

2. Pass through a 2-layer MLP with GELU activation:
   - First layer: Linear $(64 \rightarrow 128)$ + GELU
   - Output layer: Linear $(128 \rightarrow 4)$ producing 4 logits corresponding to:
     - Which relator to replace: $i \in \{1, 2\}$
     - Whether to invert $r_2$: $j \in \{-1, 1\}$ (using $r_2^j$)

3. Apply a **semantic mask** that sets invalid actions to probability zero: only positions where $r_1[k_1] = \pm r_2[k_2]$ (enabling cancellation) receive non-zero probability.

This gives a distribution over $|r_1| \times |r_2| \times 4$ possible substitution moves, with invalid moves automatically masked out.

## A.5. Value Head

The value head estimates the expected return. It applies masked mean pooling over each relator (averaging only non-padding positions), concatenates the two pooled vectors (dimension $2D = 64$), and passes through a 3-layer MLP with GELU activation:

- First layer: Linear $(64 \rightarrow 256)$ + GELU
- Second layer: Linear $(256 \rightarrow 256)$ + GELU
- Output layer: Linear $(256 \rightarrow 1)$ producing the scalar value estimate

## A.6. Hyperparameters

- Embedding dimension $D$: 32 (= num_heads × head_dim = $4 \times 8$)
- Number of transformer blocks: 2
- Number of attention heads (per block): 4
- Head dimension: 8
- Feed-forward hidden dimension: 32
- Maximum sequence length $L$: 24

- Vocabulary size: 5 (tokens $\{-2, -1, 0, 1, 2\}$ with +2 shift)
- Activation function: GELU
- Policy head MLP: $(2D \rightarrow 128 \rightarrow 4)$
- Value head MLP: $(2D \rightarrow 256 \rightarrow 256 \rightarrow 1)$

Training details including PPO hyperparameters, reward function, and environment allocation are in Appendix B.

# B. PPO Training Details

This appendix provides complete details on the PPO training setup used for the results in Section 5.

## B.1. Reward Function

We use a length-based reward function:

$$r(s, a, s') = \begin{cases} 1000 & \text{if } s' \text{ trivial} \\ \max(-(|r_1'| + |r_2'|), -10) & \text{otherwise} \end{cases} \quad (2)$$

where $|r_i'|$ denotes the length of relator $i$ in state $s'$. This encourages length reduction close to the trivial presentation and incentivizes finding shorter paths. It is similar to the most effective reward function in (Shehper et al., 2025).

## B.2. Environment Allocation and Sampling Strategy

**Parallel Environment Setup.** We train with 2,380 parallel environments. The first 640 environments are reserved for the 640 presentations from the Miller-Schupp benchmark known to be AC-trivial (determined via greedy search). The remaining 1,740 environments sample from **AC-19** or **AC-1M** using adaptive sampling (described below).

We found that this allocation outperformed using all 1190 benchmark presentations: we never solved any presentation outside the initial 640 AC-trivial set, suggesting the remaining 550 presentations may be substantially harder or potentially not AC-trivial.

**Adaptive Sampling Function.** For the 1,740 sampling environments, we use an adaptive sampling strategy that balances exploration (trying rarely-attempted presentations) with exploitation (focusing on presentations near the boundary of solvability). Each presentation $p \in$ AC-19 or $p \in$ AC-1M is assigned a sampling weight:

$$w(p) = \frac{1 - (s(p))}{(1 + n(p))^{1/2}} \quad (3)$$

where $s(p) = \frac{n_{\text{solved}}(p)}{n(p) + 5}$ approximates the historical solve rate for presentation $p$, $n_{\text{solved}}(p)$ is the number of times $p$ has

been solved, and $n(p)$ is the total number of times $p$ has been attempted.

This function is a natural choice because:

- The numerator $(1 - s(p))$ prioritizes unsolved or rarely-solved presentations, which are more informative for learning than repeatedly solving easy presentations.

- The offset $+5$ ensures that presentations are solved many times before being passed on for harder presentations.

- The denominator $(1 + n(p))^{1/2}$ down-weights presentations that have been attempted many times, encouraging diversity and preventing over-sampling of presentations that happen to be at the current difficulty frontier.

This sampling strategy effectively implements a form of curriculum learning that automatically adapts to the agent's current capability level, focusing training on the "valley" between trivially easy and impossibly hard presentations.

### B.3. PPO Hyperparameters and Training Settings

We use Proximal Policy Optimization (Schulman et al., 2017) based on the PureJaxRL implementation (Lu et al., 2022). All training is done in JAX (Bradbury et al., 2018).

**Training hyperparameters:**

- Total training steps: 250M environment interactions

- Horizon length (rollout length per update): 96 steps

- Number of parallel environments: 2,380 (640 fixed + 1,740 sampling)

- Learning rate: $2.5 \times 10^{-3}$ (constant, no annealing)

- Number of update epochs per rollout: 3

- Number of minibatches per update: 8

- PPO clip parameter $\epsilon$: 0.2

- Discount factor $\gamma$: 0.999

- GAE lambda $\lambda$: 0.95

- Entropy coefficient: 0.01

- Value function coefficient: 0.5

- Maximum gradient norm (gradient clipping): 0.5

**Evaluation:**

- We train 5 independent agents with different random seeds

- We report mean, min, and max number of solved presentations over the 5 seeds

### B.4. Compute and Reproducibility

**Hardware.**

- Greedy search experiments: a single AMD EPYC 7502P (2.50 GHz, 32 cores, 512 GB RAM), parallelized into 60 subprocesses.

- PPO experiments: a single AMD Instinct MI210 GPU (64 GB HBM).

**Wall-clock time.**

- PPO training (250M environment interactions): approximately 11 hours.

- Greedy search with a 10M-node budget: up to 60 minutes per presentation.

- Generator-solver training: approximately 36 hours on the GS hardware above.

- **AC-19** enumeration: approximately 20,000 CPU-hours.

- **AC-1M** generation: approximately 2,900 CPU-hours.

**Training stability.** As a representative figure, PPO-Sub-DRT + AC-19 achieves $607.2 \pm 2.2$ solves over the 5 seeds (range 605–610), indicating stable training. Full per-method statistics appear in Table 1.

## C. Generator-Solver Game

We formulated a two-player asymmetric game to generate more "hard-but-solvable" data for curriculum training. The **Generator** applies automorphisms to an existing solved presentation, with the primary reward being the transformation of the presentation into one that the Solver can no longer solve, while also increasing its difficulty in terms of path length, and length growth. The **Solver**, implemented either as a reinforcement-learning agent or via greedy search, then attempts to trivialize the resulting presentation.

As explained in Section 3, if a presentation $P$ is trivialized by a Solver, then its automorphic image $\phi(P)$ is also AC-trivial. However, a trivialization path for $\phi(P)$ may be harder to find than that of $P$ due to different length statistics.

## C.1. Setup

**Generator:** We implement the Generator as a PPO Actor-Critic agent. The value head is a feed-forward network with two hidden layers of 128 units each. The actor is a dual-head feed-forward network, also with two hidden layers of 128 units each: one head outputs logits for the number of automorphisms $n$ to apply, while the other outputs the logits for sampling each automorphism $a_i$. Each automorphism is either inversion ($x_i \mapsto x_i^{-1}$) or concatenation ($x_i \mapsto x_i x_j$ for $i \neq j$).

To mitigate potential length explosion from concatenation moves, we impose a hard constraint on presentation length. At each step $i$, a candidate automorphism $a_i$ is sampled from the actor policy and applied to the current presentation only if the resulting presentation $P$ does not exceed a fixed length threshold $L_{\max}$; otherwise, $a_i$ is discarded and the presentation remains unchanged. An action $a$ is thus defined as the subsequence of valid elements of $(a_1, \cdots, a_i)$.

The reward for the generator $r_{\text{gen}}(s, a, s')$ is evaluated using a complexity measure score, defined as a linear combination of feedback signals returned by the Solver:

$$r_{\text{gen}}(s, a, s') = -w_s\, s + w_l\, l + w_i\, i + w_n\, n.$$

Here:

- $s$ (solved indicator): equals 1 if the Solver successfully solves the presentation and 0 otherwise. Note that a solved presentation incurs a penalty.

- $l$ (solution path length): the length of the Solver's solution path when the instance is solved, 0 otherwise.

- $i$ (intermediate length increase): the maximum length attained by any intermediate presentation along the Generator's trajectory. If no path was found, $i$ is given by the difference between the lengths of the initial presentation and the maximum length among all presentations considered. For greedy search, this puts a lower bound on length increase for any path that could be potentially discovered.

- $n$ (nodes visited): the number of nodes explored by the Solver's greedy search, when applicable.

**Solver:** The Solver can be a greedy search solver or a reinforcement agent equipped with Dual-Ring Transformer policy and value heads. In the experiments, we used exclusively the greedy search solver.

## C.2. Generating AC-1M

To generate **AC-1M**, we employ greedy search with a maximum of 10,000 nodes as the Solver. The trained Generator

generates 10 actions per episode and the length threshold $L_{\max} = 30$.

**Training:** The training data and validation data are taken from the 1007 presentations from **AC-19** requiring between 5,000 and 10,000 nodes to trivialize, with a train-test split of 80% to 20%. Evaluation is performed after each PPO optimization step. Reward features are saved after training and validation.

We set the weights at $w_s = 10{,}000$, $w_l = 1$, $w_i = 1{,}000$ and $w_n = 10$. This is meant to incentivize presentations that challenge the solver, push the left hump rightwards, and populate the "valley." We also explicitly reward presentations where the length increase is large, since solving any potential counterexample requires taking a path with this property. Finally, $w_s$ and $w_n$ are used to balance the fact that $l = 0$ for unsolved presentations.

The hyperparameters used for training were as follows:

- Learning rate: $2.5 \times 10^{-4}$ with linear decay

- Clip parameter $\epsilon$: 0.2

- Value function coefficient: 0.5

- Entropy coefficient: 0.01

- GAE $\lambda$: 0.95

- Discount factor $\gamma$: 0.99

- Batch size: 40000

- Number of epochs per update: 2

**Result:** To construct **AC-1M**, the trained network is applied to the entire **AC-19**, producing 10 automorphic images for each presentation.

# D. Automorphisms and the Automorphic Number

This appendix provides minimal mathematical background on automorphisms of free groups for readers with limited mathematical background.

## D.1. Automorphisms of Free Groups

An **automorphism** of the free group $F_2 = \langle x, y \rangle$ is an isomorphism $\phi : F_2 \to F_2$—a bijective homomorphism from the group to itself. Every automorphism is uniquely determined by where it sends the generators $x$ and $y$.

The automorphism group $\text{Aut}(F_2)$ can be generated by four

elementary automorphisms:

$$\alpha_1 : (x, y) \mapsto (x^{-1}, y) \quad \text{(invert first generator)} \tag{4}$$

$$\alpha_2 : (x, y) \mapsto (x, y^{-1}) \quad \text{(invert second generator)} \tag{5}$$

$$\alpha_3 : (x, y) \mapsto (y, x) \quad \text{(swap generators)} \tag{6}$$

$$\alpha_4 : (x, y) \mapsto (xy, y) \quad \text{(multiply first by second)} \tag{7}$$

Any automorphism can be expressed as a composition of these elementary operations.

### D.2. The Automorphic Number

Given a presentation $P = \langle x, y \mid r_1, r_2 \rangle$, its **automorphic number** $m(P)$ (or $m(r)$ for a single relator) is:

$$m(r) = \min_{\phi \in \text{Aut}(F_2)} |\phi(r)| \tag{8}$$

where $|r|$ denotes word length. Intuitively, $m(r)$ is the shortest length we can achieve for relator $r$ by applying any automorphism.

**Key property:** If $m(r) = 1$, then the relator $r$ can be reduced to a single generator (say $x$) under some automorphism. This means the presentation $\langle x, y \mid r, r_2 \rangle$ is AC-trivial, because after applying the automorphism we can use AC2 moves to eliminate all occurrences of $x$ from $r_2$.

Computing $m(r)$ exactly can be done using Whitehead's algorithm (Lyndon & Schupp, 2001), though this is computationally expensive.

### D.3. Automorphic Reward for Greedy Search

We experimented with using the automorphic number as an alternative heuristic for greedy search. Instead of prioritizing presentations with small total length $|r_1| + |r_2|$, we can prioritize presentations with small automorphic complexity $m(r_1) + m(r_2)$.

The automorphic reward function is defined as:

$$r(s) = \begin{cases} -m(r_1) - m(r_2), & \text{if } \min(m(r_1), m(r_2)) > 1, \\ H, & \text{otherwise,} \end{cases} \tag{9}$$

where $H$ is a large terminal reward. A state is terminal if either $m(r_1) = 1$ or $m(r_2) = 1$, since such presentations are guaranteed to be AC-trivial.

**Results:** This reward function has both advantages and disadvantages:

- **Advantage:** Expands the set of recognized terminal states. On the 1190-presentation Miller–Schupp benchmark, 98 presentations are already terminal under the automorphic criterion (compared to only the trivial presentation for length-based search). For presentations

solved by both methods, the automorphic reward often finds shorter solution paths.

- **Disadvantage:** Computing $m(r)$ using Whitehead's algorithm incurs approximately $100\times$ computational overhead per step. This makes the approach computationally expensive and currently infeasible for RL training. Despite the theoretical advantages, the automorphic greedy search did not solve any additional presentations on the Miller–Schupp benchmark compared to length-based greedy search, likely due to the restricted computational budget imposed by the expensive reward computation.

Figure 9 compares the number of search nodes required by length-based versus automorphic greedy search for presentations solved by both methods.

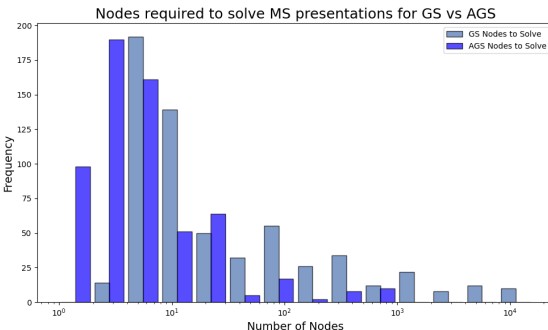

*Figure 9.* Comparison of greedy search with substitutions using length-based vs automorphic rewards on the Miller–Schupp dataset (in the figure, AGS denotes automorphic greedy search). Plotted are the number of nodes required for presentations solved by both approaches. The automorphic reward often requires fewer nodes but has much higher computational cost per node.

## E. Canonical Form

Relators in group presentations are topologically cyclic objects—rotation does not change the underlying structure. Similarly, inversion and relator ordering are symmetries that do not affect AC-triviality. There are two main approaches to handling these symmetries in neural architectures:

1. **Canonical form preprocessing (PPO-Sub-Canon):** Reduce all presentations to a unique canonical representative before feeding to a standard Transformer with absolute positional encodings.

2. **Learned symmetry-aware representations (PPO-Sub-DRT):** Use cyclic relative positional encodings that allow the network to flexibly attend to cyclically-related positions throughout processing.

As shown in Table 1, the learned approach (PPO-Sub-DRT) outperforms canonical form preprocessing (PPO-Sub-Canon) by approximately 25 presentations. This section describes the canonical form approach used in the baseline comparison.

### E.1. Canonical Form Definition

As discussed in Section 2.3, AC1 and AC3 moves preserve the set of available substitutions, allowing us to mod out by these symmetry operations. We achieve this by maintaining all presentations in a canonical form, which uniquely represents each equivalence class under cyclic rotation, inversion, and relator ordering.

**Definition E.1** (Canonical Form). A balanced presentation $\langle x, y \mid r_1, r_2 \rangle$ is in **canonical form** if:

1. Each relator $r_i$ is the lexicographically minimal element among all its cyclic rotations and the cyclic rotations of its inverse:

$$r_i = \min_{\text{lex}} \left\{ \text{rot}^k(r_i), \text{rot}^k(r_i^{-1}) \mid k \in \mathbb{Z} \right\}$$

2. The relators are ordered lexicographically: $r_1 \leq_{\text{lex}} r_2$

Every presentation can be reduced to a canonical form.

**Implication for substitutions.** Note that the relators $r_1 r_2^{-1}$, $r_2^{-1} r_1$, $r_1^{-1} r_2$ and $r_2 r_1^{-1}$ all have the same canonical form. This is why, for substitution moves, we only need to invert $r_2$ and only append $r_2^{\pm 1}$ onto $r_1$—the canonical form automatically handles the other variants. This observation significantly reduces the effective action space while maintaining completeness.

## F. Benchmark Dataset Classification

This appendix provides a complete classification of the 1,190-presentation Miller–Schupp benchmark into equivalence classes and minimal representatives.

### F.1. Full Classification Table

The first table (Table 2) classifies all 1,190 presentations from the Miller–Schupp benchmark, indexed by family parameters $(n, w)$. Entries marked *trivial* are the 640 presentations we successfully trivialized using greedy search with substitutions. The remaining entries give a counterexample class label, where the number is the minimal length of the class and the subscript is the class index; the minimal representatives of these classes are listed in Table 3.

*Table 2.* Classification results for benchmark dataset. Presentations indexed by $n$ and word $w$. For unsolved presentations, the counterexample class is given, where the number refers to the minimal length and the subscript to the index. The list of minimal representatives for each class is given in Table 3.

| $w$ | $n=1$ | $n=2$ | $n=3$ | $n=4$ | $n=5$ | $n=6$ | $n=7$ |
|---|---|---|---|---|---|---|---|
| $y^{-1}$ | trivial | trivial | trivial | trivial | trivial | trivial | trivial |
| $y$ | trivial | trivial | trivial | trivial | trivial | trivial | trivial |
| $y^2$ | trivial | trivial | trivial | trivial | trivial | trivial | trivial |
| $y^{-2}$ | trivial | trivial | trivial | trivial | trivial | trivial | trivial |
| $y^3$ | trivial | trivial | trivial | trivial | trivial | trivial | trivial |
| $y^{-3}$ | trivial | trivial | trivial | trivial | trivial | trivial | trivial |
| $y^{-4}$ | trivial | trivial | trivial | trivial | trivial | trivial | trivial |
| $x^{-1}y^{-1}xy$ | trivial | trivial | trivial | trivial | trivial | trivial | trivial |
| $x^{-1}y^{-1}xy^{-1}$ | trivial | trivial | $13_1$ | $15_1$ | $16_1$ | $20_9$ | $22_{14}$ |
| $yxyx^{-1}$ | trivial | trivial | $13_1$ | $15_1$ | $16_1$ | $20_8$ | $22_{13}$ |
| $y^4$ | trivial | trivial | trivial | trivial | trivial | trivial | trivial |
| $yxy^{-1}x^{-1}$ | trivial | trivial | trivial | trivial | trivial | trivial | trivial |
| $x^{-1}y^{-1}xy^2$ | trivial | trivial | trivial | $14_1$ | trivial | $19_{34}$ | $19_{38}$ |
| $yx^{-1}y^{-1}xy$ | trivial | trivial | trivial | trivial | trivial | trivial | trivial |
| $yxyx^{-1}y^{-1}$ | trivial | trivial | $13_1$ | $15_1$ | $16_1$ | $17_1$ | $21_1$ |
| $yxyx^{-1}y$ | trivial | trivial | $13_1$ | $15_1$ | $16_1$ | $20_8$ | $23_{21}$ |
| $x^{-1}y^{-1}xy^{-2}$ | trivial | trivial | trivial | $15_{11}$ | $14_1$ | $17_{36}$ | $21_{37}$ |
| $yxy^{-1}x^{-1}y$ | trivial | trivial | trivial | trivial | trivial | trivial | trivial |
| $y^{-1}xy^{-1}x^{-1}y^{-1}$ | trivial | trivial | $13_1$ | $15_1$ | $16_1$ | $17_1$ | $23_{24}$ |
| $yxy^{-2}x^{-1}$ | trivial | trivial | trivial | $14_1$ | trivial | $19_{33}$ | $19_{37}$ |
| $y^{-1}xyx^{-1}y^{-1}$ | trivial | trivial | trivial | trivial | trivial | trivial | trivial |
| $y^2xyx^{-1}$ | trivial | trivial | trivial | trivial | $17_{30}$ | $16_6$ | $23_{22}$ |
| $y^2xy^{-1}x^{-1}$ | trivial | trivial | $13_1$ | trivial | $17_{31}$ | $16_7$ | $21_{35}$ |
| $x^{-1}y^{-2}xy^{-1}$ | trivial | trivial | trivial | trivial | $17_{33}$ | $16_9$ | $23_{23}$ |
| $yxy^2x^{-1}$ | trivial | trivial | trivial | $15_{10}$ | $14_1$ | $17_{34}$ | $21_{34}$ |
| $yxy^{-1}x^{-1}y^{-1}$ | trivial | trivial | trivial | trivial | trivial | trivial | trivial |
| $y^5$ | trivial | trivial | trivial | trivial | trivial | trivial | trivial |
| $y^{-5}$ | trivial | trivial | trivial | trivial | trivial | trivial | trivial |
| $yx^{-1}y^{-1}xy^{-1}$ | trivial | trivial | $13_1$ | $15_1$ | $16_1$ | $17_{35}$ | $21_{36}$ |
| $x^{-1}y^{-2}xy$ | trivial | trivial | $13_1$ | trivial | $17_{32}$ | $16_8$ | $21_{38}$ |
| $yx^{-1}y^{-1}xy^2$ | trivial | trivial | trivial | $14_1$ | trivial | $19_{34}$ | $17_{42}$ |
| $yxy^{-2}x^{-1}y$ | trivial | trivial | trivial | $14_1$ | trivial | $19_{33}$ | $19_{37}$ |
| $y^{-1}xy^{-2}x^{-1}y^{-1}$ | trivial | trivial | trivial | $15_{11}$ | $14_1$ | $17_{36}$ | $21_{37}$ |
| $y^{-1}xy^{-1}x^{-1}y^{-2}$ | trivial | trivial | $13_1$ | $15_1$ | $16_1$ | $17_1$ | $18_7$ |
| $yx^{-1}y^{-1}xy^{-2}$ | trivial | trivial | trivial | $15_{11}$ | $14_1$ | $17_{36}$ | $15_{13}$ |
| $y^2xy^{-2}x^{-1}$ | trivial | trivial | trivial | trivial | trivial | trivial | trivial |
| $y^{-1}xy^2x^{-1}y^{-1}$ | trivial | trivial | trivial | $14_1$ | trivial | $19_{34}$ | $20_{12}$ |
| $y^2x^{-1}y^{-1}xy^{-1}$ | trivial | trivial | $13_1$ | $15_1$ | $16_1$ | $17_{35}$ | $18_6$ |
| $y^2xy^{-1}x^{-1}y$ | trivial | trivial | $13_1$ | trivial | $17_{31}$ | $16_7$ | $21_{35}$ |
| $y^2xy^{-1}x^{-1}y^{-1}$ | trivial | trivial | $13_1$ | trivial | $17_{31}$ | $16_7$ | $21_{35}$ |
| $y^6$ | trivial | trivial | trivial | trivial | trivial | trivial | trivial |
| $yx^{-1}y^{-2}xy^{-1}$ | trivial | trivial | trivial | trivial | $17_{33}$ | $16_9$ | $23_{23}$ |
| $y^2x^{-1}y^{-1}xy$ | trivial | trivial | trivial | trivial | trivial | trivial | trivial |
| $yx^{-1}y^{-2}xy$ | trivial | trivial | $13_1$ | trivial | $17_{32}$ | $16_8$ | $21_{38}$ |
| $yxy^{-1}x^{-1}y^2$ | trivial | trivial | trivial | trivial | trivial | trivial | trivial |
| $y^{-1}xyx^{-1}y^{-2}$ | trivial | trivial | trivial | trivial | trivial | trivial | trivial |
| $yxy^{-2}x^{-1}y^{-1}$ | trivial | trivial | trivial | $14_1$ | trivial | $19_{33}$ | $17_{41}$ |
| $y^2xyx^{-1}y^{-1}$ | trivial | trivial | trivial | trivial | $17_{30}$ | $16_6$ | $23_{22}$ |
| $x^{-1}y^{-2}xy^{-2}$ | trivial | trivial | trivial | trivial | $16_5$ | $19_{36}$ | $20_{11}$ |

Table 2 – continued from previous page

| $w$ | $n = 1$ | $n = 2$ | $n = 3$ | $n = 4$ | $n = 5$ | $n = 6$ | $n = 7$ |
|---|---|---|---|---|---|---|---|
| $yxyx^{-1}y^{-2}$ | trivial | trivial | $13_1$ | $15_1$ | $16_1$ | $17_1$ | $18_7$ |
| $x^{-2}y^{-1}x^2y^{-1}$ | trivial | $14_1$ | $16_2$ | $18_3$ | $20_3$ | $22_4$ | $24_4$ |
| $x^{-2}y^{-1}x^2y$ | trivial | $14_2$ | $16_3$ | $18_2$ | $20_2$ | $22_3$ | $24_3$ |
| $yx^2yx^{-2}$ | trivial | $14_1$ | $16_2$ | $18_1$ | $20_1$ | $22_1$ | $24_1$ |
| $yxyx^{-1}y^2$ | trivial | trivial | $13_1$ | $15_1$ | $16_1$ | $17_{35}$ | $18_6$ |
| $y^3xyx^{-1}$ | trivial | trivial | trivial | trivial | $17_{32}$ | trivial | $19_{40}$ |
| $x^{-1}y^{-3}xy^{-1}$ | trivial | trivial | trivial | trivial | $17_{31}$ | trivial | $19_{42}$ |
| $y^3xy^{-1}x^{-1}$ | trivial | trivial | trivial | $15_1$ | $17_{33}$ | trivial | $18_9$ |
| $x^{-1}y^{-3}xy$ | trivial | trivial | trivial | $15_1$ | $17_{30}$ | trivial | $18_{11}$ |
| $yxy^{-3}x^{-1}$ | trivial | trivial | $13_1$ | $15_{10}$ | trivial | $17_{38}$ | $18_8$ |
| $yx^2y^{-1}x^{-2}$ | trivial | $14_2$ | $16_3$ | $18_2$ | $20_2$ | $22_2$ | $24_2$ |
| $x^{-1}y^{-2}xy^2$ | trivial | trivial | trivial | trivial | trivial | trivial | trivial |
| $yxy^{-1}x^{-1}y^{-2}$ | trivial | trivial | trivial | trivial | trivial | trivial | trivial |
| $yxy^2x^{-1}y^{-1}$ | trivial | trivial | trivial | $15_{10}$ | $14_1$ | $17_{34}$ | $15_{12}$ |
| $y^{-6}$ | trivial | trivial | trivial | trivial | trivial | trivial | trivial |
| $yxy^2x^{-1}y$ | trivial | trivial | trivial | $15_{10}$ | $14_1$ | $17_{34}$ | $21_{34}$ |
| $x^{-1}y^{-1}xy^3$ | trivial | trivial | $13_1$ | $15_{11}$ | trivial | $17_{39}$ | $18_{10}$ |
| $y^2xy^2x^{-1}$ | trivial | trivial | trivial | trivial | $16_4$ | $19_{35}$ | $20_{10}$ |
| $y^{-1}x^{-1}y^{-2}xy$ | trivial | trivial | $13_1$ | trivial | $17_{32}$ | $16_8$ | $21_{38}$ |
| $yxy^3x^{-1}$ | trivial | trivial | trivial | $14_1$ | trivial | $17_{37}$ | $19_{39}$ |
| $x^{-1}y^{-1}xy^{-3}$ | trivial | trivial | trivial | $14_1$ | trivial | $17_{40}$ | $19_{41}$ |
| $y^2xyx^{-1}y$ | trivial | trivial | trivial | trivial | $17_{30}$ | $16_6$ | $23_{22}$ |
| $y^{-1}x^{-1}y^{-2}xy^{-1}$ | trivial | trivial | trivial | trivial | $17_{33}$ | $16_9$ | $23_{23}$ |
| $yxyx^{-1}y^{-3}$ | trivial | trivial | $13_1$ | $15_1$ | $16_1$ | $17_1$ | $18_7$ |
| $yxy^4x^{-1}$ | trivial | trivial | trivial | trivial | trivial | $17_{38}$ | trivial |
| $yxy^3x^{-1}y$ | trivial | trivial | trivial | $14_1$ | trivial | $17_{37}$ | $19_{39}$ |
| $yxyxy^{-1}x^{-2}$ | trivial | $15_7$ | $17_{13}$ | $19_{12}$ | $21_{11}$ | $23_6$ | $25_{12}$ |
| $yxyxyx^{-2}$ | trivial | $15_6$ | $17_{12}$ | $19_{11}$ | $21_{10}$ | $23_5$ | $25_{11}$ |
| $yxy^2x^{-1}y^2$ | trivial | trivial | trivial | $15_{10}$ | $14_1$ | $17_{34}$ | $21_{34}$ |
| $yxy^2x^{-1}y^{-2}$ | trivial | trivial | trivial | $15_{10}$ | $14_1$ | $17_{34}$ | $15_{14}$ |
| $yxyx^{-1}y^3$ | trivial | trivial | $13_1$ | $15_1$ | $16_1$ | $17_{35}$ | $18_6$ |
| $yxy^3x^{-1}y^{-1}$ | trivial | trivial | trivial | $14_1$ | trivial | $17_{37}$ | $19_{39}$ |
| $x^{-2}y^{-2}x^2y^{-1}$ | trivial | trivial | $17_9$ | $19_{28}$ | $21_{26}$ | $23_{19}$ | $25_{28}$ |
| $y^2x^{-1}y^{-2}xy$ | trivial | trivial | $13_1$ | trivial | $17_{32}$ | $16_8$ | $21_{38}$ |
| $y^4xyx^{-1}$ | trivial | trivial | $13_1$ | trivial | trivial | $16_8$ | $19_{46}$ |
| $y^4xy^{-1}x^{-1}$ | trivial | trivial | trivial | trivial | $16_1$ | $16_9$ | $19_{47}$ |
| $y^3xy^2x^{-1}$ | trivial | trivial | trivial | $14_1$ | trivial | trivial | $21_{40}$ |
| $y^3xyx^{-1}y$ | trivial | trivial | trivial | trivial | $17_{32}$ | trivial | $19_{40}$ |
| $y^3xyx^{-1}y^{-1}$ | trivial | trivial | trivial | trivial | $17_{32}$ | trivial | $19_{40}$ |
| $y^3xy^{-2}x^{-1}$ | trivial | trivial | trivial | $15_{11}$ | $16_5$ | trivial | $19_{45}$ |
| $y^3xy^{-1}x^{-1}y$ | trivial | trivial | trivial | $15_1$ | $17_{33}$ | trivial | $18_9$ |
| $y^3xy^{-1}x^{-1}y^{-1}$ | trivial | trivial | trivial | $15_1$ | $17_{33}$ | trivial | $18_9$ |
| $y^3x^{-1}y^{-1}xy$ | trivial | trivial | trivial | trivial | trivial | trivial | trivial |
| $y^3x^{-1}y^{-1}xy^{-1}$ | trivial | trivial | $13_1$ | $15_1$ | $16_1$ | $17_{35}$ | $18_6$ |
| $y^2xy^3x^{-1}$ | trivial | trivial | $13_1$ | trivial | trivial | $18_4$ | $21_{39}$ |
| $y^2xy^2x^{-1}y$ | trivial | trivial | trivial | trivial | $16_4$ | $19_{35}$ | $20_{10}$ |
| $y^2xy^2x^{-1}y^{-1}$ | trivial | trivial | trivial | trivial | $16_4$ | $19_{35}$ | $20_{10}$ |
| $y^2xyx^{-1}y^2$ | trivial | trivial | trivial | trivial | $17_{30}$ | $16_{10}$ | $23_{22}$ |
| $y^2xyx^{-1}y^{-2}$ | trivial | trivial | trivial | trivial | $17_{30}$ | $16_6$ | $19_{43}$ |
| $y^2x^2yx^{-2}$ | trivial | trivial | $17_{16}$ | $19_{15}$ | $21_{14}$ | $23_9$ | $25_{15}$ |
| $y^2x^2y^{-1}x^{-2}$ | trivial | trivial | $17_{17}$ | $19_{16}$ | $21_{15}$ | $23_{10}$ | $25_{16}$ |

Continued on next page

Table 2 – continued from previous page

| $w$ | $n=1$ | $n=2$ | $n=3$ | $n=4$ | $n=5$ | $n=6$ | $n=7$ |
|---|---|---|---|---|---|---|---|
| $y^2xy^{-3}x^{-1}$ | trivial | trivial | trivial | trivial | trivial | trivial | $19_{44}$ |
| $y^2xy^{-2}x^{-1}y^{-1}$ | trivial | trivial | trivial | trivial | trivial | trivial | trivial |
| $y^2xy^{-1}x^{-1}y^2$ | trivial | trivial | $13_1$ | trivial | $17_{31}$ | $16_7$ | $21_{35}$ |
| $y^2xy^{-1}x^{-1}y^{-2}$ | trivial | trivial | $13_1$ | trivial | $17_{31}$ | $16_7$ | $21_{35}$ |
| $y^2x^{-1}y^{-1}xy^2$ | trivial | trivial | trivial | $14_1$ | trivial | $19_{34}$ | $17_{42}$ |
| $y^2x^{-1}y^{-1}xy^{-2}$ | trivial | trivial | trivial | $15_{11}$ | $14_1$ | $17_{36}$ | $15_{15}$ |
| $y^2x^{-1}y^{-2}xy^{-1}$ | trivial | trivial | trivial | trivial | $17_{33}$ | $16_9$ | $19_{48}$ |
| $y^2xy^{-2}x^{-1}y$ | trivial | trivial | trivial | trivial | trivial | trivial | trivial |
| $yxy^{-1}x^{-1}y^3$ | trivial | trivial | trivial | trivial | trivial | trivial | trivial |
| $yx^2yx^{-1}yx^{-1}$ | trivial | $15_3$ | $17_3$ | $19_2$ | $20_4$ | $21_{30}$ | $25_2$ |
| $y^{-1}xy^{-1}x^{-1}y^{-3}$ | trivial | trivial | $13_1$ | $15_1$ | $16_1$ | $17_1$ | $18_7$ |
| $y^{-2}xyx^{-1}y^{-2}$ | trivial | trivial | $13_1$ | trivial | $17_{32}$ | $16_8$ | $21_{38}$ |
| $y^{-2}xy^{-1}x^{-1}y^{-2}$ | trivial | trivial | trivial | trivial | $17_{33}$ | $16_9$ | $23_{23}$ |
| $y^{-7}$ | trivial | trivial | trivial | trivial | trivial | trivial | trivial |
| $y^{-1}x^{-1}y^{-2}xy^2$ | trivial | trivial | trivial | trivial | trivial | trivial | trivial |
| $y^{-1}x^{-1}y^{-2}xy^{-2}$ | trivial | trivial | trivial | trivial | $16_5$ | $19_{36}$ | $20_{11}$ |
| $y^{-1}x^{-1}y^{-3}xy$ | trivial | trivial | trivial | $15_1$ | $17_{30}$ | trivial | $18_{11}$ |
| $y^{-1}x^{-1}y^{-3}xy^{-1}$ | trivial | trivial | trivial | trivial | $17_{31}$ | trivial | $19_{42}$ |
| $x^{-1}y^{-1}xy^4$ | trivial | trivial | trivial | $15_1$ | $14_1$ | $17_{40}$ | trivial |
| $x^{-1}y^{-1}x^2yx^{-1}y^{-1}$ | trivial | $15_8$ | $17_{26}$ | $19_{29}$ | $21_{27}$ | $23_{20}$ | $25_{29}$ |
| $x^{-1}y^{-1}x^2y^{-1}x^{-1}y^{-1}$ | trivial | $15_9$ | $17_{27}$ | $19_{30}$ | $20_7$ | $21_{33}$ | $25_{30}$ |
| $x^{-1}y^{-1}xy^{-4}$ | trivial | trivial | trivial | trivial | trivial | $17_{39}$ | trivial |
| $x^{-1}y^{-2}xy^3$ | trivial | trivial | trivial | trivial | trivial | trivial | $19_{49}$ |
| $x^{-1}y^{-2}xy^{-3}$ | trivial | trivial | $13_1$ | trivial | trivial | $18_5$ | $21_{41}$ |
| $x^{-1}y^{-3}xy^2$ | trivial | trivial | trivial | $15_{10}$ | $16_4$ | trivial | $19_{50}$ |
| $x^{-1}y^{-3}xy^{-2}$ | trivial | trivial | trivial | $14_1$ | trivial | trivial | $21_{42}$ |
| $x^{-1}y^{-4}xy$ | trivial | trivial | trivial | trivial | $16_1$ | $16_6$ | $19_{51}$ |
| $x^{-1}y^{-4}xy^{-1}$ | trivial | trivial | $13_1$ | trivial | trivial | $16_7$ | $19_{52}$ |
| $x^{-1}y^{-1}x^{-1}y^{-1}x^2y$ | trivial | $14_1$ | $17_{28}$ | $19_{31}$ | $21_{28}$ | $22_{11}$ | $25_{31}$ |
| $x^{-1}y^{-1}x^{-1}y^{-1}x^2y^{-1}$ | trivial | $14_2$ | $17_{29}$ | $19_{32}$ | $21_{29}$ | $22_{12}$ | $25_{32}$ |
| $x^{-2}y^{-1}xyxy$ | trivial | $15_3$ | $17_{22}$ | $19_{23}$ | $21_{21}$ | $23_{14}$ | $25_{23}$ |
| $x^{-2}y^{-1}xyxy^{-1}$ | trivial | $15_5$ | $17_{23}$ | $19_{24}$ | $21_{22}$ | $23_{15}$ | $25_{24}$ |
| $x^{-2}y^{-1}x^2y^2$ | trivial | $15_9$ | $17_{20}$ | $19_{21}$ | $21_{19}$ | $23_{12}$ | $25_{21}$ |
| $x^{-2}y^{-1}x^2y^{-2}$ | trivial | $15_6$ | $17_{21}$ | $19_{22}$ | $21_{20}$ | $23_{13}$ | $25_{22}$ |
| $x^{-2}y^{-1}xy^{-1}xy$ | trivial | $15_4$ | $17_{24}$ | $19_{25}$ | $21_{23}$ | $23_{16}$ | $25_{25}$ |
| $x^{-2}y^{-1}xy^{-1}xy^{-1}$ | trivial | $15_2$ | $17_{25}$ | $19_{26}$ | $21_{24}$ | $23_{17}$ | $25_{26}$ |
| $x^{-2}y^{-2}x^2y$ | trivial | trivial | $17_5$ | $19_{27}$ | $21_{25}$ | $23_{18}$ | $25_{27}$ |
| $y^{-1}xy^{-2}x^{-1}y^{-2}$ | trivial | trivial | trivial | $15_{11}$ | $14_1$ | $17_{36}$ | $21_{37}$ |
| $yx^2y^2x^{-2}$ | trivial | $15_2$ | $17_2$ | $19_1$ | $21_2$ | $23_1$ | $25_1$ |
| $y^{-1}xy^{-3}x^{-1}y^{-1}$ | trivial | trivial | trivial | $14_1$ | trivial | $17_{40}$ | $19_{41}$ |
| $y^{-1}xy^2x^{-1}y^{-2}$ | trivial | trivial | trivial | $14_1$ | trivial | $19_{34}$ | $20_{12}$ |
| $yx^2yx^{-1}y^{-1}x^{-1}$ | trivial | $15_4$ | $17_6$ | $19_5$ | $21_5$ | $23_2$ | $25_5$ |
| $yx^2yx^{-2}y$ | trivial | $14_2$ | $17_4$ | $19_3$ | $21_3$ | $22_5$ | $25_3$ |
| $yx^2yx^{-2}y^{-1}$ | trivial | $14_2$ | $17_5$ | $19_4$ | $21_4$ | $22_6$ | $25_4$ |
| $yx^2y^{-2}x^{-2}$ | trivial | $15_3$ | $17_{11}$ | $19_{10}$ | $21_9$ | $23_4$ | $25_{10}$ |
| $yx^2y^{-1}x^{-1}yx^{-1}$ | trivial | $15_5$ | $17_7$ | $19_6$ | $21_6$ | $23_3$ | $25_6$ |
| $yx^2y^{-1}x^{-1}y^{-1}x^{-1}$ | trivial | $15_2$ | $17_{10}$ | $19_9$ | $20_5$ | $21_{31}$ | $25_9$ |
| $yx^2y^{-1}x^{-2}y$ | trivial | $14_1$ | $17_8$ | $19_7$ | $21_7$ | $22_7$ | $25_7$ |
| $yx^2y^{-1}x^{-2}y^{-1}$ | trivial | $14_1$ | $17_9$ | $19_8$ | $21_8$ | $22_8$ | $25_8$ |
| $yxy^{-1}xyx^{-2}$ | trivial | $15_8$ | $17_{14}$ | $19_{13}$ | $21_{12}$ | $23_7$ | $25_{13}$ |
| $yxy^{-1}xy^{-1}x^{-2}$ | trivial | $15_9$ | $17_{15}$ | $19_{14}$ | $21_{13}$ | $23_8$ | $25_{14}$ |

Table 2 – continued from previous page

| $w$ | $n=1$ | $n=2$ | $n=3$ | $n=4$ | $n=5$ | $n=6$ | $n=7$ |
|---|---|---|---|---|---|---|---|
| $yxy^{-4}x^{-1}$ | trivial | trivial | trivial | $15_1$ | $14_1$ | $17_{37}$ | trivial |
| $yxy^{-3}x^{-1}y$ | trivial | trivial | $13_1$ | $15_{10}$ | trivial | $17_{38}$ | $18_8$ |
| $yxy^{-3}x^{-1}y^{-1}$ | trivial | trivial | $13_1$ | $15_{10}$ | trivial | $17_{38}$ | $18_8$ |
| $yxy^{-2}x^{-1}y^2$ | trivial | trivial | trivial | $14_1$ | trivial | $19_{33}$ | $19_{37}$ |
| $yxy^{-2}x^{-1}y^{-2}$ | trivial | trivial | trivial | $14_1$ | trivial | $19_{33}$ | $17_{41}$ |
| $yxy^{-1}x^{-1}y^{-3}$ | trivial | trivial | trivial | trivial | trivial | trivial | trivial |
| $yx^{-1}y^{-1}xy^3$ | trivial | trivial | $13_1$ | $15_{11}$ | trivial | $17_{39}$ | $18_{10}$ |
| $yx^{-1}y^{-1}x^2yx^{-1}$ | trivial | $14_1$ | $17_{16}$ | $19_{19}$ | $21_{17}$ | $22_9$ | $25_{19}$ |
| $yx^{-1}y^{-1}x^2y^{-1}x^{-1}$ | trivial | $14_2$ | $17_{17}$ | $19_{20}$ | $21_{18}$ | $22_{10}$ | $25_{20}$ |
| $yx^{-1}y^{-1}xy^{-3}$ | trivial | trivial | trivial | $14_1$ | trivial | $17_{40}$ | $19_{41}$ |
| $yx^{-1}y^{-2}xy^2$ | trivial | trivial | trivial | trivial | trivial | trivial | trivial |
| $yx^{-1}y^{-2}xy^{-2}$ | trivial | trivial | trivial | trivial | $16_5$ | $19_{36}$ | $20_{11}$ |
| $yx^{-1}y^{-3}xy$ | trivial | trivial | trivial | $15_1$ | $17_{30}$ | trivial | $18_{11}$ |
| $yx^{-1}y^{-3}xy^{-1}$ | trivial | trivial | trivial | trivial | $17_{31}$ | trivial | $19_{42}$ |
| $yx^{-2}y^{-1}x^2y$ | trivial | $15_6$ | $17_{18}$ | $19_{17}$ | $20_6$ | $21_{32}$ | $25_{17}$ |
| $yx^{-2}y^{-1}x^2y^{-1}$ | trivial | $15_7$ | $17_{19}$ | $19_{18}$ | $21_{16}$ | $23_{11}$ | $25_{18}$ |
| $y^{-1}xy^3x^{-1}y^{-1}$ | trivial | trivial | $13_1$ | $15_{11}$ | trivial | $17_{39}$ | $18_{10}$ |
| $y^{-1}xyx^{-1}y^{-3}$ | trivial | trivial | trivial | trivial | trivial | trivial | trivial |
| $y^7$ | trivial | trivial | trivial | trivial | trivial | trivial | trivial |

## F.2. Unsolved Presentations: Equivalence Classes

The second table classifies the remaining 550 unsolved presentations into 261 distinct AC-equivalence classes. For each equivalence class, we provide the minimal representative for the class and the number of times it appears in the MS dataset (the above table).

This classification reveals that many unsolved presentations are AC-equivalent to each other, reducing the 550 unsolved instances to just 261 genuinely distinct hard cases. This provides the community with a concrete checklist of minimal potential counterexamples for future algorithmic development.

*Table 3.* Minimal counterexamples from benchmark dataset with number of occurrences.

| $r_1$ | $r_2$ | name | occurrences |
|---|---|---|---|
| $y^{-1}x^{-1}yx^{-1}y^{-1}x$ | $y^{-3}x^{-4}$ | $13_1$ | 34 |
| $y^{-3}x^{-1}y^2x$ | $y^{-1}x^{-2}y^{-1}x^3$ | $14_1$ | 36 |
| $y^{-3}xy^2x^{-1}$ | $y^{-1}x^{-2}yx^3$ | $14_2$ | 6 |
| $y^{-2}x^{-1}yx^{-1}yx$ | $y^{-2}x^4yx^{-1}$ | $15_1$ | 22 |
| $y^{-1}x^{-3}yx^2$ | $y^{-3}x^{-2}y^2x^{-1}$ | $15_2$ | 3 |
| $y^{-3}x^{-1}y^2x$ | $y^{-1}x^{-2}y^{-1}x^2y^{-1}x$ | $15_3$ | 3 |
| $y^{-3}x^{-1}y^2x$ | $y^{-1}x^{-1}yx^{-1}yx^3$ | $15_4$ | 2 |
| $y^{-3}x^{-1}y^2x$ | $y^{-1}x^{-1}yx^{-2}yx^2$ | $15_5$ | 2 |
| $y^{-1}x^{-2}yx^3$ | $y^{-3}x^{-1}yx^{-1}yx^{-1}$ | $15_6$ | 3 |
| $y^{-3}x^{-1}y^2x$ | $y^{-1}x^{-1}y^{-1}x^3yx^{-1}$ | $15_7$ | 2 |
| $y^{-3}x^{-1}y^2x$ | $y^{-1}x^{-2}y^{-1}x^2yx^{-1}$ | $15_8$ | 2 |
| $y^{-1}x^{-3}yx^2$ | $y^{-3}x^2y^2x^{-1}$ | $15_9$ | 3 |
| $y^{-1}x^{-1}yx^2yx^{-1}$ | $y^{-2}x^{-1}y^2xy^{-1}x$ | $15_{10}$ | 9 |
| $y^{-3}x^{-1}yx^2$ | $y^{-3}x^2y^2x^{-1}$ | $15_{11}$ | 9 |
| $y^{-2}x^{-1}y^2x^2$ | $y^{-1}x^{-3}yxy^{-1}x$ | $15_{12}$ | 1 |
| $y^{-2}x^{-2}y^2x$ | $y^{-1}x^{-1}y^{-1}x^{-1}yx^3$ | $15_{13}$ | 1 |
| $y^{-2}x^{-2}yx^2$ | $y^{-3}xyx^{-1}yx^{-1}$ | $15_{14}$ | 1 |
| $y^{-2}x^{-2}yx^2$ | $y^{-3}xyxyx^{-1}$ | $15_{15}$ | 1 |
| $y^{-2}x^{-1}yx^{-1}yx$ | $y^{-2}x^5yx^{-1}$ | $16_1$ | 16 |

Table 3 – continued from previous page

| $r_1$ | $r_2$ | name | occurrences |
|---|---|---|---|
| $y^{-3}x^{-1}y^2x^{-1}$ | $y^{-1}x^{-4}yx^3$ | $16_2$ | 2 |
| $y^{-3}x^{-1}y^2x$ | $y^{-1}x^{-4}yx^3$ | $16_3$ | 2 |
| $y^{-2}x^{-2}yx^{-2}$ | $y^{-3}xy^2x^{-1}yx^{-1}$ | $16_4$ | 4 |
| $y^{-2}x^2yx^2$ | $y^{-3}xy^2xyx^{-1}$ | $16_5$ | 4 |
| $y^{-2}xyxy^{-1}x^{-1}$ | $y^{-1}x^{-1}yx^{-2}yx^3$ | $16_6$ | 5 |
| $y^{-2}xy^{-1}xyx^{-1}$ | $y^{-1}x^{-3}yx^2y^{-1}x$ | $16_7$ | 6 |
| $y^{-2}xyx^{-1}y^{-1}x^{-1}$ | $y^{-1}x^{-1}y^{-1}x^{-2}yx^3$ | $16_8$ | 6 |
| $y^{-2}x^{-1}yx^{-1}y^{-1}x^{-1}$ | $y^{-2}x^{-4}yx^2$ | $16_9$ | 6 |
| $y^{-2}x^2yx^{-2}$ | $y^{-4}x^{-1}y^{-1}xy^{-1}x$ | $16_{10}$ | 1 |
| $y^{-1}x^{-1}y^{-1}x^{-1}yx^2$ | $y^{-5}xyx^{-1}yx^{-1}$ | $17_1$ | 6 |
| $y^{-2}x^{-2}y^{-1}x^3$ | $y^{-3}x^{-1}y^{-1}xy^{-1}x^2$ | $17_2$ | 1 |
| $y^{-1}x^{-2}y^{-1}x^2y^{-1}x$ | $y^{-4}x^{-1}y^3x$ | $17_3$ | 1 |
| $y^{-1}x^{-2}y^{-1}xy^{-1}x^2$ | $y^{-4}x^{-1}y^3x$ | $17_4$ | 1 |
| $y^{-2}x^2yx^{-3}$ | $y^{-4}x^{-1}y^3x$ | $17_5$ | 2 |
| $y^{-1}x^{-2}y^{-1}x^2yx$ | $y^{-4}x^{-1}y^3x$ | $17_6$ | 1 |
| $y^{-1}x^{-1}yx^{-2}yx^2$ | $y^{-4}x^{-1}y^3x$ | $17_7$ | 1 |
| $y^{-1}x^{-2}yx^{-1}yx^2$ | $y^{-4}x^{-1}y^3x$ | $17_8$ | 1 |
| $y^{-1}x^{-1}yx^2y^{-1}x^{-2}$ | $y^{-4}x^{-1}y^3x$ | $17_9$ | 2 |
| $y^{-1}x^{-1}y^{-1}x^{-2}yx^2$ | $y^{-4}x^{-1}y^3x$ | $17_{10}$ | 1 |
| $y^{-2}x^{-3}yx^2$ | $y^{-1}x^{-1}yx^2yx^{-1}y^{-1}x$ | $17_{11}$ | 1 |
| $y^{-1}x^{-1}y^{-1}x^{-1}y^{-1}x^3$ | $y^{-4}x^{-1}y^3x$ | $17_{12}$ | 1 |
| $y^{-1}x^{-1}y^{-1}x^3yx^{-1}$ | $y^{-4}x^{-1}y^3x$ | $17_{13}$ | 1 |
| $y^{-1}x^{-1}yx^{-1}y^{-1}x^3$ | $y^{-4}x^{-1}y^3x$ | $17_{14}$ | 1 |
| $y^{-1}x^{-3}yxy^{-1}x$ | $y^{-4}x^{-1}y^3x$ | $17_{15}$ | 1 |
| $y^{-1}x^{-2}yxy^{-1}x^2$ | $y^{-4}x^{-1}y^3x$ | $17_{16}$ | 2 |
| $y^{-1}x^{-2}yx^{-1}y^{-1}x^2$ | $y^{-4}x^{-1}y^3x$ | $17_{17}$ | 2 |
| $y^{-1}x^{-2}yx^2y^{-1}x$ | $y^{-4}x^{-1}y^3x$ | $17_{18}$ | 1 |
| $y^{-1}x^{-1}yx^{-2}y^{-1}x^2$ | $y^{-4}x^{-1}y^3x$ | $17_{19}$ | 1 |
| $y^{-2}x^{-2}yx^3$ | $y^{-4}x^{-1}y^3x$ | $17_{20}$ | 1 |
| $y^{-2}x^{-3}y^{-1}x^2$ | $y^{-3}x^{-2}y^{-1}x^{-1}y^{-1}x$ | $17_{21}$ | 1 |
| $y^{-1}x^{-1}y^{-1}x^{-1}yx^3$ | $y^{-4}x^{-1}y^3x$ | $17_{22}$ | 1 |
| $y^{-1}x^{-1}yx^3yx^{-1}$ | $y^{-4}x^{-1}y^3x$ | $17_{23}$ | 1 |
| $y^{-1}x^{-1}yx^{-1}yx^3$ | $y^{-4}x^{-1}y^3x$ | $17_{24}$ | 1 |
| $y^{-1}x^{-3}y^{-1}xy^{-1}x$ | $y^{-4}x^{-1}y^3x$ | $17_{25}$ | 1 |
| $y^{-1}x^{-2}y^{-1}x^2yx^{-1}$ | $y^{-4}x^{-1}y^3x$ | $17_{26}$ | 1 |
| $y^{-1}x^{-1}y^{-1}x^{-2}y^{-1}x^2$ | $y^{-4}x^{-1}y^3x$ | $17_{27}$ | 1 |
| $y^{-1}x^{-1}y^{-1}x^2yx^{-2}$ | $y^{-4}x^{-1}y^3x$ | $17_{28}$ | 1 |
| $y^{-1}x^{-1}y^{-1}x^2y^{-1}x^{-2}$ | $y^{-4}x^{-1}y^3x$ | $17_{29}$ | 1 |
| $y^{-3}xyx^{-2}$ | $y^{-2}x^2yx^{-1}yx^{-1}y^{-1}x^{-1}$ | $17_{30}$ | 8 |
| $y^{-2}x^2yx^{-1}$ | $y^{-5}x^{-1}yx^2yx$ | $17_{31}$ | 8 |
| $y^{-2}xyx^{-2}$ | $y^{-3}x^{-2}yxyxy^{-1}x^{-1}$ | $17_{32}$ | 8 |
| $y^{-3}xyxyx^{-1}$ | $y^{-1}x^{-1}yx^2yx^{-1}yx^{-1}$ | $17_{33}$ | 8 |
| $y^{-2}x^{-2}yx^2$ | $y^{-5}x^{-1}yx^{-1}yx$ | $17_{34}$ | 5 |
| $y^{-2}x^{-1}yxyx$ | $y^{-1}x^{-1}yx^{-1}y^{-1}x^5$ | $17_{35}$ | 5 |
| $y^{-2}x^{-2}yx^2$ | $y^{-2}xyxyxy^{-1}x^{-2}$ | $17_{36}$ | 5 |
| $y^{-1}x^{-1}yx^{-2}y^{-1}x^2$ | $y^{-4}x^{-1}yx^{-1}yx$ | $17_{37}$ | 4 |
| $y^{-3}x^{-2}yx$ | $y^{-2}x^{-1}y^{-1}xyxyx^{-2}$ | $17_{38}$ | 4 |
| $y^{-3}x^{-1}yxyx$ | $y^{-1}x^{-1}y^{-1}x^{-1}yx^{-1}yx^2$ | $17_{39}$ | 4 |
| $y^{-1}x^{-2}y^{-1}x^2yx$ | $y^{-3}x^{-1}yx^{-1}yx^2$ | $17_{40}$ | 4 |
| $y^{-2}x^{-1}y^{-1}x^{-1}yx$ | $y^{-1}x^{-1}y^{-1}x^3yx^{-3}$ | $17_{41}$ | 2 |
| $y^{-2}x^{-1}yxy^{-1}x$ | $y^{-1}x^{-3}yx^{-1}yx^3$ | $17_{42}$ | 2 |

Table 3 – continued from previous page

| $r_1$ | $r_2$ | name | occurrences |
|---|---|---|---|
| $y^{-1}x^{-2}y^{-1}x^3$ | $y^{-5}x^{-1}y^4x$ | $18_1$ | 1 |
| $y^{-1}x^{-3}yx^2$ | $y^{-5}x^{-1}y^4x$ | $18_2$ | 2 |
| $y^{-1}x^{-3}y^{-1}x^2$ | $y^{-5}x^{-1}y^4x$ | $18_3$ | 1 |
| $y^{-3}x^{-2}y^{-1}xyx^{-1}$ | $y^{-2}xy^{-1}x^{-2}y^{-1}x^{-2}$ | $18_4$ | 1 |
| $y^{-3}xyx^{-1}y^{-1}x^2$ | $y^{-2}x^2y^{-1}x^2y^{-1}x^{-1}$ | $18_5$ | 1 |
| $y^{-1}x^{-1}y^{-1}x^{-1}yx^2$ | $y^{-6}xyx^{-1}yx$ | $18_6$ | 4 |
| $y^{-1}x^{-1}y^{-1}x^{-1}yx^2$ | $y^{-7}x^{-2}yx$ | $18_7$ | 4 |
| $y^{-3}x^{-2}yx$ | $y^{-2}x^{-1}y^{-1}x^{-1}yx^{-1}yx^3$ | $18_8$ | 3 |
| $y^{-3}x^2yx^{-1}$ | $y^{-1}x^{-1}y^{-1}x^{-1}yx^{-1}yx^2y^{-1}x$ | $18_9$ | 3 |
| $y^{-3}x^{-1}yx^2$ | $y^{-2}x^{-3}yxyxy^{-1}x$ | $18_{10}$ | 3 |
| $y^{-3}xyx^{-2}$ | $y^{-1}x^{-1}y^{-1}x^{-2}yxyxy^{-1}x$ | $18_{11}$ | 3 |
| $y^{-2}x^{-2}y^{-1}x^3$ | $y^{-5}x^{-1}y^4x$ | $19_1$ | 1 |
| $y^{-1}x^{-1}yx^3y^{-1}x^{-2}$ | $y^{-4}x^{-3}yx^2$ | $19_2$ | 1 |
| $y^{-1}x^{-2}y^{-1}xy^{-1}x^2$ | $y^{-5}x^{-1}y^4x$ | $19_3$ | 1 |
| $y^{-1}x^{-1}yx^2yx^{-2}$ | $y^{-5}x^{-1}y^4x$ | $19_4$ | 1 |
| $y^{-1}x^{-2}y^{-1}x^2yx$ | $y^{-5}x^{-1}y^4x$ | $19_5$ | 1 |
| $y^{-1}x^{-1}yx^{-2}yx^2$ | $y^{-5}x^{-1}y^4x$ | $19_6$ | 1 |
| $y^{-1}x^{-2}yx^{-1}yx^2$ | $y^{-5}x^{-1}y^4x$ | $19_7$ | 1 |
| $y^{-1}x^{-1}yx^2y^{-1}x^{-2}$ | $y^{-5}x^{-1}y^4x$ | $19_8$ | 1 |
| $y^{-1}x^{-1}y^{-1}x^{-2}yx^2$ | $y^{-5}x^{-1}y^4x$ | $19_9$ | 1 |
| $y^{-2}x^{-3}yx^2$ | $y^{-5}x^{-1}y^4x$ | $19_{10}$ | 1 |
| $y^{-1}x^{-1}y^{-1}x^{-1}y^{-1}x^3$ | $y^{-5}x^{-1}y^4x$ | $19_{11}$ | 1 |
| $y^{-1}x^{-1}y^{-1}x^3yx^{-1}$ | $y^{-5}x^{-1}y^4x$ | $19_{12}$ | 1 |
| $y^{-1}x^{-1}yx^{-1}y^{-1}x^3$ | $y^{-5}x^{-1}y^4x$ | $19_{13}$ | 1 |
| $y^{-1}x^{-3}yxy^{-1}x$ | $y^{-5}x^{-1}y^4x$ | $19_{14}$ | 1 |
| $y^{-2}x^3y^{-1}x^{-2}$ | $y^{-1}x^{-1}y^{-1}x^2yx^{-1}yx^{-1}y^{-1}x$ | $19_{15}$ | 1 |
| $y^{-2}x^3yx^{-2}$ | $y^{-5}x^2yxyx^{-1}$ | $19_{16}$ | 1 |
| $y^{-1}x^{-2}yx^{-1}y^{-1}x^3$ | $y^{-4}x^{-3}y^{-1}x^2$ | $19_{17}$ | 1 |
| $y^{-1}x^{-1}yx^{-2}y^{-1}x^2$ | $y^{-5}x^{-1}y^4x$ | $19_{18}$ | 1 |
| $y^{-1}x^{-2}yxy^{-1}x^2$ | $y^{-5}x^{-1}y^4x$ | $19_{19}$ | 1 |
| $y^{-1}x^{-2}yx^{-1}y^{-1}x^2$ | $y^{-5}x^{-1}y^4x$ | $19_{20}$ | 1 |
| $y^{-2}x^{-2}yx^3$ | $y^{-5}x^{-1}y^4x$ | $19_{21}$ | 1 |
| $y^{-2}x^{-3}y^{-1}x^2$ | $y^{-5}x^{-1}y^4x$ | $19_{22}$ | 1 |
| $y^{-1}x^{-1}y^{-1}x^{-1}yx^3$ | $y^{-5}x^{-1}y^4x$ | $19_{23}$ | 1 |
| $y^{-1}x^{-1}yx^3yx^{-1}$ | $y^{-5}x^{-1}y^4x$ | $19_{24}$ | 1 |
| $y^{-1}x^{-1}yx^{-1}yx^3$ | $y^{-5}x^{-1}y^4x$ | $19_{25}$ | 1 |
| $y^{-1}x^{-3}y^{-1}xy^{-1}x$ | $y^{-5}x^{-1}y^4x$ | $19_{26}$ | 1 |
| $y^{-2}x^2yx^{-3}$ | $y^{-1}x^{-1}yx^2yx^{-1}yx^{-1}y^{-1}x$ | $19_{27}$ | 1 |
| $y^{-2}x^2y^{-1}x^{-3}$ | $y^{-5}xy^{-1}x^{-1}y^{-1}x^{-2}$ | $19_{28}$ | 1 |
| $y^{-1}x^{-2}y^{-1}x^2yx^{-1}$ | $y^{-5}x^{-1}y^4x$ | $19_{29}$ | 1 |
| $y^{-1}x^{-1}y^{-1}x^{-2}y^{-1}x^2$ | $y^{-5}x^{-1}y^4x$ | $19_{30}$ | 1 |
| $y^{-1}x^{-1}y^{-1}x^2yx^{-2}$ | $y^{-5}x^{-1}y^4x$ | $19_{31}$ | 1 |
| $y^{-1}x^{-1}y^{-1}x^2y^{-1}x^{-2}$ | $y^{-5}x^{-1}y^4x$ | $19_{32}$ | 1 |
| $y^{-2}x^{-2}yx$ | $y^{-3}x^{-1}y^{-1}xyxyx^{-2}yx^{-1}$ | $19_{33}$ | 5 |
| $y^{-2}x^{-1}yx^2$ | $y^{-2}x^2y^{-1}x^2yx^{-1}yx^{-1}y^{-1}x$ | $19_{34}$ | 5 |
| $y^{-4}x^{-2}y^2x$ | $y^{-5}xy^2x^{-2}$ | $19_{35}$ | 3 |
| $y^{-2}x^{-1}y^2x^{-1}y^{-2}x$ | $y^{-7}x^{-3}$ | $19_{36}$ | 3 |
| $y^{-2}x^{-2}yx$ | $y^{-1}x^{-1}yx^{-1}yx^{-1}yx^2y^{-1}xy^{-1}x$ | $19_{37}$ | 3 |
| $y^{-2}x^{-1}yx^2$ | $y^{-1}x^{-1}y^{-1}x^{-1}y^{-1}xy^{-1}x^6$ | $19_{38}$ | 1 |
| $y^{-3}x^{-1}y^{-1}xyx$ | $y^{-2}x^{-2}y^{-1}x^2yx^{-1}yx$ | $19_{39}$ | 3 |
| $y^{-3}x^2y^{-1}x^{-1}$ | $y^{-1}x^{-1}yx^3yx^{-1}yx^{-1}yx^{-1}$ | $19_{40}$ | 3 |

Table 3 – continued from previous page

| $r_1$ | $r_2$ | name | occurrences |
|---|---|---|---|
| $y^{-3}x^{-1}yx^{-1}y^{-1}x$ | $y^{-1}x^{-2}y^{-1}xy^{-1}x^{-2}yx^2$ | $19_{41}$ | 3 |
| $y^{-3}xy^{-1}x^{-2}$ | $y^{-1}x^{-1}y^{-1}x^{-1}y^{-1}x^{-1}yx^{-1}y^{-1}x^3$ | $19_{42}$ | 3 |
| $y^{-2}x^{-1}y^2x^2y^{-1}x^{-1}$ | $y^{-5}xyx^{-1}yx^{-1}$ | $19_{43}$ | 1 |
| $y^{-3}x^{-2}y^2x$ | $y^{-3}xy^2x^{-2}y^{-1}x^{-2}$ | $19_{44}$ | 1 |
| $y^{-3}x^2y^2x^{-1}$ | $y^{-4}x^{-1}yx^2y^{-1}x^2$ | $19_{45}$ | 1 |
| $y^{-4}x^2y^{-1}x^{-1}$ | $y^{-3}xyxy^{-1}xy^{-1}x^{-2}$ | $19_{46}$ | 1 |
| $y^{-4}xyxyx^{-1}$ | $y^{-2}xy^{-1}x^{-2}y^{-1}xyx$ | $19_{47}$ | 1 |
| $y^{-2}xy^{-1}x^{-2}y^2x$ | $y^{-5}xyxyx^{-1}$ | $19_{48}$ | 1 |
| $y^{-3}x^{-1}y^2x^2$ | $y^{-3}x^2y^{-1}x^2y^2x^{-1}$ | $19_{49}$ | 1 |
| $y^{-3}xy^2x^{-2}$ | $y^{-4}x^{-2}y^{-1}x^{-2}yx$ | $19_{50}$ | 1 |
| $y^{-4}xyx^{-2}$ | $y^{-1}x^{-1}yx^2yx^{-1}yx^{-1}y^{-1}x$ | $19_{51}$ | 1 |
| $y^{-4}xy^{-1}x^{-2}$ | $y^{-1}x^{-1}y^{-1}x^{-1}yxyx^{-1}yx^2$ | $19_{52}$ | 1 |
| $y^{-3}xy^2x$ | $y^{-1}x^{-5}yx^6$ | $20_1$ | 1 |
| $y^{-1}x^{-2}yx^3$ | $y^{-6}xy^5x^{-1}$ | $20_2$ | 2 |
| $y^{-1}x^{-3}y^{-1}x^2$ | $y^{-6}x^{-1}y^5x$ | $20_3$ | 1 |
| $y^{-1}x^{-1}yx^3y^{-1}x^{-2}$ | $y^{-5}x^{-3}yx^2$ | $20_4$ | 1 |
| $y^{-1}x^{-1}yx^3yx^{-2}$ | $y^{-5}x^{-2}y^{-1}x^3$ | $20_5$ | 1 |
| $y^{-1}x^{-2}yx^{-1}y^{-1}x^3$ | $y^{-5}x^{-3}y^{-1}x^2$ | $20_6$ | 1 |
| $y^{-1}x^{-3}yxy^{-1}x^2$ | $y^{-5}x^{-2}yx^3$ | $20_7$ | 1 |
| $y^{-1}x^{-1}y^{-1}x^2$ | $y^{-7}x^{-1}y^6x$ | $20_8$ | 2 |
| $y^{-1}x^{-2}y^{-1}x$ | $y^{-7}x^{-1}y^6x$ | $20_9$ | 1 |
| $y^{-2}x^{-1}y^{-2}xy^2x$ | $y^{-7}x^4$ | $20_{10}$ | 3 |
| $y^{-2}x^{-1}y^2x^{-1}y^{-2}x$ | $y^{-7}x^{-4}$ | $20_{11}$ | 3 |
| $y^{-2}x^{-1}yxyx$ | $y^{-3}x^{-2}y^2x^{-1}yxy^{-1}x^{-2}$ | $20_{12}$ | 2 |
| $y^{-1}x^{-1}y^{-1}xyx$ | $y^{-8}x^7$ | $21_1$ | 1 |
| $y^{-2}x^{-2}y^{-1}x^3$ | $y^{-6}x^{-1}y^5x$ | $21_2$ | 1 |
| $y^{-1}x^{-2}y^{-1}xy^{-1}x^2$ | $y^{-6}x^{-1}y^5x$ | $21_3$ | 1 |
| $y^{-1}x^{-1}yx^2yx^{-2}$ | $y^{-6}x^{-1}y^5x$ | $21_4$ | 1 |
| $y^{-1}x^{-2}y^{-1}x^2yx$ | $y^{-6}x^{-1}y^5x$ | $21_5$ | 1 |
| $y^{-1}x^{-1}yx^{-2}yx^2$ | $y^{-6}x^{-1}y^5x$ | $21_6$ | 1 |
| $y^{-1}x^{-1}yx^{-2}y^{-1}x^3$ | $y^{-6}x^3yx^{-2}$ | $21_7$ | 1 |
| $y^{-1}x^{-1}yx^2y^{-1}x^{-2}$ | $y^{-6}x^{-1}y^5x$ | $21_8$ | 1 |
| $y^{-2}x^{-3}yx^2$ | $y^{-3}x^{-2}y^{-1}xyx^{-1}yx^{-1}yx$ | $21_9$ | 1 |
| $y^{-1}x^{-1}y^{-1}x^{-1}y^{-1}x^3$ | $y^{-6}x^{-1}y^5x$ | $21_{10}$ | 1 |
| $y^{-1}x^{-1}y^{-1}x^3yx^{-1}$ | $y^{-6}x^{-1}y^5x$ | $21_{11}$ | 1 |
| $y^{-1}x^{-1}yx^{-1}y^{-1}x^3$ | $y^{-6}x^{-1}y^5x$ | $21_{12}$ | 1 |
| $y^{-1}x^{-3}yxy^{-1}x$ | $y^{-6}x^{-1}y^5x$ | $21_{13}$ | 1 |
| $y^{-2}x^3y^{-1}x^{-2}$ | $y^{-6}x^{-1}y^5x$ | $21_{14}$ | 1 |
| $y^{-2}x^3yx^{-2}$ | $y^{-6}x^{-1}y^5x$ | $21_{15}$ | 1 |
| $y^{-1}x^{-1}yx^{-2}y^{-1}x^2$ | $y^{-6}x^{-1}y^5x$ | $21_{16}$ | 1 |
| $y^{-1}x^{-2}yxy^{-1}x^2$ | $y^{-6}x^{-1}y^5x$ | $21_{17}$ | 1 |
| $y^{-1}x^{-2}yx^{-1}y^{-1}x^2$ | $y^{-6}x^{-1}y^5x$ | $21_{18}$ | 1 |
| $y^{-2}x^{-2}yx^3$ | $y^{-1}x^{-1}y^{-1}x^{-1}y^{-1}xyx^{-2}yxyx$ | $21_{19}$ | 1 |
| $y^{-2}x^{-3}y^{-1}x^2$ | $y^{-1}x^{-1}y^{-1}x^{-1}y^{-1}xy^{-1}x^{-2}yxyx$ | $21_{20}$ | 1 |
| $y^{-1}x^{-1}y^{-1}x^{-1}yx^3$ | $y^{-6}x^{-1}y^5x$ | $21_{21}$ | 1 |
| $y^{-1}x^{-1}yx^3yx^{-1}$ | $y^{-6}x^{-1}y^5x$ | $21_{22}$ | 1 |
| $y^{-1}x^{-1}yx^{-1}yx^3$ | $y^{-6}x^{-1}y^5x$ | $21_{23}$ | 1 |
| $y^{-1}x^{-3}y^{-1}xy^{-1}x$ | $y^{-6}x^{-1}y^5x$ | $21_{24}$ | 1 |
| $y^{-2}x^2yx^{-3}$ | $y^{-6}x^{-1}y^5x$ | $21_{25}$ | 1 |
| $y^{-2}x^2y^{-1}x^{-3}$ | $y^{-6}x^{-1}y^5x$ | $21_{26}$ | 1 |
| $y^{-1}x^{-2}y^{-1}x^2yx^{-1}$ | $y^{-6}x^{-1}y^5x$ | $21_{27}$ | 1 |

Table 3 – continued from previous page

| $r_1$ | $r_2$ | name | occurrences |
|---|---|---|---|
| $y^{-1}x^{-1}y^{-1}x^2yx^{-2}$ | $y^{-6}x^{-1}y^5x$ | $21_{28}$ | 1 |
| $y^{-1}x^{-1}yx^{-2}yx^3$ | $y^{-6}x^2y^{-1}x^{-3}$ | $21_{29}$ | 1 |
| $y^{-1}x^{-1}yx^3y^{-1}x^{-2}$ | $y^{-6}x^{-3}yx^2$ | $21_{30}$ | 1 |
| $y^{-1}x^{-1}yx^3yx^{-2}$ | $y^{-6}x^{-2}y^{-1}x^3$ | $21_{31}$ | 1 |
| $y^{-1}x^{-2}yx^{-1}y^{-1}x^3$ | $y^{-6}x^{-3}y^{-1}x^2$ | $21_{32}$ | 1 |
| $y^{-1}x^{-3}yxy^{-1}x^2$ | $y^{-6}x^{-2}yx^3$ | $21_{33}$ | 1 |
| $y^{-2}x^{-1}y^{-1}x^2$ | $y^{-6}xy^{-1}xy^{-1}xy^{-1}x^{-1}yx$ | $21_{34}$ | 3 |
| $y^{-2}x^2yx^{-1}$ | $y^{-7}xyxyx^{-1}yx^2$ | $21_{35}$ | 5 |
| $y^{-1}x^{-1}yx^{-1}y^{-1}x$ | $y^{-7}x^{-8}$ | $21_{36}$ | 1 |
| $y^{-2}x^{-2}y^{-1}x$ | $y^{-2}xy^{-1}xyxyx^{-1}y^{-1}x^{-1}y^{-1}x^{-1}y^{-1}x^{-1}$ | $21_{37}$ | 3 |
| $y^{-2}xyx^{-2}$ | $y^{-7}x^{-1}yx^{-2}yxyx^{-1}$ | $21_{38}$ | 5 |
| $y^{-3}x^{-2}y^2x^2$ | $y^{-5}x^{-1}y^2x^{-1}y^2x$ | $21_{39}$ | 1 |
| $y^{-4}xy^2x^{-1}yx^{-1}$ | $y^{-2}x^{-1}y^{-1}x^{-1}yx^{-1}y^{-1}xyx$ | $21_{40}$ | 1 |
| $y^{-3}x^{-2}y^2x^2$ | $y^{-5}x^{-1}y^2xy^2x$ | $21_{41}$ | 1 |
| $y^{-4}x^2y^2x^{-1}$ | $y^{-4}x^{-1}y^{-1}x^2y^{-2}x^2$ | $21_{42}$ | 1 |
| $y^{-1}x^{-2}y^{-1}x^3$ | $y^{-7}x^{-1}y^6x$ | $22_1$ | 1 |
| $y^{-3}xy^2x^{-1}$ | $y^{-1}x^{-7}yx^6$ | $22_2$ | 1 |
| $y^{-3}xy^2x^{-1}$ | $y^{-1}x^{-6}yx^7$ | $22_3$ | 1 |
| $y^{-1}x^{-3}y^{-1}x^2$ | $y^{-7}x^{-1}y^6x$ | $22_4$ | 1 |
| $y^{-1}x^{-2}y^{-1}x^3yx^{-1}$ | $y^{-7}x^3y^{-1}x^{-2}$ | $22_5$ | 1 |
| $y^{-5}x^2yx^{-3}$ | $y^{-2}x^{-1}yx^2yxy^{-1}x^{-2}$ | $22_6$ | 1 |
| $y^{-1}x^{-1}yx^{-2}y^{-1}x^3$ | $y^{-7}x^3yx^{-2}$ | $22_7$ | 1 |
| $y^{-5}x^2y^{-1}x^{-3}$ | $y^{-2}x^{-1}yx^2y^{-1}xyx^{-2}$ | $22_8$ | 1 |
| $y^{-5}x^3y^{-1}x^{-2}$ | $y^{-2}x^2yx^{-1}y^{-1}x^{-2}yx$ | $22_9$ | 1 |
| $y^{-5}x^3yx^{-2}$ | $y^{-2}x^2y^{-1}x^{-1}yx^{-2}yx$ | $22_{10}$ | 1 |
| $y^{-1}x^{-2}yx^3yx^{-1}$ | $y^{-7}x^2yx^{-3}$ | $22_{11}$ | 1 |
| $y^{-1}x^{-1}yx^{-2}yx^3$ | $y^{-7}x^2y^{-1}x^{-3}$ | $22_{12}$ | 1 |
| $y^{-1}x^{-1}y^{-1}x^2$ | $y^{-8}xy^7x^{-1}$ | $22_{13}$ | 1 |
| $y^{-1}x^{-2}y^{-1}x$ | $y^{-8}x^{-1}y^7x$ | $22_{14}$ | 1 |
| $y^{-2}x^{-2}y^{-1}x^3$ | $y^{-7}x^{-1}y^6x$ | $23_1$ | 1 |
| $y^{-1}x^{-2}y^{-1}x^2yx$ | $y^{-7}x^{-1}y^6x$ | $23_2$ | 1 |
| $y^{-1}x^{-1}yx^{-2}yx^2$ | $y^{-7}x^{-1}y^6x$ | $23_3$ | 1 |
| $y^{-2}x^{-3}yx^2$ | $y^{-7}x^{-1}y^6x$ | $23_4$ | 1 |
| $y^{-1}x^{-1}y^{-1}x^{-1}y^{-1}x^3$ | $y^{-7}x^{-1}y^6x$ | $23_5$ | 1 |
| $y^{-1}x^{-1}y^{-1}x^3yx^{-1}$ | $y^{-7}x^{-1}y^6x$ | $23_6$ | 1 |
| $y^{-1}x^{-1}yx^{-1}y^{-1}x^3$ | $y^{-7}x^{-1}y^6x$ | $23_7$ | 1 |
| $y^{-1}x^{-3}yxy^{-1}x$ | $y^{-7}x^{-1}y^6x$ | $23_8$ | 1 |
| $y^{-2}x^3y^{-1}x^{-2}$ | $y^{-1}x^{-1}y^{-1}x^2yx^{-1}yx^{-1}yx^{-1}y^{-1}xy^{-1}x$ | $23_9$ | 1 |
| $y^{-2}x^3yx^{-2}$ | $y^{-5}x^2y^{-1}x^{-1}yxyxyx^{-1}$ | $23_{10}$ | 1 |
| $y^{-1}x^{-1}yx^{-2}y^{-1}x^2$ | $y^{-7}x^{-1}y^6x$ | $23_{11}$ | 1 |
| $y^{-2}x^{-2}yx^3$ | $y^{-7}x^{-1}y^6x$ | $23_{12}$ | 1 |
| $y^{-2}x^{-3}y^{-1}x^2$ | $y^{-7}x^{-1}y^6x$ | $23_{13}$ | 1 |
| $y^{-1}x^{-1}y^{-1}x^{-1}yx^3$ | $y^{-7}x^{-1}y^6x$ | $23_{14}$ | 1 |
| $y^{-1}x^{-1}yx^3yx^{-1}$ | $y^{-7}x^{-1}y^6x$ | $23_{15}$ | 1 |
| $y^{-1}x^{-1}yx^{-1}yx^3$ | $y^{-7}x^{-1}y^6x$ | $23_{16}$ | 1 |
| $y^{-1}x^{-3}y^{-1}xy^{-1}x$ | $y^{-7}x^{-1}y^6x$ | $23_{17}$ | 1 |
| $y^{-2}x^2yx^{-3}$ | $y^{-7}x^{-1}y^6x$ | $23_{18}$ | 1 |
| $y^{-2}x^2y^{-1}x^{-3}$ | $y^{-5}xy^{-1}x^{-1}y^{-1}x^{-1}y^{-1}xyx^{-2}$ | $23_{19}$ | 1 |
| $y^{-1}x^{-2}y^{-1}x^2yx^{-1}$ | $y^{-7}x^{-1}y^6x$ | $23_{20}$ | 1 |
| $y^{-1}x^{-1}y^{-1}xy^{-1}x$ | $y^{-8}xy^7x^{-1}$ | $23_{21}$ | 1 |
| $y^{-2}x^2y^{-1}x^{-1}$ | $y^{-5}xyx^{-1}yx^{-1}y^{-1}x^{-1}y^3x^2$ | $23_{22}$ | 4 |

Table 3 – continued from previous page

| $r_1$ | $r_2$ | name | occurrences |
|---|---|---|---|
| $y^{-2}xy^{-1}x^{-2}$ | $y^{-8}x^{-1}y^5xy^{-1}x^{-1}$ | $23_{23}$ | 4 |
| $y^{-1}x^{-1}y^{-1}x^{-1}y^{-1}x$ | $y^{-8}x^{-1}y^7x$ | $23_{24}$ | 1 |
| $y^{-1}x^{-2}y^{-1}x^3$ | $y^{-8}x^{-1}y^7x$ | $24_1$ | 1 |
| $y^{-3}x^{-1}y^2x$ | $y^{-1}x^{-7}yx^8$ | $24_2$ | 1 |
| $y^{-1}x^{-2}yx^3$ | $y^{-8}x^{-1}y^7x$ | $24_3$ | 1 |
| $y^{-1}x^{-3}y^{-1}x^2$ | $y^{-8}x^{-1}y^7x$ | $24_4$ | 1 |
| $y^{-2}x^{-2}y^{-1}x^3$ | $y^{-8}x^{-1}y^7x$ | $25_1$ | 1 |
| $y^{-1}x^{-2}y^{-1}x^2y^{-1}x$ | $y^{-8}x^{-1}y^7x$ | $25_2$ | 1 |
| $y^{-1}x^{-2}y^{-1}xy^{-1}x^2$ | $y^{-8}x^{-1}y^7x$ | $25_3$ | 1 |
| $y^{-1}x^{-1}yx^2yx^{-2}$ | $y^{-8}x^{-1}y^7x$ | $25_4$ | 1 |
| $y^{-1}x^{-2}y^{-1}x^2yx$ | $y^{-8}x^{-1}y^7x$ | $25_5$ | 1 |
| $y^{-1}x^{-1}yx^{-2}yx^2$ | $y^{-8}x^{-1}y^7x$ | $25_6$ | 1 |
| $y^{-1}x^{-2}yx^{-1}yx^2$ | $y^{-8}x^{-1}y^7x$ | $25_7$ | 1 |
| $y^{-1}x^{-1}yx^2y^{-1}x^{-2}$ | $y^{-8}x^{-1}y^7x$ | $25_8$ | 1 |
| $y^{-1}x^{-1}y^{-1}x^{-2}yx^2$ | $y^{-8}x^{-1}y^7x$ | $25_9$ | 1 |
| $y^{-2}x^{-3}yx^2$ | $y^{-7}x^{-2}yx^{-1}yx^{-1}yx^{-1}yx$ | $25_{10}$ | 1 |
| $y^{-1}x^{-1}y^{-1}x^{-1}y^{-1}x^3$ | $y^{-8}x^{-1}y^7x$ | $25_{11}$ | 1 |
| $y^{-1}x^{-1}y^{-1}x^3yx^{-1}$ | $y^{-8}x^{-1}y^7x$ | $25_{12}$ | 1 |
| $y^{-1}x^{-1}yx^{-1}y^{-1}x^3$ | $y^{-8}x^{-1}y^7x$ | $25_{13}$ | 1 |
| $y^{-1}x^{-3}yxy^{-1}x$ | $y^{-8}x^{-1}y^7x$ | $25_{14}$ | 1 |
| $y^{-2}x^3y^{-1}x^{-2}$ | $y^{-8}x^{-1}y^7x$ | $25_{15}$ | 1 |
| $y^{-2}x^3yx^{-2}$ | $y^{-8}x^{-1}y^7x$ | $25_{16}$ | 1 |
| $y^{-1}x^{-2}yx^2y^{-1}x$ | $y^{-8}x^{-1}y^7x$ | $25_{17}$ | 1 |
| $y^{-1}x^{-1}yx^{-2}y^{-1}x^2$ | $y^{-8}x^{-1}y^7x$ | $25_{18}$ | 1 |
| $y^{-1}x^{-2}yxy^{-1}x^2$ | $y^{-8}x^{-1}y^7x$ | $25_{19}$ | 1 |
| $y^{-1}x^{-2}yx^{-1}y^{-1}x^2$ | $y^{-8}x^{-1}y^7x$ | $25_{20}$ | 1 |
| $y^{-2}x^{-2}yx^3$ | $y^{-3}x^{-1}yxyxyxyx^{-1}y^{-1}x^{-1}y^{-1}x^2$ | $25_{21}$ | 1 |
| $y^{-2}x^{-3}y^{-1}x^2$ | $y^{-3}x^{-2}yxyxy^{-1}x^{-1}y^{-1}x^{-1}y^{-1}x^{-1}y^{-1}x$ | $25_{22}$ | 1 |
| $y^{-1}x^{-1}y^{-1}x^{-1}yx^3$ | $y^{-8}x^{-1}y^7x$ | $25_{23}$ | 1 |
| $y^{-1}x^{-1}yx^3yx^{-1}$ | $y^{-8}x^{-1}y^7x$ | $25_{24}$ | 1 |
| $y^{-1}x^{-1}yx^{-1}yx^3$ | $y^{-8}x^{-1}y^7x$ | $25_{25}$ | 1 |
| $y^{-1}x^{-3}y^{-1}xy^{-1}x$ | $y^{-8}x^{-1}y^7x$ | $25_{26}$ | 1 |
| $y^{-2}x^2yx^{-3}$ | $y^{-8}x^{-1}y^7x$ | $25_{27}$ | 1 |
| $y^{-2}x^2y^{-1}x^{-3}$ | $y^{-8}x^{-1}y^7x$ | $25_{28}$ | 1 |
| $y^{-1}x^{-2}y^{-1}x^2yx^{-1}$ | $y^{-8}x^{-1}y^7x$ | $25_{29}$ | 1 |
| $y^{-1}x^{-1}y^{-1}x^{-2}y^{-1}x^2$ | $y^{-8}x^{-1}y^7x$ | $25_{30}$ | 1 |
| $y^{-1}x^{-1}y^{-1}x^2yx^{-2}$ | $y^{-8}x^{-1}y^7x$ | $25_{31}$ | 1 |
| $y^{-1}x^{-1}y^{-1}x^2y^{-1}x^{-2}$ | $y^{-8}x^{-1}y^7x$ | $25_{32}$ | 1 |

# G. PPO and GS Path Lengths

This appendix provides detailed path length comparisons between PPO and greedy search (GS) for all presentations solved by both methods.

The table shows, for each Miller–Schupp presentation $(n, w)$ solved by both PPO and GS:

- The presentation

- The solution path length found by PPO-Sub-DRT and PPO-Sub-DRT + **AC-19** (min over 5 seeds)

- The solution path length found by GS-AC and GS-Sub (deterministic)

As discussed in the main text (Result #3), PPO consistently finds substantially shorter paths than greedy search. The

advantage is especially pronounced on harder presentations, where PPO's learned value function identifies length-increasing detours that enable more efficient overall solutions. This demonstrates that the learned policy discovers non-obvious solution strategies beyond what the greedy length-reduction heuristic can find.

*Table 4.* Path lengths for presentations where at least one PPO method found a solution.

| Presentation | GS-AC | GS-Sub | PPO-Sub-DRT | PPO-Sub-DRT + **AC-19** |
|---|---|---|---|---|
| $MS(1, y)$ | 5 | 2 | 2 | 2 |
| $MS(1, y^{-1})$ | 5 | 2 | 2 | 2 |
| $MS(1, y^2)$ | 5 | 3 | 3 | 3 |
| $MS(1, y^{-2})$ | 5 | 3 | 3 | 3 |
| $MS(1, y^3)$ | 7 | 4 | 4 | 4 |
| $MS(1, y^{-3})$ | 8 | 4 | 4 | 4 |
| $MS(1, yxyx^{-1})$ | 11 | 5 | 5 | 5 |
| $MS(1, yxy^{-1}x^{-1})$ | 7 | 3 | 3 | 3 |
| $MS(1, y^4)$ | 8 | 5 | 5 | 5 |
| $MS(1, x^{-1}y^{-1}xy)$ | 8 | 3 | 3 | 3 |
| $MS(1, x^{-1}y^{-1}xy^{-1})$ | 11 | 5 | 5 | 5 |
| $MS(1, y^{-4})$ | 8 | 5 | 5 | 5 |
| $MS(1, yxy^2x^{-1})$ | 11 | 4 | 4 | 4 |
| $MS(1, yxyx^{-1}y)$ | 15 | 6 | 6 | 6 |
| $MS(1, yxyx^{-1}y^{-1})$ | 10 | 4 | 4 | 4 |
| $MS(1, yxy^{-1}x^{-1}y)$ | 13 | 4 | 3 | 4 |
| $MS(1, yxy^{-1}x^{-1}y^{-1})$ | 8 | 3 | 2 | 2 |
| $MS(1, yxy^{-2}x^{-1})$ | 7 | 3 | 3 | 3 |
| $MS(1, y^2xyx^{-1})$ | 23 | 8 | 8 | 8 |
| $MS(1, y^2xy^{-1}x^{-1})$ | 12 | 6 | 6 | 6 |
| $MS(1, y^5)$ | 15 | 6 | 6 | 6 |
| $MS(1, yx^{-1}y^{-1}xy)$ | 6 | 3 | 2 | 2 |
| $MS(1, yx^{-1}y^{-1}xy^{-1})$ | 8 | 4 | 4 | 4 |
| $MS(1, x^{-1}y^{-1}xy^2)$ | 6 | 3 | 3 | 3 |
| $MS(1, x^{-1}y^{-1}xy^{-2})$ | 7 | 4 | 4 | 4 |
| $MS(1, x^{-1}y^{-2}xy)$ | 13 | 6 | 6 | 6 |
| $MS(1, x^{-1}y^{-2}xy^{-1})$ | 16 | 8 | 8 | 8 |
| $MS(1, y^{-1}xyx^{-1}y^{-1})$ | 13 | 4 | 4 | 4 |
| $MS(1, y^{-1}xy^{-1}x^{-1}y^{-1})$ | 12 | 6 | 6 | 6 |
| $MS(1, y^{-5})$ | 10 | 6 | 6 | 6 |
| $MS(1, yx^2yx^{-2})$ | 26 | 9 | 9 | 9 |
| $MS(1, yx^2y^{-1}x^{-2})$ | 19 | 7 | 7 | 7 |
| $MS(1, yxy^3x^{-1})$ | 17 | 7 | 7 | 7 |
| $MS(1, yxy^2x^{-1}y)$ | 14 | 5 | 5 | 5 |
| $MS(1, yxy^2x^{-1}y^{-1})$ | 7 | 3 | 3 | 3 |
| $MS(1, yxyx^{-1}y^2)$ | 21 | 7 | 7 | 7 |
| $MS(1, yxyx^{-1}y^{-2})$ | 11 | 3 | 3 | 3 |
| $MS(1, yxy^{-1}x^{-1}y^2)$ | 17 | 5 | 5 | 5 |
| $MS(1, yxy^{-1}x^{-1}y^{-2})$ | 8 | 3 | 3 | 3 |
| $MS(1, yxy^{-2}x^{-1}y)$ | 8 | 3 | 3 | 3 |
| $MS(1, yxy^{-2}x^{-1}y^{-1})$ | 8 | 3 | 3 | 3 |
| $MS(1, yxy^{-3}x^{-1})$ | 11 | 3 | 3 | 3 |
| $MS(1, y^2xy^2x^{-1})$ | 12 | 5 | 5 | 5 |
| $MS(1, y^2xyx^{-1}y)$ | 34 | 9 | 9 | 9 |
| $MS(1, y^2xyx^{-1}y^{-1})$ | 15 | 7 | 7 | 7 |
| $MS(1, y^2xy^{-1}x^{-1}y)$ | 16 | 7 | 7 | 7 |
| $MS(1, y^2xy^{-1}x^{-1}y^{-1})$ | 12 | 5 | 5 | 5 |

Table 4 – continued from previous page

| Presentation | GS-AC | GS-Sub | PPO-Sub-DRT | PPO-Sub-DRT + **AC-19** |
|---|---|---|---|---|
| $MS(1, y^2xy^{-2}x^{-1})$ | 8 | 3 | 3 | 3 |
| $MS(1, y^3xyx^{-1})$ | 32 | 11 | 11 | 11 |
| $MS(1, y^3xy^{-1}x^{-1})$ | 24 | 9 | 9 | 9 |
| $MS(1, y^6)$ | 17 | 7 | 7 | 7 |
| $MS(1, y^2x^{-1}y^{-1}xy)$ | 6 | 3 | 3 | 3 |
| $MS(1, y^2x^{-1}y^{-1}xy^{-1})$ | 6 | 3 | 3 | 3 |
| $MS(1, yx^{-1}y^{-1}xy^2)$ | 7 | 3 | 3 | 3 |
| $MS(1, yx^{-1}y^{-1}xy^{-2})$ | 8 | 3 | 3 | 3 |
| $MS(1, yx^{-1}y^{-2}xy)$ | 11 | 5 | 5 | 5 |
| $MS(1, yx^{-1}y^{-2}xy^{-1})$ | 20 | 7 | 7 | 7 |
| $MS(1, x^{-2}y^{-1}x^2y)$ | 18 | 7 | 7 | 7 |
| $MS(1, x^{-2}y^{-1}x^2y^{-1})$ | 19 | 9 | 9 | 9 |
| $MS(1, x^{-1}y^{-1}xy^3)$ | 9 | 3 | 3 | 3 |
| $MS(1, x^{-1}y^{-1}xy^{-3})$ | 15 | 7 | 7 | 7 |
| $MS(1, x^{-1}y^{-2}xy^2)$ | 9 | 3 | 3 | 3 |
| $MS(1, x^{-1}y^{-2}xy^{-2})$ | 10 | 5 | 5 | 5 |
| $MS(1, x^{-1}y^{-3}xy)$ | 19 | 9 | 9 | 9 |
| $MS(1, x^{-1}y^{-3}xy^{-1})$ | 34 | 11 | 11 | 11 |
| $MS(1, y^{-1}xy^2x^{-1}y^{-1})$ | 6 | 3 | 3 | 3 |
| $MS(1, y^{-1}xyx^{-1}y^{-2})$ | 11 | 5 | 5 | 5 |
| $MS(1, y^{-1}xy^{-1}x^{-1}y^{-2})$ | 15 | 7 | 7 | 7 |
| $MS(1, y^{-1}xy^{-2}x^{-1}y^{-1})$ | 13 | 5 | 5 | 5 |
| $MS(1, y^{-1}x^{-1}y^{-2}xy)$ | 14 | 7 | 7 | 7 |
| $MS(1, y^{-1}x^{-1}y^{-2}xy^{-1})$ | 19 | 9 | 9 | 9 |
| $MS(1, y^{-6})$ | 12 | 7 | 7 | 7 |
| $MS(1, yx^2y^2x^{-2})$ | 13 | 6 | 6 | 6 |
| $MS(1, yx^2yx^{-1}yx^{-1})$ | 29 | 10 | 10 | 10 |
| $MS(1, yx^2yx^{-2}y)$ | 44 | 12 | 12 | 12 |
| $MS(1, yx^2yx^{-2}y^{-1})$ | 15 | 6 | 6 | 6 |
| $MS(1, yx^2yx^{-1}y^{-1}x^{-1})$ | 22 | 8 | 8 | 8 |
| $MS(1, yx^2y^{-1}x^{-1}yx^{-1})$ | 29 | 8 | 8 | 8 |
| $MS(1, yx^2y^{-1}x^{-2}y)$ | 31 | 10 | 10 | 10 |
| $MS(1, yx^2y^{-1}x^{-2}y^{-1})$ | 14 | 4 | 4 | 4 |
| $MS(1, yx^2y^{-1}x^{-1}y^{-1}x^{-1})$ | 21 | 6 | 6 | 6 |
| $MS(1, yx^2y^{-2}x^{-2})$ | 10 | 4 | 4 | 4 |
| $MS(1, yxyxyx^{-2})$ | 45 | 12 | 12 | 12 |
| $MS(1, yxyxy^{-1}x^{-2})$ | 28 | 10 | 10 | 10 |
| $MS(1, yxy^4x^{-1})$ | 15 | 6 | 6 | 6 |
| $MS(1, yxy^3x^{-1}y)$ | 24 | 8 | 8 | 8 |
| $MS(1, yxy^3x^{-1}y^{-1})$ | 13 | 6 | 6 | 6 |
| $MS(1, yxy^2x^{-1}y^2)$ | 16 | 6 | 6 | 6 |
| $MS(1, yxy^2x^{-1}y^{-2})$ | 8 | 3 | 2 | 3 |
| $MS(1, yxyx^{-1}y^3)$ | 23 | 8 | 8 | 8 |
| $MS(1, yxyx^{-1}y^{-3})$ | 10 | 3 | 3 | 3 |
| $MS(1, yxy^{-1}xyx^{-2})$ | 21 | 6 | 6 | 6 |
| $MS(1, yxy^{-1}xy^{-1}x^{-2})$ | 14 | 4 | 4 | 4 |
| $MS(1, yxy^{-1}x^{-1}y^3)$ | 22 | 6 | 6 | 6 |
| $MS(1, yxy^{-1}x^{-1}y^{-3})$ | 10 | 4 | 4 | 4 |
| $MS(1, yxy^{-2}x^{-1}y^2)$ | 9 | 4 | 4 | 4 |
| $MS(1, yxy^{-2}x^{-1}y^{-2})$ | 9 | 4 | 4 | 4 |
| $MS(1, yxy^{-3}x^{-1}y)$ | 12 | 3 | 3 | 3 |

Table 4 – continued from previous page

| Presentation | GS-AC | GS-Sub | PPO-Sub-DRT | PPO-Sub-DRT + **AC-19** |
|---|---|---|---|---|
| MS(1, $yxy^{-3}x^{-1}y^{-1}$) | 11 | 4 | 4 | 4 |
| MS(1, $yxy^{-4}x^{-1}$) | 7 | 4 | 4 | 4 |
| MS(1, $y^2x^2yx^{-2}$) | 65 | 18 | 16 | 16 |
| MS(1, $y^2x^2y^{-1}x^{-2}$) | 49 | 14 | 14 | 14 |
| MS(1, $y^2xy^3x^{-1}$) | 35 | 10 | 10 | 10 |
| MS(1, $y^2xy^2x^{-1}y$) | 16 | 6 | 6 | 6 |
| MS(1, $y^2xy^2x^{-1}y^{-1}$) | 8 | 4 | 4 | 4 |
| MS(1, $y^2xyx^{-1}y^2$) | 36 | 12 | 10 | 10 |
| MS(1, $y^2xyx^{-1}y^{-2}$) | 15 | 6 | 6 | 6 |
| MS(1, $y^2xy^{-1}x^{-1}y^2$) | 31 | 8 | 8 | 8 |
| MS(1, $y^2xy^{-1}x^{-1}y^{-2}$) | 14 | 4 | 4 | 4 |
| MS(1, $y^2xy^{-2}x^{-1}y$) | 9 | 4 | 4 | 4 |
| MS(1, $y^2xy^{-2}x^{-1}y^{-1}$) | 10 | 3 | 3 | 3 |
| MS(1, $y^2xy^{-3}x^{-1}$) | 18 | 4 | 4 | 4 |
| MS(1, $y^3xy^2x^{-1}$) | 14 | 6 | 6 | 6 |
| MS(1, $y^3xyx^{-1}y$) | 52 | 14 | 12 | 12 |
| MS(1, $y^3xyx^{-1}y^{-1}$) | 27 | 10 | 10 | 10 |
| MS(1, $y^3xy^{-1}x^{-1}y$) | 28 | 10 | 10 | 10 |
| MS(1, $y^3xy^{-1}x^{-1}y^{-1}$) | 22 | 8 | 8 | 8 |
| MS(1, $y^3xy^{-2}x^{-1}$) | 8 | 4 | 4 | 4 |
| MS(1, $y^4xyx^{-1}$) | 50 | 16 | 14 | 14 |
| MS(1, $y^4xy^{-1}x^{-1}$) | 42 | 12 | 12 | 12 |
| MS(1, $y^7$) | 19 | 8 | 8 | 8 |
| MS(1, $y^3x^{-1}y^{-1}xy$) | 7 | 4 | 4 | 4 |
| MS(1, $y^3x^{-1}y^{-1}xy^{-1}$) | 8 | 3 | 3 | 3 |
| MS(1, $y^2x^{-1}y^{-1}xy^2$) | 9 | 4 | 4 | 4 |
| MS(1, $y^2x^{-1}y^{-1}xy^{-2}$) | 6 | 3 | 2 | 2 |
| MS(1, $y^2x^{-1}y^{-2}xy$) | 10 | 4 | 4 | 4 |
| MS(1, $y^2x^{-1}y^{-2}xy^{-1}$) | 12 | 6 | 6 | 6 |
| MS(1, $yx^{-2}y^{-1}x^2y$) | 19 | 6 | 6 | 6 |
| MS(1, $yx^{-2}y^{-1}x^2y^{-1}$) | 19 | 8 | 8 | 8 |
| MS(1, $yx^{-1}y^{-1}x^2yx^{-1}$) | 11 | 4 | 4 | 4 |
| MS(1, $yx^{-1}y^{-1}x^2y^{-1}x^{-1}$) | 14 | 6 | 6 | 6 |
| MS(1, $yx^{-1}y^{-1}xy^3$) | 11 | 4 | 4 | 4 |
| MS(1, $yx^{-1}y^{-1}xy^{-3}$) | 19 | 6 | 6 | 6 |
| MS(1, $yx^{-1}y^{-2}xy^2$) | 8 | 3 | 3 | 3 |
| MS(1, $yx^{-1}y^{-2}xy^{-2}$) | 8 | 4 | 4 | 4 |
| MS(1, $yx^{-1}y^{-3}xy$) | 22 | 8 | 8 | 8 |
| MS(1, $yx^{-1}y^{-3}xy^{-1}$) | 26 | 10 | 10 | 10 |
| MS(1, $x^{-2}y^{-1}x^2y^2$) | 11 | 4 | 4 | 4 |
| MS(1, $x^{-2}y^{-1}x^2y^{-2}$) | 13 | 6 | 6 | 6 |
| MS(1, $x^{-2}y^{-1}xyxy$) | 13 | 4 | 4 | 4 |
| MS(1, $x^{-2}y^{-1}xyxy^{-1}$) | 21 | 6 | 6 | 6 |
| MS(1, $x^{-2}y^{-1}xy^{-1}xy$) | 22 | 10 | 10 | 10 |
| MS(1, $x^{-2}y^{-1}xy^{-1}xy^{-1}$) | 39 | 12 | 12 | 12 |
| MS(1, $x^{-2}y^{-2}x^2y$) | 43 | 14 | 14 | 13 |
| MS(1, $x^{-2}y^{-2}x^2y^{-1}$) | 60 | 18 | 16 | 16 |
| MS(1, $x^{-1}y^{-1}x^2yx^{-1}y^{-1}$) | 23 | 8 | 8 | 8 |
| MS(1, $x^{-1}y^{-1}x^2y^{-1}x^{-1}y^{-1}$) | 24 | 10 | 10 | 10 |
| MS(1, $x^{-1}y^{-1}xy^4$) | 11 | 4 | 4 | 4 |
| MS(1, $x^{-1}y^{-1}xy^{-4}$) | 15 | 6 | 6 | 6 |

Table 4 – continued from previous page

| Presentation | GS-AC | GS-Sub | PPO-Sub-DRT | PPO-Sub-DRT + **AC-19** |
|---|---|---|---|---|
| MS(1, $x^{-1}y^{-1}x^{-1}y^{-1}x^2y$) | 24 | 10 | 10 | 10 |
| MS(1, $x^{-1}y^{-1}x^{-1}y^{-1}x^2y^{-1}$) | 37 | 12 | 12 | 13 |
| MS(1, $x^{-1}y^{-2}xy^3$) | 12 | 4 | 4 | 4 |
| MS(1, $x^{-1}y^{-2}xy^{-3}$) | 31 | 10 | 10 | 10 |
| MS(1, $x^{-1}y^{-3}xy^2$) | 10 | 4 | 4 | 4 |
| MS(1, $x^{-1}y^{-3}xy^{-2}$) | 10 | 6 | 6 | 6 |
| MS(1, $x^{-1}y^{-4}xy$) | 37 | 12 | 12 | 12 |
| MS(1, $x^{-1}y^{-4}xy^{-1}$) | 52 | 16 | 14 | 14 |
| MS(1, $y^{-1}xy^3x^{-1}y^{-1}$) | 10 | 3 | 3 | 3 |
| MS(1, $y^{-1}xy^2x^{-1}y^{-2}$) | 10 | 4 | 4 | 4 |
| MS(1, $y^{-1}xyx^{-1}y^{-3}$) | 21 | 6 | 6 | 6 |
| MS(1, $y^{-1}xy^{-1}x^{-1}y^{-3}$) | 17 | 8 | 8 | 8 |
| MS(1, $y^{-1}xy^{-2}x^{-1}y^{-2}$) | 12 | 6 | 6 | 6 |
| MS(1, $y^{-1}xy^{-3}x^{-1}y^{-1}$) | 19 | 8 | 8 | 8 |
| MS(1, $y^{-1}x^{-1}y^{-2}xy^2$) | 11 | 4 | 4 | 4 |
| MS(1, $y^{-1}x^{-1}y^{-2}xy^{-2}$) | 11 | 6 | 6 | 6 |
| MS(1, $y^{-1}x^{-1}y^{-3}xy$) | 22 | 10 | 10 | 10 |
| MS(1, $y^{-1}x^{-1}y^{-3}xy^{-1}$) | 47 | 14 | 12 | 12 |
| MS(1, $y^{-2}xyx^{-1}y^{-2}$) | 26 | 8 | 8 | 8 |
| MS(1, $y^{-2}xy^{-1}x^{-1}y^{-2}$) | 31 | 12 | 10 | 10 |
| MS(1, $y^{-7}$) | 14 | 8 | 8 | 8 |
| MS(2, $y$) | 7 | 2 | 2 | 2 |
| MS(2, $y^{-1}$) | 6 | 2 | 2 | 2 |
| MS(2, $y^2$) | 11 | 3 | 3 | 3 |
| MS(2, $y^{-2}$) | 6 | 3 | 3 | 3 |
| MS(2, $y^3$) | 7 | 4 | 4 | 4 |
| MS(2, $y^{-3}$) | 6 | 4 | 4 | 4 |
| MS(2, $yxyx^{-1}$) | 32 | 10 | 8 | 11 |
| MS(2, $yxy^{-1}x^{-1}$) | 21 | 7 | 6 | 7 |
| MS(2, $y^4$) | 18 | 5 | 5 | 5 |
| MS(2, $x^{-1}y^{-1}xy$) | 19 | 7 | 6 | 7 |
| MS(2, $x^{-1}y^{-1}xy^{-1}$) | 33 | 10 | 12 | 10 |
| MS(2, $y^{-4}$) | 9 | 5 | 5 | 5 |
| MS(2, $yxy^2x^{-1}$) | 41 | 12 | 12 | 13 |
| MS(2, $yxyx^{-1}y$) | 52 | 13 | 10 | 14 |
| MS(2, $yxyx^{-1}y^{-1}$) | 26 | 8 | 7 | 7 |
| MS(2, $yxy^{-1}x^{-1}y$) | 20 | 8 | 8 | 7 |
| MS(2, $yxy^{-1}x^{-1}y^{-1}$) | 25 | 8 | 7 | 6 |
| MS(2, $yxy^{-2}x^{-1}$) | 16 | 7 | 7 | 6 |
| MS(2, $y^2xyx^{-1}$) | 18 | 7 | 6 | 6 |
| MS(2, $y^2xy^{-1}x^{-1}$) | 11 | 4 | 4 | 4 |
| MS(2, $y^5$) | 17 | 6 | 6 | 6 |
| MS(2, $yx^{-1}y^{-1}xy$) | 23 | 8 | 7 | 7 |
| MS(2, $yx^{-1}y^{-1}xy^{-1}$) | 25 | 8 | 9 | 9 |
| MS(2, $x^{-1}y^{-1}xy^2$) | 21 | 7 | 7 | 7 |
| MS(2, $x^{-1}y^{-1}xy^{-2}$) | 36 | 12 | 13 | 14 |
| MS(2, $x^{-1}y^{-2}xy$) | 8 | 4 | 4 | 4 |
| MS(2, $x^{-1}y^{-2}xy^{-1}$) | 19 | 7 | 6 | 6 |
| MS(2, $y^{-1}xyx^{-1}y^{-1}$) | 23 | 8 | 9 | 7 |
| MS(2, $y^{-1}xy^{-1}x^{-1}y^{-1}$) | 45 | 13 | 15 | 11 |
| MS(2, $y^{-5}$) | 15 | 6 | 6 | 6 |

Table 4 – continued from previous page

| Presentation | GS-AC | GS-Sub | PPO-Sub-DRT | PPO-Sub-DRT + **AC-19** |
|---|---|---|---|---|
| $MS(2,\ yxy^3x^{-1})$ | 16 | 5 | 5 | 5 |
| $MS(2,\ yxy^2x^{-1}y)$ | 55 | 15 | 15 | 16 |
| $MS(2,\ yxy^2x^{-1}y^{-1})$ | 24 | 9 | 10 | 8 |
| $MS(2,\ yxyx^{-1}y^2)$ | 63 | 16 | 13 | 17 |
| $MS(2,\ yxyx^{-1}y^{-2})$ | 19 | 8 | 8 | 7 |
| $MS(2,\ yxy^{-1}x^{-1}y^2)$ | 34 | 11 | 11 | 11 |
| $MS(2,\ yxy^{-1}x^{-1}y^{-2})$ | 28 | 8 | 10 | 8 |
| $MS(2,\ yxy^{-2}x^{-1}y)$ | 27 | 8 | 6 | 7 |
| $MS(2,\ yxy^{-2}x^{-1}y^{-1})$ | 25 | 8 | 8 | 9 |
| $MS(2,\ yxy^{-3}x^{-1})$ | 9 | 4 | 3 | 3 |
| $MS(2,\ y^2xy^2x^{-1})$ | 23 | 7 | 7 | 7 |
| $MS(2,\ y^2xyx^{-1}y)$ | 24 | 7 | 7 | 7 |
| $MS(2,\ y^2xyx^{-1}y^{-1})$ | 16 | 5 | 5 | 5 |
| $MS(2,\ y^2xy^{-1}x^{-1}y)$ | 12 | 5 | 5 | 4 |
| $MS(2,\ y^2xy^{-1}x^{-1}y^{-1})$ | 10 | 3 | 3 | 3 |
| $MS(2,\ y^2xy^{-2}x^{-1})$ | 9 | 4 | 3 | 3 |
| $MS(2,\ y^3xyx^{-1})$ | 70 | 18 | 15 | 19 |
| $MS(2,\ y^3xy^{-1}x^{-1})$ | 45 | 13 | 14 | 14 |
| $MS(2,\ y^6)$ | 19 | 7 | 7 | 7 |
| $MS(2,\ y^2x^{-1}y^{-1}xy)$ | 23 | 8 | 8 | 9 |
| $MS(2,\ y^2x^{-1}y^{-1}xy^{-1})$ | 21 | 8 | 6 | 7 |
| $MS(2,\ yx^{-1}y^{-1}xy^2)$ | 19 | 8 | 8 | 9 |
| $MS(2,\ yx^{-1}y^{-1}xy^{-2})$ | 26 | 9 | 10 | 11 |
| $MS(2,\ yx^{-1}y^{-2}xy)$ | 10 | 3 | 3 | 3 |
| $MS(2,\ yx^{-1}y^{-2}xy^{-1})$ | 9 | 5 | 5 | 5 |
| $MS(2,\ x^{-1}y^{-1}xy^3)$ | 10 | 4 | 3 | 3 |
| $MS(2,\ x^{-1}y^{-1}xy^{-3})$ | 8 | 5 | 5 | 5 |
| $MS(2,\ x^{-1}y^{-2}xy^2)$ | 10 | 4 | 3 | 3 |
| $MS(2,\ x^{-1}y^{-2}xy^{-2})$ | 20 | 7 | 7 | 7 |
| $MS(2,\ x^{-1}y^{-3}xy)$ | 39 | 13 | 14 | 14 |
| $MS(2,\ x^{-1}y^{-3}xy^{-1})$ | 61 | 18 | 20 | 16 |
| $MS(2,\ y^{-1}xy^2x^{-1}y^{-1})$ | 26 | 8 | 8 | 7 |
| $MS(2,\ y^{-1}xyx^{-1}y^{-2})$ | 34 | 11 | 12 | 12 |
| $MS(2,\ y^{-1}xy^{-1}x^{-1}y^{-2})$ | 56 | 16 | 19 | 16 |
| $MS(2,\ y^{-1}xy^{-2}x^{-1}y^{-1})$ | 49 | 15 | 16 | 17 |
| $MS(2,\ y^{-1}x^{-1}y^{-2}xy)$ | 9 | 5 | 5 | 5 |
| $MS(2,\ y^{-1}x^{-1}y^{-2}xy^{-1})$ | 18 | 7 | 7 | 7 |
| $MS(2,\ y^{-6})$ | 15 | 7 | 7 | 7 |
| $MS(2,\ yxy^4x^{-1})$ | 64 | 17 | 15 | 19 |
| $MS(2,\ yxy^3x^{-1}y)$ | 22 | 6 | 6 | 6 |
| $MS(2,\ yxy^3x^{-1}y^{-1})$ | 9 | 4 | 4 | 4 |
| $MS(2,\ yxy^2x^{-1}y^2)$ | 70 | 18 | 19 | 19 |
| $MS(2,\ yxy^2x^{-1}y^{-2})$ | 24 | 8 | 7 | 8 |
| $MS(2,\ yxyx^{-1}y^3)$ | 73 | 19 | 16 | 20 |
| $MS(2,\ yxyx^{-1}y^{-3})$ | 28 | 8 | 10 | 8 |
| $MS(2,\ yxy^{-1}x^{-1}y^3)$ | 43 | 14 | 15 | 15 |
| $MS(2,\ yxy^{-1}x^{-1}y^{-3})$ | 36 | 11 | 11 | 10 |
| $MS(2,\ yxy^{-2}x^{-1}y^2)$ | 27 | 9 | 7 | 9 |
| $MS(2,\ yxy^{-2}x^{-1}y^{-2})$ | 29 | 10 | 11 | 12 |
| $MS(2,\ yxy^{-3}x^{-1}y)$ | 11 | 3 | 3 | 3 |
| $MS(2,\ yxy^{-3}x^{-1}y^{-1})$ | 11 | 4 | 4 | 4 |

Table 4 – continued from previous page

| Presentation | GS-AC | GS-Sub | PPO-Sub-DRT | PPO-Sub-DRT + **AC-19** |
|---|---|---|---|---|
| $MS(2, yxy^{-4}x^{-1})$ | 29 | 9 | 10 | 10 |
| $MS(2, y^2x^2yx^{-2})$ | 74 | 19 | 17 | 19 |
| $MS(2, y^2x^2y^{-1}x^{-2})$ | 53 | 14 | 14 | 15 |
| $MS(2, y^2xy^3x^{-1})$ | 13 | 6 | 6 | 6 |
| $MS(2, y^2xy^2x^{-1}y)$ | 28 | 8 | 8 | 8 |
| $MS(2, y^2xy^2x^{-1}y^{-1})$ | 19 | 6 | 6 | 6 |
| $MS(2, y^2xyx^{-1}y^2)$ | 28 | 8 | 8 | 9 |
| $MS(2, y^2xyx^{-1}y^{-2})$ | 12 | 4 | 4 | 4 |
| $MS(2, y^2xy^{-1}x^{-1}y^2)$ | 25 | 6 | 6 | 6 |
| $MS(2, y^2xy^{-1}x^{-1}y^{-2})$ | 12 | 3 | 2 | 2 |
| $MS(2, y^2xy^{-2}x^{-1}y)$ | 16 | 4 | 4 | 4 |
| $MS(2, y^2xy^{-2}x^{-1}y^{-1})$ | 10 | 3 | 3 | 3 |
| $MS(2, y^2xy^{-3}x^{-1})$ | 8 | 3 | 3 | 3 |
| $MS(2, y^3xy^2x^{-1})$ | 78 | 20 | 21 | 21 |
| $MS(2, y^3xyx^{-1}y)$ | 82 | 21 | 18 | 22 |
| $MS(2, y^3xyx^{-1}y^{-1})$ | 55 | 15 | 12 | 16 |
| $MS(2, y^3xy^{-1}x^{-1}y)$ | 59 | 16 | 17 | 17 |
| $MS(2, y^3xy^{-1}x^{-1}y^{-1})$ | 34 | 10 | 11 | 9 |
| $MS(2, y^3xy^{-2}x^{-1})$ | 41 | 11 | 9 | 12 |
| $MS(2, y^4xyx^{-1})$ | 34 | 10 | 10 | 10 |
| $MS(2, y^4xy^{-1}x^{-1})$ | 24 | 8 | 8 | 8 |
| $MS(2, y^7)$ | 24 | 8 | 8 | 8 |
| $MS(2, y^3x^{-1}y^{-1}xy)$ | 37 | 11 | 9 | 12 |
| $MS(2, y^3x^{-1}y^{-1}xy^{-1})$ | 22 | 8 | 7 | 8 |
| $MS(2, y^2x^{-1}y^{-1}xy^2)$ | 29 | 10 | 10 | 11 |
| $MS(2, y^2x^{-1}y^{-1}xy^{-2})$ | 25 | 8 | 7 | 8 |
| $MS(2, y^2x^{-1}y^{-2}xy)$ | 8 | 3 | 2 | 2 |
| $MS(2, y^2x^{-1}y^{-2}xy^{-1})$ | 9 | 4 | 4 | 4 |
| $MS(2, yx^{-1}y^{-1}xy^3)$ | 11 | 4 | 4 | 4 |
| $MS(2, yx^{-1}y^{-1}xy^{-3})$ | 9 | 4 | 4 | 4 |
| $MS(2, yx^{-1}y^{-2}xy^2)$ | 8 | 3 | 3 | 3 |
| $MS(2, yx^{-1}y^{-2}xy^{-2})$ | 21 | 6 | 6 | 6 |
| $MS(2, yx^{-1}y^{-3}xy)$ | 29 | 10 | 11 | 12 |
| $MS(2, yx^{-1}y^{-3}xy^{-1})$ | 51 | 15 | 17 | 13 |
| $MS(2, x^{-2}y^{-2}x^2y)$ | 44 | 14 | 15 | 15 |
| $MS(2, x^{-2}y^{-2}x^2y^{-1})$ | 64 | 19 | 21 | 17 |
| $MS(2, x^{-1}y^{-1}xy^4)$ | 34 | 9 | 8 | 10 |
| $MS(2, x^{-1}y^{-1}xy^{-4})$ | 58 | 17 | 17 | 14 |
| $MS(2, x^{-1}y^{-2}xy^3)$ | 8 | 3 | 3 | 3 |
| $MS(2, x^{-1}y^{-2}xy^{-3})$ | 11 | 6 | 6 | 6 |
| $MS(2, x^{-1}y^{-3}xy^2)$ | 38 | 11 | 13 | 11 |
| $MS(2, x^{-1}y^{-3}xy^{-2})$ | 71 | 20 | 22 | 22 |
| $MS(2, x^{-1}y^{-4}xy)$ | 21 | 8 | 8 | 8 |
| $MS(2, x^{-1}y^{-4}xy^{-1})$ | 26 | 10 | 10 | 10 |
| $MS(2, y^{-1}xy^3x^{-1}y^{-1})$ | 9 | 3 | 3 | 3 |
| $MS(2, y^{-1}xy^2x^{-1}y^{-2})$ | 28 | 9 | 11 | 9 |
| $MS(2, y^{-1}xyx^{-1}y^{-3})$ | 46 | 14 | 18 | 14 |
| $MS(2, y^{-1}xy^{-1}x^{-1}y^{-3})$ | 66 | 19 | 19 | 18 |
| $MS(2, y^{-1}xy^{-2}x^{-1}y^{-2})$ | 64 | 18 | 19 | 20 |
| $MS(2, y^{-1}xy^{-3}x^{-1}y^{-1})$ | 21 | 6 | 6 | 6 |
| $MS(2, y^{-1}x^{-1}y^{-2}xy^2)$ | 11 | 4 | 4 | 4 |

Table 4 – continued from previous page

| Presentation | GS-AC | GS-Sub | PPO-Sub-DRT | PPO-Sub-DRT + **AC-19** |
|---|---|---|---|---|
| $MS(2, y^{-1}x^{-1}y^{-2}xy^{-2})$ | 20 | 8 | 8 | 8 |
| $MS(2, y^{-1}x^{-1}y^{-3}xy)$ | 52 | 16 | 14 | 18 |
| $MS(2, y^{-1}x^{-1}y^{-3}xy^{-1})$ | 72 | 21 | 23 | 19 |
| $MS(2, y^{-2}xyx^{-1}y^{-2})$ | 21 | 6 | 6 | 6 |
| $MS(2, y^{-2}xy^{-1}x^{-1}y^{-2})$ | 20 | 8 | 8 | 9 |
| $MS(2, y^{-7})$ | 18 | 8 | 8 | 8 |
| $MS(3, y)$ | 8 | 2 | 2 | 2 |
| $MS(3, y^{-1})$ | 7 | 2 | 2 | 2 |
| $MS(3, y^2)$ | 10 | 3 | 3 | 3 |
| $MS(3, y^{-2})$ | 7 | 3 | 3 | 3 |
| $MS(3, y^3)$ | 11 | 4 | 4 | 4 |
| $MS(3, y^{-3})$ | 7 | 4 | 4 | 4 |
| $MS(3, yxy^{-1}x^{-1})$ | 44 | 15 | 14 | 15 |
| $MS(3, y^4)$ | 8 | 5 | 5 | 5 |
| $MS(3, x^{-1}y^{-1}xy)$ | 52 | 17 | 15 | 15 |
| $MS(3, y^{-4})$ | 7 | 5 | 5 | 5 |
| $MS(3, yxy^2x^{-1})$ | 30 | 10 | 14 | 13 |
| $MS(3, yxy^{-1}x^{-1}y)$ | 69 | 22 | 18 | 16 |
| $MS(3, yxy^{-1}x^{-1}y^{-1})$ | 64 | 17 | 18 | 16 |
| $MS(3, yxy^{-2}x^{-1})$ | 25 | 8 | 8 | 7 |
| $MS(3, y^2xyx^{-1})$ | 106 | 29 | 25 | 24 |
| $MS(3, y^5)$ | 23 | 6 | 6 | 6 |
| $MS(3, yx^{-1}y^{-1}xy)$ | 66 | 19 | 17 | 18 |
| $MS(3, x^{-1}y^{-1}xy^2)$ | 22 | 8 | 8 | 8 |
| $MS(3, x^{-1}y^{-1}xy^{-2})$ | 27 | 10 | 16 | 13 |
| $MS(3, x^{-1}y^{-2}xy^{-1})$ | 101 | 29 | 26 | 22 |
| $MS(3, y^{-1}xyx^{-1}y^{-1})$ | 70 | 22 | 19 | 18 |
| $MS(3, y^{-5})$ | 14 | 6 | 6 | 6 |
| $MS(3, yxy^3x^{-1})$ | 125 | 38 | 23 | 23 |
| $MS(3, yxy^2x^{-1}y)$ | 87 | 24 | 17 | 18 |
| $MS(3, yxy^2x^{-1}y^{-1})$ | 34 | 9 | 11 | 10 |
| $MS(3, yxy^{-1}x^{-1}y^2)$ | 116 | 35 | 20 | 21 |
| $MS(3, yxy^{-1}x^{-1}y^{-2})$ | 84 | 24 | 20 | 17 |
| $MS(3, yxy^{-2}x^{-1}y)$ | 27 | 8 | 8 | 7 |
| $MS(3, yxy^{-2}x^{-1}y^{-1})$ | 28 | 9 | 10 | 9 |
| $MS(3, y^2xy^2x^{-1})$ | 66 | 18 | 17 | 17 |
| $MS(3, y^2xyx^{-1}y)$ | 118 | 32 | 26 | 27 |
| $MS(3, y^2xyx^{-1}y^{-1})$ | 98 | 26 | 20 | 22 |
| $MS(3, y^2xy^{-2}x^{-1})$ | 21 | 8 | 8 | 8 |
| $MS(3, y^3xyx^{-1})$ | 21 | 8 | 7 | 7 |
| $MS(3, y^3xy^{-1}x^{-1})$ | 12 | 5 | 5 | 5 |
| $MS(3, y^6)$ | 30 | 7 | 7 | 7 |
| $MS(3, y^2x^{-1}y^{-1}xy)$ | 84 | 24 | 18 | 18 |
| $MS(3, yx^{-1}y^{-1}xy^2)$ | 26 | 9 | 10 | 9 |
| $MS(3, yx^{-1}y^{-1}xy^{-2})$ | 22 | 9 | 10 | 11 |
| $MS(3, yx^{-1}y^{-2}xy^{-1})$ | 90 | 26 | 23 | 20 |
| $MS(3, x^{-1}y^{-1}xy^{-3})$ | 120 | 32 | 26 | 25 |
| $MS(3, x^{-1}y^{-2}xy^2)$ | 25 | 8 | 8 | 8 |
| $MS(3, x^{-1}y^{-2}xy^{-2})$ | 65 | 18 | 17 | 14 |
| $MS(3, x^{-1}y^{-3}xy)$ | 9 | 5 | 5 | 5 |
| $MS(3, x^{-1}y^{-3}xy^{-1})$ | 22 | 8 | 7 | 7 |

Table 4 – continued from previous page

| Presentation | GS-AC | GS-Sub | PPO-Sub-DRT | PPO-Sub-DRT + **AC-19** |
|---|---|---|---|---|
| $MS(3, y^{-1}xy^2x^{-1}y^{-1})$ | 21 | 8 | 9 | 8 |
| $MS(3, y^{-1}xyx^{-1}y^{-2})$ | 107 | 29 | 22 | 21 |
| $MS(3, y^{-1}xy^{-2}x^{-1}y^{-1})$ | 87 | 24 | 19 | 14 |
| $MS(3, y^{-1}x^{-1}y^{-2}xy^{-1})$ | 112 | 32 | 29 | 25 |
| $MS(3, y^{-6})$ | 17 | 7 | 7 | 7 |
| $MS(3, yxy^4x^{-1})$ | 16 | 6 | 6 | 6 |
| $MS(3, yxy^3x^{-1}y^{-1})$ | 114 | 35 | 21 | 22 |
| $MS(3, yxy^2x^{-1}y^2)$ | 134 | 32 | 20 | 20 |
| $MS(3, yxy^2x^{-1}y^{-2})$ | 28 | 8 | 8 | 8 |
| $MS(3, yxy^{-1}x^{-1}y^3)$ | 116 | 37 | 23 | 25 |
| $MS(3, yxy^{-1}x^{-1}y^{-3})$ | 96 | 27 | 24 | 20 |
| $MS(3, yxy^{-2}x^{-1}y^2)$ | 24 | 9 | 9 | 9 |
| $MS(3, yxy^{-2}x^{-1}y^{-2})$ | 50 | 13 | 12 | 12 |
| $MS(3, yxy^{-4}x^{-1})$ | 10 | 5 | 4 | 4 |
| $MS(3, y^2xy^2x^{-1}y)$ | 81 | 21 | 20 | 19 |
| $MS(3, y^2xy^2x^{-1}y^{-1})$ | 54 | 15 | 13 | 15 |
| $MS(3, y^2xyx^{-1}y^2)$ | 131 | 35 | 32 | 31 |
| $MS(3, y^2xyx^{-1}y^{-2})$ | 81 | 23 | 17 | 17 |
| $MS(3, y^2xy^{-2}x^{-1}y)$ | 29 | 9 | 9 | 10 |
| $MS(3, y^2xy^{-2}x^{-1}y^{-1})$ | 26 | 8 | 8 | 7 |
| $MS(3, y^2xy^{-3}x^{-1})$ | 53 | 17 | 15 | 16 |
| $MS(3, y^3xy^2x^{-1})$ | 28 | 9 | 8 | 9 |
| $MS(3, y^3xyx^{-1}y)$ | 29 | 9 | 8 | 9 |
| $MS(3, y^3xyx^{-1}y^{-1})$ | 18 | 7 | 6 | 6 |
| $MS(3, y^3xy^{-1}x^{-1}y)$ | 16 | 6 | 5 | 6 |
| $MS(3, y^3xy^{-1}x^{-1}y^{-1})$ | 16 | 4 | 4 | 4 |
| $MS(3, y^3xy^{-2}x^{-1})$ | 11 | 5 | 4 | 4 |
| $MS(3, y^4xy^{-1}x^{-1})$ | 125 | 39 | 26 | 25 |
| $MS(3, y^7)$ | 26 | 8 | 8 | 8 |
| $MS(3, y^3x^{-1}y^{-1}xy)$ | 95 | 27 | 21 | 23 |
| $MS(3, y^2x^{-1}y^{-1}xy^2)$ | 46 | 13 | 11 | 13 |
| $MS(3, y^2x^{-1}y^{-1}xy^{-2})$ | 21 | 8 | 8 | 8 |
| $MS(3, y^2x^{-1}y^{-2}xy^{-1})$ | 81 | 24 | 19 | 17 |
| $MS(3, yx^{-1}y^{-1}xy^{-3})$ | 108 | 29 | 24 | 22 |
| $MS(3, yx^{-1}y^{-2}xy^2)$ | 25 | 8 | 8 | 8 |
| $MS(3, yx^{-1}y^{-2}xy^{-2})$ | 50 | 15 | 14 | 11 |
| $MS(3, yx^{-1}y^{-3}xy)$ | 10 | 4 | 4 | 4 |
| $MS(3, yx^{-1}y^{-3}xy^{-1})$ | 18 | 7 | 6 | 6 |
| $MS(3, x^{-1}y^{-1}xy^4)$ | 18 | 5 | 4 | 4 |
| $MS(3, x^{-1}y^{-1}xy^{-4})$ | 10 | 6 | 6 | 6 |
| $MS(3, x^{-1}y^{-2}xy^3)$ | 44 | 15 | 15 | 15 |
| $MS(3, x^{-1}y^{-3}xy^2)$ | 13 | 5 | 4 | 4 |
| $MS(3, x^{-1}y^{-3}xy^{-2})$ | 26 | 9 | 8 | 8 |
| $MS(3, x^{-1}y^{-4}xy)$ | 123 | 33 | 27 | 26 |
| $MS(3, y^{-1}xy^2x^{-1}y^{-2})$ | 21 | 9 | 11 | 11 |
| $MS(3, y^{-1}xyx^{-1}y^{-3})$ | 118 | 31 | 26 | 24 |
| $MS(3, y^{-1}xy^{-2}x^{-1}y^{-2})$ | 136 | 29 | 22 | 18 |
| $MS(3, y^{-1}x^{-1}y^{-2}xy^2)$ | 27 | 9 | 10 | 10 |
| $MS(3, y^{-1}x^{-1}y^{-2}xy^{-2})$ | 78 | 21 | 20 | 17 |
| $MS(3, y^{-1}x^{-1}y^{-3}xy)$ | 12 | 6 | 6 | 6 |
| $MS(3, y^{-1}x^{-1}y^{-3}xy^{-1})$ | 25 | 9 | 8 | 8 |

Table 4 – continued from previous page

| Presentation | GS-AC | GS-Sub | PPO-Sub-DRT | PPO-Sub-DRT + **AC-19** |
|---|---|---|---|---|
| $MS(3, y^{-2}xy^{-1}x^{-1}y^{-2})$ | 127 | 35 | 32 | 28 |
| $MS(3, y^{-7})$ | 23 | 8 | 8 | 8 |
| $MS(4, y)$ | 9 | 2 | 2 | 2 |
| $MS(4, y^{-1})$ | 8 | 2 | 2 | 2 |
| $MS(4, y^2)$ | 15 | 3 | 3 | 3 |
| $MS(4, y^{-2})$ | 8 | 3 | 3 | 3 |
| $MS(4, y^3)$ | 13 | 4 | 4 | 4 |
| $MS(4, y^{-3})$ | 8 | 4 | 4 | 4 |
| $MS(4, yxy^{-1}x^{-1})$ | 136 | 41 | 31 | 32 |
| $MS(4, y^4)$ | 12 | 5 | 5 | 5 |
| $MS(4, x^{-1}y^{-1}xy)$ | 140 | 43 | 33 | 31 |
| $MS(4, y^{-4})$ | 8 | 5 | 5 | 5 |
| $MS(4, yxy^{-1}x^{-1}y)$ | 189 | 52 | 34 | 34 |
| $MS(4, yxy^{-1}x^{-1}y^{-1})$ | 179 | 47 | 34 | 35 |
| $MS(4, y^2xy^{-1}x^{-1})$ | 50 | 15 | 13 | 13 |
| $MS(4, y^5)$ | 9 | 6 | 6 | 6 |
| $MS(4, yx^{-1}y^{-1}xy)$ | 179 | 47 | 36 | 34 |
| $MS(4, x^{-1}y^{-2}xy)$ | 48 | 15 | 14 | 14 |
| $MS(4, y^{-1}xyx^{-1}y^{-1})$ | 189 | 52 | 36 | 34 |
| $MS(4, y^{-5})$ | 8 | 6 | 6 | 6 |
| $MS(4, yxy^{-1}x^{-1}y^{-2})$ | 234 | 54 | 37 | 38 |
| $MS(4, y^2xy^2x^{-1})$ | 130 | 21 | 21 | 19 |
| $MS(4, y^2xyx^{-1}y^{-1})$ | 77 | 20 | 16 | 15 |
| $MS(4, y^2xy^{-1}x^{-1}y)$ | 84 | 22 | 15 | 18 |
| $MS(4, y^2xy^{-1}x^{-1}y^{-1})$ | 36 | 12 | 13 | 12 |
| $MS(4, y^2xy^{-2}x^{-1})$ | 28 | 9 | 9 | 9 |
| $MS(4, y^6)$ | 27 | 7 | 7 | 7 |
| $MS(4, y^2x^{-1}y^{-1}xy)$ | 223 | 54 | 38 | 37 |
| $MS(4, yx^{-1}y^{-2}xy)$ | 37 | 12 | 14 | 13 |
| $MS(4, yx^{-1}y^{-2}xy^{-1})$ | 72 | 20 | 17 | 17 |
| $MS(4, x^{-1}y^{-2}xy^2)$ | 26 | 9 | 9 | 9 |
| $MS(4, x^{-1}y^{-2}xy^{-2})$ | 80 | 21 | 21 | 18 |
| $MS(4, y^{-1}x^{-1}y^{-2}xy)$ | 93 | 22 | 17 | 17 |
| $MS(4, y^{-6})$ | 16 | 7 | 7 | 7 |
| $MS(4, y^2xy^3x^{-1})$ | 118 | 27 | 22 | 22 |
| $MS(4, y^2xy^2x^{-1}y^{-1})$ | 72 | 19 | 18 | 17 |
| $MS(4, y^2xyx^{-1}y^{-2})$ | 57 | 17 | 13 | 13 |
| $MS(4, y^2xy^{-1}x^{-1}y^{-2})$ | 39 | 12 | 12 | 12 |
| $MS(4, y^2xy^{-2}x^{-1}y)$ | 33 | 10 | 12 | 11 |
| $MS(4, y^2xy^{-2}x^{-1}y^{-1})$ | 32 | 9 | 10 | 8 |
| $MS(4, y^2xy^{-3}x^{-1})$ | 25 | 9 | 9 | 10 |
| $MS(4, y^4xyx^{-1})$ | 24 | 9 | 8 | 8 |
| $MS(4, y^4xy^{-1}x^{-1})$ | 14 | 6 | 6 | 6 |
| $MS(4, y^7)$ | 36 | 8 | 8 | 8 |
| $MS(4, y^2x^{-1}y^{-2}xy)$ | 40 | 12 | 11 | 11 |
| $MS(4, y^2x^{-1}y^{-2}xy^{-1})$ | 62 | 17 | 14 | 14 |
| $MS(4, yx^{-1}y^{-2}xy^2)$ | 33 | 11 | 9 | 9 |
| $MS(4, yx^{-1}y^{-2}xy^{-2})$ | 73 | 19 | 16 | 16 |
| $MS(4, x^{-1}y^{-2}xy^3)$ | 28 | 9 | 9 | 9 |
| $MS(4, x^{-1}y^{-2}xy^{-3})$ | 118 | 23 | 22 | 21 |
| $MS(4, x^{-1}y^{-4}xy)$ | 10 | 6 | 6 | 6 |

Table 4 – continued from previous page

| Presentation | GS-AC | GS-Sub | PPO-Sub-DRT | PPO-Sub-DRT + **AC-19** |
|---|---|---|---|---|
| $MS(4, x^{-1}y^{-4}xy^{-1})$ | 24 | 9 | 8 | 9 |
| $MS(4, y^{-1}x^{-1}y^{-2}xy^2)$ | 34 | 10 | 12 | 9 |
| $MS(4, y^{-7})$ | 22 | 8 | 8 | 8 |
| $MS(5, y)$ | 10 | 2 | 2 | 2 |
| $MS(5, y^{-1})$ | 9 | 2 | 2 | 2 |
| $MS(5, y^2)$ | 14 | 3 | 3 | 3 |
| $MS(5, y^{-2})$ | 9 | 3 | 3 | 3 |
| $MS(5, y^3)$ | 11 | 4 | 4 | 4 |
| $MS(5, y^{-3})$ | 9 | 4 | 4 | 4 |
| $MS(5, yxy^{-1}x^{-1})$ | 333 | 99 | 67 | 67 |
| $MS(5, y^4)$ | 26 | 5 | 5 | 5 |
| $MS(5, x^{-1}y^{-1}xy)$ | 344 | 101 | 92 | 65 |
| $MS(5, y^{-4})$ | 9 | 5 | 5 | 5 |
| $MS(5, y^5)$ | 13 | 6 | 6 | 6 |
| $MS(5, y^{-5})$ | 9 | 6 | 6 | 6 |
| $MS(5, yxy^3x^{-1})$ | 88 | 23 | 21 | 21 |
| $MS(5, yxy^{-3}x^{-1})$ | 43 | 15 | 16 | 14 |
| $MS(5, y^2xy^{-2}x^{-1})$ | 55 | 17 | 17 | 12 |
| $MS(5, y^6)$ | 10 | 7 | 7 | 8 |
| $MS(5, x^{-1}y^{-1}xy^3)$ | 47 | 15 | 15 | 14 |
| $MS(5, x^{-1}y^{-1}xy^{-3})$ | 82 | 23 | 24 | 20 |
| $MS(5, x^{-1}y^{-2}xy^2)$ | 51 | 18 | 17 | 13 |
| $MS(5, y^{-6})$ | 9 | 7 | 7 | 7 |
| $MS(5, yxy^3x^{-1}y^{-1})$ | 79 | 20 | 18 | 17 |
| $MS(5, yxy^{-3}x^{-1}y)$ | 33 | 12 | 13 | 11 |
| $MS(5, yxy^{-3}x^{-1}y^{-1})$ | 76 | 20 | 19 | 15 |
| $MS(5, y^2xy^{-2}x^{-1}y)$ | 109 | 28 | 20 | 15 |
| $MS(5, y^2xy^{-2}x^{-1}y^{-1})$ | 83 | 22 | 20 | 15 |
| $MS(5, y^2xy^{-3}x^{-1})$ | 32 | 10 | 10 | 9 |
| $MS(5, y^7)$ | 56 | 8 | 8 | 8 |
| $MS(5, yx^{-1}y^{-1}xy^3)$ | 78 | 20 | 18 | 17 |
| $MS(5, yx^{-1}y^{-1}xy^{-3})$ | 64 | 20 | 24 | 15 |
| $MS(5, yx^{-1}y^{-2}xy^2)$ | 82 | 22 | 20 | 16 |
| $MS(5, x^{-1}y^{-2}xy^3)$ | 38 | 12 | 10 | 10 |
| $MS(5, x^{-1}y^{-2}xy^{-3})$ | 93 | 26 | 24 | 22 |
| $MS(5, y^{-1}xy^3x^{-1}y^{-1})$ | 33 | 12 | 13 | 13 |
| $MS(5, y^{-1}x^{-1}y^{-2}xy^2)$ | 113 | 28 | 20 | 18 |
| $MS(5, y^{-7})$ | 18 | 8 | 8 | 8 |
| $MS(6, y)$ | 11 | 2 | 2 | 2 |
| $MS(6, y^{-1})$ | 10 | 2 | 2 | 2 |
| $MS(6, y^2)$ | 19 | 3 | 3 | 3 |
| $MS(6, y^{-2})$ | 10 | 3 | 3 | 3 |
| $MS(6, y^3)$ | 16 | 4 | 4 | 4 |
| $MS(6, y^{-3})$ | 10 | 4 | 4 | 4 |
| $MS(6, y^4)$ | 30 | 5 | 5 | 5 |
| $MS(6, y^{-4})$ | 10 | 5 | 5 | 5 |
| $MS(6, y^5)$ | 33 | 6 | 6 | 6 |
| $MS(6, y^{-5})$ | 10 | 6 | 6 | 6 |
| $MS(6, y^2xy^{-2}x^{-1})$ | 79 | 21 | 17 | 17 |
| $MS(6, y^6)$ | 14 | 7 | 7 | 8 |
| $MS(6, x^{-1}y^{-2}xy^2)$ | 86 | 23 | 17 | 17 |

Table 4 – continued from previous page

| Presentation | GS-AC | GS-Sub | PPO-Sub-DRT | PPO-Sub-DRT + **AC-19** |
|---|---|---|---|---|
| $MS(6, y^{-6})$ | 10 | 7 | 7 | 7 |
| $MS(6, y^2xy^{-2}x^{-1}y)$ | 142 | 36 | 20 | 20 |
| $MS(6, y^2xy^{-2}x^{-1}y^{-1})$ | 117 | 29 | 20 | 20 |
| $MS(6, y^3xy^{-1}x^{-1}y^{-1})$ | 131 | 32 | 24 | 24 |
| $MS(6, y^3xy^{-2}x^{-1})$ | 64 | 17 | 21 | 15 |
| $MS(6, y^7)$ | 11 | 8 | 8 | 9 |
| $MS(6, yx^{-1}y^{-2}xy^2)$ | 120 | 29 | 20 | 20 |
| $MS(6, yx^{-1}y^{-3}xy)$ | 136 | 32 | 26 | 22 |
| $MS(6, x^{-1}y^{-3}xy^2)$ | 57 | 17 | 22 | 16 |
| $MS(6, y^{-1}x^{-1}y^{-2}xy^2)$ | 150 | 36 | 20 | 17 |
| $MS(6, y^{-7})$ | 10 | 8 | 8 | 8 |
| $MS(7, y)$ | 12 | 2 | 2 | 2 |
| $MS(7, y^{-1})$ | 11 | 2 | 2 | 2 |
| $MS(7, y^2)$ | 18 | 3 | 3 | 3 |
| $MS(7, y^{-2})$ | 11 | 3 | 3 | 3 |
| $MS(7, y^3)$ | 17 | 4 | 4 | 4 |
| $MS(7, y^{-3})$ | 11 | 4 | 4 | 4 |
| $MS(7, y^4)$ | 12 | 5 | 5 | 5 |
| $MS(7, y^{-4})$ | 10 | 5 | 5 | 5 |
| $MS(7, y^5)$ | 33 | 6 | 6 | 6 |
| $MS(7, y^{-5})$ | 11 | 6 | 6 | 6 |
| $MS(7, y^2xy^{-2}x^{-1})$ | 180 | 49 | 33 | 34 |
| $MS(7, y^6)$ | 32 | 7 | 7 | 8 |
| $MS(7, x^{-1}y^{-2}xy^2)$ | 174 | 51 | 35 | 33 |
| $MS(7, y^{-6})$ | 11 | 7 | 7 | 7 |
| $MS(7, y^7)$ | 15 | 8 | 8 | 9 |
| $MS(7, y^{-7})$ | 11 | 8 | 8 | 8 |
| $MS(3, yxy^3x^{-1}y)$ | – | 43 | 28 | 28 |
| $MS(3, y^{-1}xy^{-3}x^{-1}y^{-1})$ | – | 37 | 29 | 28 |
| $MS(4, y^2xyx^{-1})$ | – | 25 | 19 | 19 |
| $MS(4, x^{-1}y^{-2}xy^{-1})$ | – | 23 | 20 | 18 |
| $MS(4, yxy^{-1}x^{-1}y^2)$ | – | 73 | 37 | 38 |
| $MS(4, y^2xyx^{-1}y)$ | – | 25 | 22 | 22 |
| $MS(4, y^3xyx^{-1})$ | – | 76 | 47 | 45 |
| $MS(4, x^{-1}y^{-3}xy^{-1})$ | – | 76 | 45 | 47 |
| $MS(4, y^{-1}xyx^{-1}y^{-2})$ | – | 67 | 39 | 37 |
| $MS(4, y^{-1}x^{-1}y^{-2}xy^{-1})$ | – | 25 | 23 | 21 |
| $MS(4, yxy^4x^{-1})$ | – | 81 | 44 | 44 |
| $MS(4, yxy^{-1}x^{-1}y^3)$ | – | 75 | 40 | 41 |
| $MS(4, yxy^{-1}x^{-1}y^{-3})$ | – | 69 | 40 | 41 |
| $MS(4, y^2xy^2x^{-1}y)$ | – | 24 | 24 | 23 |
| $MS(4, y^2xyx^{-1}y^2)$ | – | 28 | 25 | 25 |
| $MS(4, y^2xy^{-1}x^{-1}y^2)$ | – | 28 | 18 | 19 |
| $MS(4, y^3xyx^{-1}y)$ | – | 79 | 50 | 48 |
| $MS(4, y^3xyx^{-1}y^{-1})$ | – | 71 | 44 | 42 |
| $MS(4, y^3x^{-1}y^{-1}xy)$ | – | 69 | 42 | 40 |
| $MS(4, yx^{-1}y^{-3}xy^{-1})$ | – | 71 | 42 | 43 |
| $MS(4, x^{-1}y^{-1}xy^{-4})$ | – | 75 | 46 | 44 |
| $MS(4, y^{-1}xyx^{-1}y^{-3})$ | – | 69 | 42 | 40 |
| $MS(4, y^{-1}x^{-1}y^{-2}xy^{-2})$ | – | 24 | 24 | 22 |
| $MS(4, y^{-1}x^{-1}y^{-3}xy^{-1})$ | – | 79 | 49 | 50 |

Table 4 – continued from previous page

| Presentation | GS-AC | GS-Sub | PPO-Sub-DRT | PPO-Sub-DRT + **AC-19** |
|---|---|---|---|---|
| MS(4, $y^{-2}xyx^{-1}y^{-2}$) | – | 27 | 22 | 19 |
| MS(4, $y^{-2}xy^{-1}x^{-1}y^{-2}$) | – | 28 | 26 | 24 |
| MS(5, $yxy^{-1}x^{-1}y$) | – | 116 | 68 | 70 |
| MS(5, $yxy^{-1}x^{-1}y^{-1}$) | – | 109 | 72 | 73 |
| MS(5, $yxy^{-2}x^{-1}$) | – | 44 | – | 33 |
| MS(5, $yx^{-1}y^{-1}xy$) | – | 109 | – | 70 |
| MS(5, $x^{-1}y^{-1}xy^2$) | – | 45 | – | 78 |
| MS(5, $y^{-1}xyx^{-1}y^{-1}$) | – | 116 | 86 | 70 |
| MS(5, $yxy^{-1}x^{-1}y^2$) | – | 145 | 75 | 73 |
| MS(5, $yxy^{-1}x^{-1}y^{-2}$) | – | 134 | 75 | 79 |
| MS(5, $yxy^{-2}x^{-1}y$) | – | 41 | – | 30 |
| MS(5, $yxy^{-2}x^{-1}y^{-1}$) | – | 48 | – | 36 |
| MS(5, $y^2x^{-1}y^{-1}xy$) | – | 134 | – | 72 |
| MS(5, $yx^{-1}y^{-1}xy^2$) | – | 49 | – | 45 |
| MS(5, $y^{-1}xy^2x^{-1}y^{-1}$) | – | 40 | – | 51 |
| MS(5, $y^{-1}xyx^{-1}y^{-2}$) | – | 139 | – | 72 |
| MS(5, $yxy^4x^{-1}$) | – | 63 | – | 43 |
| MS(5, $yxy^3x^{-1}y$) | – | 38 | 23 | 22 |
| MS(5, $yxy^{-1}x^{-1}y^3$) | – | 156 | 78 | 79 |
| MS(5, $yxy^{-1}x^{-1}y^{-3}$) | – | 141 | 78 | 81 |
| MS(5, $yxy^{-2}x^{-1}y^2$) | – | 52 | – | 29 |
| MS(5, $yxy^{-2}x^{-1}y^{-2}$) | – | 50 | – | 39 |
| MS(5, $y^2xy^3x^{-1}$) | – | 26 | 23 | 22 |
| MS(5, $y^3xy^2x^{-1}$) | – | 47 | 34 | 30 |
| MS(5, $y^4xyx^{-1}$) | – | 153 | – | 83 |
| MS(5, $y^3x^{-1}y^{-1}xy$) | – | 141 | – | 74 |
| MS(5, $y^2x^{-1}y^{-1}xy^2$) | – | 52 | – | 72 |
| MS(5, $x^{-1}y^{-1}xy^{-4}$) | – | 62 | – | 76 |
| MS(5, $x^{-1}y^{-3}xy^{-2}$) | – | 47 | 34 | 29 |
| MS(5, $x^{-1}y^{-4}xy^{-1}$) | – | 153 | 86 | 91 |
| MS(5, $y^{-1}xy^2x^{-1}y^{-2}$) | – | 51 | – | 44 |
| MS(5, $y^{-1}xyx^{-1}y^{-3}$) | – | 150 | – | 78 |
| MS(5, $y^{-1}xy^{-3}x^{-1}y^{-1}$) | – | 46 | 30 | 23 |
| MS(6, $y^3xyx^{-1}$) | – | 64 | 29 | 30 |
| MS(6, $y^3xy^{-1}x^{-1}$) | – | 40 | 25 | 26 |
| MS(6, $x^{-1}y^{-3}xy$) | – | 40 | 28 | 25 |
| MS(6, $x^{-1}y^{-3}xy^{-1}$) | – | 56 | 31 | 30 |
| MS(6, $y^3xy^2x^{-1}$) | – | 39 | 29 | 27 |
| MS(6, $y^3xyx^{-1}y$) | – | 63 | 34 | 33 |
| MS(6, $y^3xyx^{-1}y^{-1}$) | – | 61 | 26 | 27 |
| MS(6, $y^3xy^{-1}x^{-1}y$) | – | 56 | 30 | 29 |
| MS(6, $yx^{-1}y^{-3}xy^{-1}$) | – | 55 | 28 | 26 |
| MS(6, $x^{-1}y^{-3}xy^{-2}$) | – | 48 | 30 | 26 |
| MS(6, $y^{-1}x^{-1}y^{-3}xy$) | – | 56 | 32 | 28 |
| MS(6, $y^{-1}x^{-1}y^{-3}xy^{-1}$) | – | 62 | 34 | 32 |
| MS(7, $yxy^4x^{-1}$) | – | 56 | – | 33 |
| MS(7, $yxy^{-4}x^{-1}$) | – | 40 | 28 | 24 |
| MS(7, $y^2xy^{-2}x^{-1}y$) | – | 81 | 36 | 37 |
| MS(7, $y^2xy^{-2}x^{-1}y^{-1}$) | – | 79 | 36 | 50 |
| MS(7, $yx^{-1}y^{-2}xy^2$) | – | 79 | 37 | 36 |
| MS(7, $x^{-1}y^{-1}xy^4$) | – | 40 | 26 | 25 |

Table 4 – continued from previous page

| Presentation | GS-AC | GS-Sub | PPO-Sub-DRT | PPO-Sub-DRT + **AC-19** |
|---|---|---|---|---|
| MS(7, $x^{-1}y^{-1}xy^{-4}$) | – | 56 | 40 | 42 |
| MS(7, $y^{-1}x^{-1}y^{-2}xy^2$) | – | 83 | 36 | 36 |

## H. Atomic AC-Move Counts

To complement the substitution-step comparisons in the main text, this appendix reports path lengths in terms of primitive AC-moves. Each substitution decomposes into:

- exactly 1 concatenation (AC2),

- 0 or more inversions (AC1),

- 0 to $|r_1| + |r_2|$ conjugations (AC3).

**Aggregate statistics.** Across the 610 Miller–Schupp presentations solved by the hybrid beam search (Section 5), the mean solution comprises 12.4 substitution steps, which expand to a mean of 135.0 primitive AC-moves—an $8.7\times$ amplification ratio. Conjugations dominate the primitive-move budget (mean 96.2, 71% of all atomic moves), followed by inversions (mean 26.4, 19%) and concatenations (mean 12.4, 9%; equal to the substitution count by construction).

*Table 5.* Representative path-length breakdowns by primitive AC-move type for a sample of Miller–Schupp presentations solved by the hybrid beam search, spanning the difficulty range. "Substitutions" = supermove count; "Atomic AC" = total primitive AC-moves; AC1 / AC2 / AC3 are the per-type counts.

| Idx | Substitutions | Atomic AC | AC1 | AC2 | AC3 |
|---|---|---|---|---|---|
| 0 | 2 | 9 | 6 | 2 | 1 |
| 8 | 5 | 24 | 16 | 5 | 3 |
| 18 | 8 | 51 | 14 | 8 | 29 |
| 241 | 15 | 127 | 40 | 15 | 72 |
| 581 | 80 | 1080 | 129 | 80 | 871 |

The sample illustrates that easier presentations have AC3-to-AC2 ratios near 1, while harder presentations require many length-increasing conjugations per substitution: idx 581 needs 871 conjugations across 80 substitution steps (~11 per step), reflecting the deep length-increasing detours characteristic of the hard regime that GS's length-prioritized queue cannot reach.

## I. Hybrid Beam Search Details

We evaluate the trained policy with a fixed-width beam search of width $B = 16{,}384$, run for up to $T = 150$ steps per presentation. At each step every beam is scored by its action log-probability under the policy (log-softmax of the logits); the value head is not used in scoring. Exploration is injected via Gumbel noise, and the top $B$ candidates over all beam-action pairs are retained. Beams are pruned by removing no-op moves and by deduplicating repeated states both within a step and globally across the search. A presentation is counted as solved as soon as any beam reaches the trivial presentation.

