# OpenReview forum: "The Two-Hump Problem: Bridging the Difficulty Gap in Mathematical Reinforcement Learning"
_ICML.cc/2026/Conference — ICML 2026 regular_

### Official Review · Reviewer_yMgF · 2026-03-11

**Soundness:** 2
**Presentation:** 2
**Significance:** 2
**Originality:** 2
**Overall Recommendation:** 4
**Confidence:** 3

**Summary:**

This paper studies reinforcement learning for mathematical search through the Andrews–Curtis conjecture and argues that the main obstacle is a bimodal “Two-Hump” difficulty distribution: most instances are either trivially solvable or effectively unreachable, while the intermediate hard-but-solvable region is sparse. To address this issue, the paper proposes three main ingredients: substitution supermoves as a more appropriate action space than primitive AC moves, a Dual-Ring Transformer tailored to cyclic relator structure and large structured action spaces, and new data-generation pipelines that produce the AC-19 and AC-1M datasets to populate the difficulty gap. On the Miller–Schupp benchmark, the proposed PPO-based approach substantially improves over the previously reported PPO baseline, while greedy search with substitutions remains stronger in total number of solved instances. The paper also claims concrete mathematical progress by trivializing additional benchmark instances and compressing the remaining unsolved set into fewer equivalence classes. Overall, the paper is ambitious, technically interesting, and potentially impactful as a case study of RL for mathematical reasoning.

**Compliance With Llm Reviewing Policy:**

Affirmed.

**Final Justification:**

The rebuttal from authors usefully clarifies several technical points, especially the distinction between presentation length and per-relator input length, the computational budget, and the intended interpretation of the unresolved 550 instances. I find these clarifications partially address my concerns and improve my confidence in the technical soundness of the implementation. However, the main empirical limitation remains: the RL method still does not demonstrate actual solve-level progress on the hardest unresolved region, and the normalization across action spaces is still not fully resolved in the current submission. Therefore, I am somewhat more positive after rebuttal, but not enough to fully remove my reservations. Thus, I raise the score from 3 to 4.

**Key Questions For Authors:**

1. The paper states that no training run solved benchmark instances outside the initially known AC-trivial subset. Can the authors clarify whether the proposed RL method makes any measurable partial progress on the remaining unresolved cases, even if it does not fully solve them? For example, does it reduce canonical length, discover novel equivalences, or improve search trajectories under larger budgets?
2. Many of the reported gains are measured in substitution steps or node counts under the substitution action space. Can the authors provide a normalized comparison in terms of primitive AC operations, or at least give a principled mapping between substitution cost and atomic AC cost? This would make the efficiency claims much easier to interpret.
3. The manuscript states that AC-1M contains presentations of length at most 30, while the Dual-Ring Transformer uses a maximum input length of 24 per relator. Can the authors explain precisely how these definitions relate and how overlength cases are handled in training and evaluation?
4. Since AC-19 and AC-1M are generated and verified using search procedures closely related to the proposed solver, to what extent might the training distribution be biased toward structures favored by that solver family? Have the authors performed any cross-validation with substantially different verification procedures?
5. Can the authors provide more detail on compute usage and experimental cost, including hardware, wall-clock time, and training stability across seeds? These details are important for evaluating reproducibility and practical significance.

**Limitations:**

A major limitation of the current work is that the strongest claims are not yet matched by success in the hardest unresolved region of the benchmark. The method improves substantially over prior PPO baselines and often finds shorter solutions than greedy search, but it does not yet demonstrate that learning has crossed into the genuinely hard regime that defines the second hump. This makes the current contribution more of an important step forward than a decisive breakthrough.

The generality of the conclusions is also somewhat limited. Although the paper argues that the AC problem is a representative mathematical RL domain, it remains unclear how much of the proposed machinery transfers to other theorem-search or algebraic reasoning tasks beyond this specific group-presentation setting. The inductive biases may be highly effective here while being less generally applicable.

**Strengths And Weaknesses:**

Strengths：

The paper presents a compelling diagnosis of why reinforcement learning is difficult in this domain. The “Two-Hump” perspective is one of the most interesting aspects of the work, because it turns the usual sparse-reward discussion into a more structural claim about the distribution of instance difficulty and the resulting failure of standard curriculum strategies. This framing is intuitive, empirically motivated, and useful for future work on mathematical search problems. The proposed Dual-Ring Transformer is another strong point. Its use of cyclic relative positional encoding and cross-attention is well aligned with the problem structure, and the comparison against a canonical-form variant suggests that the architectural inductive bias is genuinely useful rather than cosmetic. The dataset contribution is also significant. AC-19 and AC-1M appear to be valuable resources for future work, especially because they are designed to populate the hard-but-solvable region that is otherwise underrepresented. If released in a reproducible form, these datasets could become an important benchmark contribution independent of the specific PPO results.

Weaknesses:

The main empirical weakness is that the learning-based method still does not appear to make progress on the hardest unresolved benchmark region. The paper explicitly notes that training never solved any instance outside the initially known AC-trivial subset, which substantially weakens the broader claim that the method bridges the hard region in a meaningful sense. In other words, the method improves success rate and path quality on instances already known to be solvable, but does not yet demonstrate clear traction on the truly difficult “second hump.”  A second weakness is the comparability of evaluation metrics across action spaces. Since substitution supermoves compress multiple primitive AC moves into one step, comparisons based on node counts or path lengths are not fully calibrated unless the paper also reports an atomic-cost view. Without such normalization, some efficiency claims are difficult to interpret. There are also reproducibility concerns. While many PPO hyperparameters are given, several practically important details remain unclear or unavailable during review, including full artifacts, runtime budgets, compute cost, and a more complete description of how length constraints and preprocessing interact with the datasets. This is particularly important because the paper positions itself as both an algorithmic and benchmark contribution.

---

> ### Author Rebuttal · Authors · 2026-03-30
>
> We thank Reviewer 4 for a detailed and thoughtful review. We particularly appreciate the careful engagement with both the strengths and limitations of our work. We address each concern below.
>
> ### Q1: Does RL make partial progress on the unresolved hard instances?
>
> Defining "partial progress" on unresolved instances is inherently difficult since reducing presentation length does not necessarily bring us closer to trivialization, as the path may require passing through longer states. With that caveat:
>
> 1. **Equivalence class discovery:** Our methods reduced the 550 unsolved presentations to 261 equivalence classes, focusing future efforts on fewer distinct candidates.
>
> 2. **PPO explores fundamentally different state space than GS.** On solved presentations, PPO finds substantially shorter paths (Figure 7), learning length-increasing strategies GS cannot discover. On hard instances like AK(3), the most studied potential counterexample, GS only reaches length 26 due to its length-based priority, while PPO explores lengths 30+, a region GS effectively never visits. We plan to use PPO's value function as a GS search heuristic and will report results for the camera-ready version (see Reviewer 3, Q1, for our full analysis).
>
> 3. **The 550 unsolved presentations may not be AC-trivial at all:** they are potential counterexamples. No algorithm can solve them; this may reflect mathematical reality. The appropriate comparison is PPO vs. prior PPO: 610 vs. 457, a substantial improvement considering that many additional solves are much harder than those in the 457. The AC conjecture has been open 60+ years; our tools lay groundwork for future progress.
>
> ### Q2: Normalized comparison in primitive AC operations
>
> Each substitution = 1 concatenation (AC2), 0–1 inversion (AC1), and up to $|r_1|+|r_2|$ conjugations (AC3). Moreover, with canonical form, we mod out by the cyclic symmetry, inherently reducing the state space. We never compare path lengths across action spaces (unfair); we compare *node usage.* The goal of substitutions and canonical form is to explore the space more efficiently, and the $1600\times$ gain is in nodes explored. We will translate to primitive AC moves and add other metrics (like wall clock time) in the revision for completeness, and the key claims (PPO shorter paths; GS saturates) hold regardless.
>
> ### Q3: Presentation length (30) vs. Transformer input length (24)
>
> These are different notions of length. **Presentation length** = $|r_1| + |r_2|$ (sum of both relators); **input length** = 24 tokens per relator (padding). Since the minimum non-trivial relator length in AC-1M is 5, individual relators rarely approach 24. We excluded the <1% where a relator exceeds 24; during training, moves exceeding this limit are blocked but the agent still receives a reward signal. We will clarify in the text.
>
> ### Q4: Dataset bias toward the solver family
>
> 1. **AC-19** is an exhaustive enumeration (Havas & Ramsay, 2000); GS only enters for AC-triviality filtering. This could introduce some bias, but on the MS benchmark PPO never solved anything GS could not, suggesting AC-19 provides a broad difficulty range.
>
> 2. **AC-1M** uses automorphisms: mathematical transformations not used by any solver. Solution paths traced through automorphisms are completely different from solver paths. No solver bias exists.
>
> 3. **External evaluation:** All Table 1 results use the Miller-Schupp benchmark from prior work, constructed independently. Improvements here demonstrate generalization.
>
> ### Q5: Compute usage and experimental cost
>
> We will add full details to the appendix. Key numbers: PPO training (250M interactions) takes ~11 hours on a single AMD Instinct MI210 GPU. GS (10M-node budget) takes at most 60 min/presentation on AMD EPYC 7502P with 60 subprocesses. Training stability: 607.2 $\pm$ 2.2 over 5 seeds (range: 605–610). Generator-solver training: ~36 hours. Dataset generation: AC-19 ~20,000 CPU-hours, AC-1M ~2,900 CPU-hours.
>
> ### Broader concern: No progress on the hardest instances
>
> See Q1 above, which addresses this concern in detail.
>
> ### Generality of conclusions
>
> The Two-Hump diagnosis, generator-solver framework, and cyclic encodings transfer directly to domains with bimodal difficulty, symmetry-based augmentation, or cyclic structure. We give concrete examples from knot theory, SAT, and number theory in our response to Reviewer 1, Q1. Substitution supermoves are AC-specific, but the principle of designing action spaces that respect mathematical structure is broadly applicable. AC is uniquely suited as an ML testbed because it admits a discrete, efficiently-checkable search formulation where RL can be rigorously evaluated. We will strengthen this discussion in the revision.

---

> > ### Author Rebuttal · Reviewer_yMgF · 2026-04-02
> >
> > The author’s reply has cleared up most of my doubts; I will consider increasing my score.

---

### Official Review · Reviewer_mKDq · 2026-03-11

**Soundness:** 3
**Presentation:** 3
**Significance:** 3
**Originality:** 3
**Overall Recommendation:** 4
**Confidence:** 1

**Summary:**

This paper applies reinforcement learning to the mathematical search problem posed by the Andrews–Curtis conjecture. The authors first identify a key structural challenge: existing problem instances tend to be either too easy or too difficult, which makes effective learning hard. To address this, they propose a novel data-generation framework in which a generator is rewarded for producing valid instances that are sufficiently challenging, while a solver attempts to solve the generated instances using greedy search. The paper also introduces supermoves, which are composite sequences of Andrews–Curtis moves that reduce the effective size of the state space. To better capture the cyclical structure of relators, the authors design Transformer-based architectures tailored to this setting. Experimentally, the proposed architecture combined with PPO outperforms several baselines, and the paper also releases new datasets for Andrews–Curtis problems.

**Compliance With Llm Reviewing Policy:**

Affirmed.

**Final Justification:**

This paper is empirically strong and has extensive experiments to support its claim, the rebuttal solves my confusions. Hence I decided to maintain positive score.

**Key Questions For Authors:**

1. Why does PPO underperform Greedy Search in terms of the number of presentations solved? More specifically, why is PPO able to find shorter solution paths, yet still solve fewer presentations overall than Greedy Search?

**Limitations:**

Yes.

**Strengths And Weaknesses:**

Strengths:
The paper is empirically strong, with extensive experiments that support its main claims. It is clearly written, well organized, and provides enough detail for reproducibility. The work is significant in identifying a key bottleneck in the AC problem and proposing both a data-generation framework and state-space reduction methods to address it. The paper is also original in proposed methods, and release of a new dataset. The gain of solving 153 more presentations is especially compelling.

Weaknesses:
The paper does not fully explain some important empirical behaviors, such as why PPO finds shorter solution paths but solves fewer presentations than Greedy Search. More discussion is also needed on how these empirical improvements translate into actual progress on the Andrews–Curtis conjecture itself.

---

> ### Author Rebuttal · Authors · 2026-03-30
>
> We thank Reviewer 3 for their positive and encouraging assessment, and for the insightful question about PPO vs. GS. We address it below.
>
> ### Why does PPO find shorter paths but solve fewer presentations than GS?
>
> This is an excellent question that gets at a fundamental difference between the two approaches.
>
> **GS is a massively parallel search** (10M-node budget, millions of candidate paths) while **PPO is a single learned policy** (one decision per step, 96-step horizon, no backtracking or memory). GS succeeds when a solution exists that does not require large length increases; PPO must learn a general strategy across all presentations, but when it succeeds it finds paths reflecting genuine mathematical insight and hence the shorter paths.
>
> **Why shorter paths?** PPO's value function learns to evaluate states beyond immediate length reduction, identifying length-*increasing* moves that lead to easier trivialization. GS, by contrast, prioritizes shorter presentations in its queue, so it tends to find longer, more circuitous paths.
>
> **GS has hit a fundamental ceiling, while PPO continues to improve.** Increasing the GS budget from 1M to 10M nodes yields zero additional solves. This is not merely a computational limitation: GS prioritizes by presentation length, so it struggles to discover paths requiring large length-increasing detours, which is exactly what hard instances demand. It is difficult to see how GS progresses further without incorporating lessons from learned approaches, a pattern that recurs across domains (e.g., Go). PPO does not face the same limitation and is still improving with better data and architecture.
>
> **Combining the two is the natural next step.** Reviewer 1 suggested running GS with a small expansion budget after each PPO-proposed substitution. This is interesting, though implementing it fruitfully is challenging: if PPO gets close enough for small-budget GS to finish, PPO could likely finish itself; if we need large-budget GS, efficiency suffers. Another promising direction, which we have begun exploring, is to use the value function learned by PPO as an alternative heuristic for GS, replacing the default length-based queue priority. Preliminary experiments show this produces shorter paths than GS alone while retaining the systematic exploration benefits of GS. We will run these experiments fully and report results for the camera-ready version.
>
> We will add this discussion to the paper.
>
> ### How do empirical improvements translate to progress on the AC conjecture?
>
> Our work makes concrete mathematical progress in several ways:
>
> 1. **New trivializations:** We trivialized 107 previously unsolved Miller-Schupp presentations and each such solution constitutes a new rigorous result. Moreover, it was another conjecture of Andrews and Curtis that substitution moves are sufficient for trivializations, and our results provide a perspective on this.
> 2. **Equivalence class reduction:** Among the 550 remaining unsolved presentations, we discovered AC-equivalences reducing them to 261 distinct equivalence classes with minimal representatives. This focuses future efforts on fewer, genuinely distinct candidates.
> 3. **Progress regardless of the conjecture's truth:** If AC is true, every trivialization contributes to a proof. If AC is false, trivializations help *narrow the search for counterexamples* by eliminating candidates (see our response to Reviewer 2, "Significance," for existing disproval strategies). Either way, the datasets and methods we release enable further progress by the community.
>
> We will expand this discussion in the revision. (See also our response to Reviewer 2, "Significance," for additional context on the topological implications.)

---

> > ### Author Rebuttal · Reviewer_mKDq · 2026-04-03
> >
> > Thank you to the authors for their thoughtful responses to my concerns. I am satisfied with their clarifications and intend to maintain my positive score.

---

### Official Review · Reviewer_AMVT · 2026-03-13

**Soundness:** 3
**Presentation:** 3
**Significance:** 3
**Originality:** 3
**Overall Recommendation:** 4
**Confidence:** 3

**Summary:**

The authors notice the Two-Hump distribution in the Andrews-Curtis conjecture (especially in MS series potential counterexamples), use automorphism perturbation to generate datapoints to populate the difficulty gap, and designed a new transformer structure to utilize the mathematical structure. They achieve better performance over baselines, and release two large-scale hard datasets.

**Compliance With Llm Reviewing Policy:**

Affirmed.

**Final Justification:**

My main concerns were about the significance of the problem and some details in the experimental comparison. The authors’ rebuttal answered these questions well.I also read the other reviewers’ comments, and overall I recognize the contribution of this work.

**Key Questions For Authors:**

1. Are the methods listed in table1 previous sota?
2. What's the inference budget of PPO agents in Table 1 in evaluation? Why does PPO agent perform poorer than greedy search methods (607.2 vs. 640)?

**Limitations:**

yes (last paragraph of conclusion)

**Strengths And Weaknesses:**

**Soundness**
Strength: The experiments are solid and convincing. The paper clearly illustrate the two-hump distribution of the AC problem, then design specific techniques (automorphism perturbation, substitution, dual-ring transformer) to address it. The experiments shows each component is effective.
Weakness: The dual-ring transformer structured PPO performs poorer than greedy search with substitution.

**Significance**
I'm not sure about if the Andrews-Curtis conjecture is a significant or well known problem. Since the paper's main methods are designed specially for this particular problem, I will see other reviewer's comment to better understand it. For now I believe it is quite significance for making breakthrough on this particular problem.


**Presentation**
The paper is generally well written. In Figure 1, the statement “Most presentations are either trivial or practically impossible” should refer to the “MS series”? This expression needs to be clarified.

**Originality**
The paper use a unique data generation process automorphism perturbation, and shrink the state space by using substitution. These are not new mathematical concepts but are used in RL for the first time.

---

> ### Author Rebuttal · Authors · 2026-03-30
>
> We thank Reviewer 2 for their careful reading and thoughtful feedback. We appreciate the time and effort invested in reviewing our work. We hope the clarifications below address the concerns raised.
>
> ### Q1: Are the methods in Table 1 previous SOTA?
>
> Yes. PPO-AC-ResNet (457 presentations solved) and GS-AC at 1M nodes (533 presentations solved) are the best previously reported results from Shehper et al. (2025), which is the prior state of the art. Our methods represent a very large improvement over both: PPO-Sub-DRT + AC-19 solves 607.2 presentations on average, a gain of 150 over the PPO baseline, which is a 33% relative improvement. GS-Sub solves 640, a gain of 107 over the GS baseline (20% relative improvement). To put this in perspective, each additional solve beyond the prior SOTA is a harder mathematical result than any that came before it, so the marginal difficulty increases steeply. We will make this clearer in the table caption.
>
> ### Q2: PPO inference budget and why PPO solves fewer presentations than GS
>
> **PPO inference:** PPO training uses 250M environment interactions. The results in Table 1 reflect the best solve count achieved over the course of training across 5 seeds. As mentioned in the text, the first 640 parallel environments (out of 2380) are reserved for the 640 presentations from the benchmark known to be AC-trivial.
>
> **Why the gap:** GS and PPO operate in fundamentally different regimes. GS is a search algorithm exploring up to 10M nodes per presentation, while PPO is a single learned policy making one decision per step with no lookahead. The 607 vs. 640 gap reflects that PPO has not yet matched GS's raw coverage, but PPO finds substantially shorter solution paths (Figure 7), demonstrating it learns genuine mathematical strategies that GS struggles to discover. Crucially, GS has hit a fundamental ceiling: increasing the budget from 1M to 10M nodes yields zero additional solves. We view these as complementary approaches and plan to combine them for the camera-ready version, specifically, using PPO's learned value function as a search heuristic for GS. (See our response to Reviewer 3, Q1, for a detailed analysis of GS's limitations and the hybrid approach.)
>
> ### Figure 1 clarification: "Most presentations are either trivial or practically impossible"
>
> In Figure 1, we give a general picture of the Two-Hump phenomenon, which is *not* limited to the MS series. We demonstrate in Section 3.2 (Figure 3/AC-19) that the same bimodal distribution persists across a much broader enumeration of all short presentations. We will revise the Figure 1 caption to specify that it is a general phenomenon, not only applying to the MS series.
>
> ### Significance of the Andrews-Curtis conjecture
>
> The AC conjecture is one of the most prominent open problems in combinatorial group theory, routinely appearing at the top of open-problem lists in the field, and appearing in Kirby's influential problem list in low-dimensional topology. Its significance extends well beyond group theory: Freedman, Gompf, Morrison, and Walker (Quantum Topology, 2010) showed that every potential AC counterexample yields a potential counterexample to the Smooth Poincaré Conjecture in dimension 4 (SPC4), a major open problem in the Poincaré conjecture family, two cases of which led to Fields medals (Freedman, Perelman). Validating such counterexamples would also disprove the Generalized Property R conjecture. There is evidence both for and against SPC4, making the fate of AC less certain than often assumed, and there exist disproval strategies for both conjectures for which strong candidate presentations are essential, and our elimination of false counterexamples directly supports this effort.
>
> Unlike many other elite open problems, AC admits an immediate, discrete, efficiently-checkable formulation as a pure search problem, exactly the setting where RL can be rigorously evaluated and scaled. This makes it not merely a representative problem, but one of the strongest possible testbeds for ML-based mathematical discovery. We discuss the broader applicability of our techniques in our response to Reviewer 1, Q1.

---

> > ### Author Rebuttal · Reviewer_AMVT · 2026-04-04
> >
> > Thank you to the authors for the helpful rebuttal. The clarifications resolved my concerns, and I am now convinced of the importance of the Andrews-Curtis conjecture and the paper’s contribution. I will increase my score.

---

### Official Review · Reviewer_tWii · 2026-03-17

**Soundness:** 3
**Presentation:** 4
**Significance:** 2
**Originality:** 3
**Overall Recommendation:** 5
**Confidence:** 3

**Summary:**

The paper addresses the Andrews-Curtis (AC) conjecture, a long-standing open problem in group theory which asserts that a specific graph only has one connected component. To this end, authors present a novel method for trivialization, i.e. guiding the path from a given graph node to a special node marked as initial corresponding to the trivial presentation of a specific group. Once such a path is found for every initial node, the conjecture will be proven true.
The new method improves upon existing Machine Learning-based approaches for trivialization, significantly pushing the state-of-the-art performance on the Miller-Schupp benchmark. This is achieved with several technical improvements. First, authors identify the two-hump problem, meaning that most training trajectories are either too easy or impossible to solve, with no medium-difficulty examples. This is partially resolved by training a Generator network to cleverly transform trivial problems into those that are harder for greedy search to solve. Second, authors introduce a novel custom architecture called Dual-Ring Transformer. Third, authors present a more efficient method for navigating the search graph using so-called substitution moves.
Although the best performance on the Miller-Schupp benchmark is still achieved using greedy search, the ML-based search yields shorter paths, identifying a promising avenue for future work.

**Compliance With Llm Reviewing Policy:**

Affirmed.

**Final Justification:**

Authors addressed my questions well in their rebuttal.

**Key Questions For Authors:**

Can you provide motivation for why the AC conjecture is a good representative problem in mathematical search, in the sense of how improvements in trivialization are more broadly useful in ML-based approaches to mathematical problems?

Can you explain how improvements in trivialization help in solving the AC conjecture? If the widely accepted belief that AC is false turns out to be correct, trivialization is not possible for some presentations.

Can you justify why ML-based methods are useful for trivialization, since they are outperformed by greedy search?

Will you release the introduced datasets and/or your codebase?

**Limitations:**

Yes

**Strengths And Weaknesses:**

Note on notation: By [104l] and [048r], I mean line 104 left and line 48 right, respectively.

*Soundness*:
The newly proposed method is correctly evaluated on a standard benchmark across multiple ablations.
Areas of improvement:
The basic premise of "the difficulty valley" presented in [048r] is intuitively correct, but the justification in Sections 3.1 and 3.2 is not satisfactory. Specifically, [048r] states that the "impossible" presentations are not solvable by any current algorithm. However, Sections 3.1 and 3.2 only evaluate the greedy algorithm, leaving open the possibility that other algorithms might be able to solve some portion of the "impossible" presentations.
In Figure 6, the Generator approach shifts most paths to the "unsolved" bucket, which does not alleviate the difficulty valley problem. On the contrary, the bucket 220 decreases in size after applying the Generator, seemingly pronouncing the problem. This should be addressed in the text.
Since the learned PPO approach excels in finding short paths while GS achieves better total performance, an important experiment would be to attempt to combine both approaches. For example, GS could be run with some minimal expansion budget after each Transformer-generated substitution. The hope is to utilize the Transformer to discover "creative" length-increasing detours together with the raw power of GS.
In Figure 7, it should be noted whether the reported path lengths in supermove-based methods (e.g. GS-Sub) is reported in terms of supermoves or after translating the supermoves back to the underlying simple substitutions.

*Presentation*:
The paper is well-structured and covers most important aspects of the presented method.
Areas of improvement:
In Section 2, the reader can be expected to know what generators of a group are, but not what relators are. Relators should be briefly explained.
In Figures 5 and 6, the legend and the axis labels should have larger font to increase readability.
Figures 1 and 6 use path length on the x axis, while Figures 3 and 4 use the number of explored nodes on the x axis. This might be correct, but the motivation for this should be explained. This discrepancy is most visible between Figure 1, which introduces the reader to the "two-hump" problem, and Figures 3 and 4, which empirically verify it.
In Figure 6 or its caption, it should be explained how the path length buckets are chosen. Presumably, bucket "10" includes all lengths at most 10, bucket 40 all lengths between 10 and 40, etc., and bucket 220 all lengths at least 190 and at most 10K. This should be stated explicitly. Figure 7 could serve as an inspiration.
[317l] Why were only 200 presentations evaluated? If it's due to compute limitations, it should be stated. Is it possible to obtain some estimate of how the difficulty is transformed over the whole AC-19 corpus?
[370l] "... with a horizon length of 96 before environment reset, ..." - It's unclear what this means.
Minor points:
The subsection titles "Contributions" [045r] and Related Work [055r] should be formatted as subsections.
[141r], [134r], [236r], [366r] - superfluous parentheses around the citation

*Significance*:
Significance of the presented approach should be more justified on three levels. First, more explanation should be provided for why the AC conjecture is the correct problem to work on, in the sense of how approaches developed for solving it might be useful more broadly in ML-based methods for mathematical search. For example, is the two-hump problem typical in mathematical search beyond AC?
Second, it's unclear how improvements in trivialization methods (e.g., as measured by the Miller-Schupp benchmark) help in solving the AC conjecture. Presumably, for AC to be decided in this way, all potential counterexamples would have to be successfully trivialized. However, there are infinitely many potential counterexamples, and furthermore, unsuccessful trivialization within a given compute budget provides no information on whether we have found a true counterexample or the algorithm simply isn't strong enough. Since AC is widely believed to be false (as per Wikipedia), meaning that true counterexamples exist, it's unclear why trivialization methods are the right way forward. This should be explained in the paper.
Third, the role of ML-based methods on the AC conjecture should be explicitly justified, since it's outperformed by a greedy search.
As a final point, it should be made clear whether and how authors plan to release the newly introduced datasets or the source code.

*Originality*:
The paper builds on top of previous work which trained a ResNet-based architecture using PPO to generate trivialization paths. Authors present several technical improvements, significantly improving state-of-the-art performance on the Miller-Schupp benchmark.
Authors correctly position their paper within existing work.

---

> ### Author Rebuttal · Authors · 2026-03-30
>
> We thank Reviewer 1 for a thorough and constructive review.
>
>
> ### Q1: Why is the AC conjecture a good representative problem for ML-based mathematical search?
>
> AC is one of the most prominent open problems in combinatorial group theory, with deep connections to low-dimensional topology (see Q2 below). Unlike many other famous open problems, it admits an immediate, discrete, and efficiently-checkable formulation as a pure search problem, exactly where RL can be rigorously evaluated and scaled. Beyond its mathematical significance, the AC conjecture exhibits several properties shared by many important problems:
>
> 1. **Infinite state space with no easy data generation:** The AC graph is infinite and one cannot work backward from the goal state (as in Rubik's cube), forcing principled data generation strategies.
>
> 2. **The Two-Hump phenomenon generalizes:** Bimodal difficulty appears in knot theory (crossing-number-increasing moves), SAT (phase transitions), and number theory (elliptic curve ranks/BSD conjecture). We will expand this in the revision.
>
> 3. **Sparse rewards with no obvious guiding heuristic:** The only signal is eventual trivialization, with no intermediate heuristic reliably indicating progress. This is shared by theorem proving, program synthesis, and combinatorial optimization.
>
> ### Q2: How do improvements in trivialization help solve the AC conjecture?
>
> We appreciate this important question. We discuss the deep connections between AC and major open problems in low-dimensional topology (SPC4, Generalized Property R) in detail in our response to Reviewer 2, "Significance of the Andrews-Curtis conjecture."
>
> 1. **Our framing is as an ML benchmark, not a conjecture-proving tool.** Our main goal was to establish that RL can understand and exploit the mathematical structure of this problem despite extremely sparse rewards (see Q3). This is vindicated by PPO finding shorter paths than GS, making the case for further development where classical solvers hit their limits.
>
> 2. **The belief that AC is false is not settled.** There is evidence both for and against (see Reviewer 2 response for details). Disproval strategies exist for which strong candidate presentations are essential and our elimination of false counterexamples directly supports this effort.
>
> 3. **Trivialization makes progress regardless.** 107 potential counterexamples from the MS series were shown to be AC-trivial, and 550 remaining ones were organized into 261 equivalence classes. This is useful for future research in both trivialization and disproval, and PPO's optimal paths deserve particular interest. Any failure to trivialize a potential counterexample, such as AK(3), can be due to it actually being a true one.
>
> ### Q3: Why are ML-based methods useful if outperformed by greedy search?
>
> GS and PPO have fundamentally different strengths. GS has hit a fundamental ceiling (1M to 10M nodes yields zero additional solves), while PPO finds substantially shorter paths (Figure 7), uncovering genuine mathematical structure: length-increasing strategies that GS cannot discover. Our goal was not to beat GS on raw solve count, but to uncover this structure. Combining the two is the natural next step (see "Combining GS and PPO" below). We provide a detailed analysis of GS's limitations, PPO's complementary strengths, and the hybrid approach in our response to Reviewer 3, Q1.
>
> ### Q4: Dataset and code release
>
> **Yes.** Upon acceptance we will release AC-19, AC-1M, Miller-Schupp equivalence class classification, and the full codebase. We will add an explicit statement in the paper.
>
> ### Combining GS and PPO
>
> Excellent suggestion. We discuss the tradeoffs and a promising alternative (using PPO's value function as GS's heuristic) in our response to Reviewer 3, Q1. Preliminary experiments show shorter paths than GS alone. We will report full results for the camera-ready version.
>
> ### Figure 6: Generator shifting presentations to "unsolved"
>
> "Unsolved" means unsolved by GS within a 10K-node budget (the "easy" boundary). All these presentations are still AC-trivial with known solution paths; a larger budget (e.g., 10M) would solve many. The Generator deliberately creates GS-hard presentations in the 10K–1M range, precisely the training data PPO needs. The shift is expected and desirable.
>
> ### Figure 7: Path lengths in supermoves vs. primitive moves
>
> Path lengths are in **substitution steps** (supermoves), each a composition of AC1, AC2, and AC3 moves. We will clarify this in the caption and add a table of atomic AC-move counts in the appendix.
>
> ### Sections 3.1–3.2: "Impossible" claim
>
> We will soften to "effectively unsolvable by current methods" and note the additional algorithms tested.
>
> ### Presentation improvements
>
> We will address all items in the revision: explain relators; increase font sizes; explain x-axis choices; add bucket definitions; clarify "horizon length of 96" (episode terminates after 96 steps); fix formatting and citations.

---

> > ### Author Rebuttal · Reviewer_tWii · 2026-04-03
> >
> > Thank you for the rebuttal, I am increasing my score from "weak accept" to "accept"

---

### Decision · Program_Chairs · 2026-04-30

**Decision:**

Accept (regular)

**Comment:**

The paper applies RL to the 60-year-old Andrews-Curtis (AC) conjecture in mathematics. The authors identify a "Two-Hump" problem: problem instances are either trivially easy or virtually impossible. There is a lack of the "hard-but-solvable" data for effective RL training. In this paper, they propose a novel data generation technique, introduce the "supermoves" technique to reduce state space, and design a custom "Dual-Ring Transformer" architecture. They also release two new large-scale datasets, AC-19 and AC-1M.

After the rebuttal the scores were generally positive. Reviewers praised the three contributions and assess them as novel and valuable. There were a number of concerns regarding the importance of the AC, comparison with GS, hard instances, and reproducability. Reviewers were satisfied with the rebuttal.